# Ecosystem age-class dynamics and distribution in the LPJ-wsl v2.0 global ecosystem model

Leonardo Calle[1,2] and Benjamin Poulter[3]

*Correspondence to*: Leonardo Calle (leonardo.calle@umontana.edu)

1 University of Montana, Department of Forest Management, WA Franke College of Forestry and Conservation, Missoula, MT 59812

2 Montana State University, Department of Ecology, Bozeman, Montana 59717, USA

3 NASA Goddard Space Flight Center, Biospheric Science Laboratory, Greenbelt, Maryland 20771, USA

*Correspondence to*: Leonardo Calle (leonardo.calle@umontana.edu)

**Abstract.** Forest ecosystem processes follow classic responses with age, peaking production around canopy closure and declining thereafter. Although age dynamics might be more dominant in certain regions over others, demographic effects on net primary production (NPP) and heterotrophic respiration (Rh) are bound to exist. Yet, explicit representation of ecosystem demography is notably absent in many global ecosystem models. This is concerning because the global community relies on these models to regularly update our collective understanding of the global carbon cycle. This paper aims to present the technical developments of a computationally-efficient approach for representing age-class dynamics within a global ecosystem model, the LPJ-wsl v2.0 Dynamic Global Vegetation Model, and to determine if explicit representation of demography influenced ecosystem stocks and fluxes at global scales or at the level of a grid cell. The modeled age classes are initially created by simulated fire, and prescribed wood harvesting or abandonment of managed land, otherwise aging naturally until an additional disturbance is simulated or prescribed. In this paper, we show that the age-module can capture classic demographic patterns in stem density and tree height compared to inventory data, and that simulated patterns of ecosystem function follow classic responses with age. We also present two scientific applications of the model to assess the modeled age-class distribution over time and to determine the demographic effect on ecosystem fluxes relative to climate. Simulations show that, between 1860 and 2016, zonal age distribution on Earth was driven predominately by fire, causing a 45-60 year difference in ages between older boreal (50N-90N) and younger tropical (23S-23N) ecosystems. Between simulation years 1860 and 2016, land-use change and land management were responsible for a decrease in zonal age by -6 years in boreal and by -21 years in both temperate (23N-50N) and tropical latitudes, with the anthropogenic effect on zonal age distribution increasing over time. A statistical model helped reduced LPJ-wsl v2.0 complexity by predicting per-gridcell annual NPP and Rh fluxes by three terms: precipitation, temperature and age class; at global scales, $R^2$ was between 0.95 and 0.98. As determined by the statistical model, the demographic effect on ecosystem function was often less than 0.10 kg C m$^{-2}$ yr$^{-1}$ but as high as 0.60 kg C m$^{-2}$ yr$^{-1}$ where the effect was greatest. In eastern forests of North America, the simulated demographic effect was of similar magnitude, or greater than, the effects of climate; simulated demographic effects were similarly important in large regions of every vegetated continent. Simulated spatial datasets are provided for global ecosystem ages and the estimated coefficients for effects of precipitation, temperature and demography on ecosystem function. The discussion focuses on our finding of an increasing role of demography in the global carbon

cycle, the effect of demography on relaxation times (resilience) following a disturbance event and its implications at global scales, and a finding of a 40 Pg C increase in biomass turnover when including age dynamics at global scales. Whereas time is the only mechanism that increases ecosystem age, any additional disturbance not explicitly modeled will decrease age. The LPJ-wsl v2.0 age-module represents another step forward towards understanding the role of

demography in global ecosystems.

## 1 Introduction

Forest ecosystem production follows predictable patterns with time since disturbance. Classic forest age-production curves from Odum (1969) suggest that net ecosystem production (NEP) peaks around canopy closure, declining thereafter due to hydraulic limitations on gross primary production (Ryan et al. 2004, Drake et al. 2010, 2011) and increases

in heterotrophic respiration from biomass turnover due stand-level declines in population density (Ptrezsch and Biber 2005, Stephenson et al. 2014). That younger forests are more productive than older forests has been long-standing knowledge in forestry, as evidenced by yield and growth tables dating back to the 18[th] Century that incorporated stand age into their calculations of lumber production (Pretzsch et al. 2008).

On global scales, forest age is a considerable factor in the global carbon cycle and comprises a large fraction of the total land carbon sink, which is estimated at $3.2 \pm 0.8$ Pg C $yr^{-1}$ for years 2008-2017 (Le Quere et al. 2018). According to country-level forest inventories, net carbon uptake from post-disturbance tropical forest regrowth is $1.6 \pm 0.5$ Pg C $yr^{-1}$ from 1990 to 2007 (Pan et al. 2011a). Although the timeframes for estimates of the total lank sink and the inventory-based regrowth flux do not perfectly overlap, the magnitude of the regrowth sink relative to the total land sink

warrants that models take regrowth dynamics into account. A multi-model global regrowth analysis with demographically-enabled DGVMs, for which LPJ-wsl v2.0 contributed, estimated that post-disturbance regrowth comprised a large global regrowth sink of 0.3 to 1.1 Pg C $yr^{-1}$ due to demography alone over years 1981-2010 (Pugh et al. 2019a). In the last decade, explicit model representation of forests as a function of time since disturbance (hereafter simply, 'ecosystem age') has been a grand challenge in an effort to quantify the demographic response of forests to changes

in climate, atmospheric $CO_2$, , land-use change and landmangement (LUCLM) and fire (Friend et al. 2014, Kondo et al 2018, Pugh et al. 2019a). Much of the focus of these global modeling studies has been on the effect of natural and anthropogenic disturbances on the carbon dynamics in old-growth versus second-growth forests (Gitz and Ciais 2003, Shevliakova et al. 2009, Kondo et al 2018, Yue et al. 2018, Pugh et al. 2019a), but lack finer distinction of demographic effects at different ageclasses. Following a call to the science community to improve demographic representation in

models (Fisher et al. 2015), there is now a growing list of global models that are capable of simulating global ecosystem demographics (Gitz and Ciais 2003, *Model*: OSCAR; Shevliakova et al. 2009, *Model*: LM3V; Haverd et al. 2014, *Model*: CABLE-POP; Lindeskog et al. 2013, *Model*: LPJ-GUESS; Yue et al. 2018, *Model*: ORCHIDEE MICT; Nabel et al. 2019, *Model*: JSBACH4), although more models need the capability to represent landscape heterogeneity in forest structure and function.


Much of the evidence for the relative importance and global distribution of large disturbances has come from either satellite retrievals of spectral indices indicating forest loss or burn scars on the land (Potter et al. 2003, Frolking et al. 2009, Pugh et al. 2019b), national forest inventory records of land-use change and forest management (Houghton 1999, FAO-FRA 2015, Williams et al. 2016), or from model-based studies (Goldewijck 2001, Arneth et al. 2017) that integrate information on historical land use (Goldewijck 2001, Hurtt et al. 2006). Other natural disturbances such as pest and pathogen outbreaks, flooding, ice storms, and volcanic eruptions are less widespread globally (Frolking et al. 2009) but are still influential drivers of landscape age-class dynamics (Dale et al. 2001, Turner 2010). In the coterminous United States, forest management is the predominant forest disturbance (1.4% of forested area converted to non-forest and then re-established annually), followed by fire (0.01-0.5% of forested area burned annually 1997-2008) (Williams et al. 2016). Although pest and pathogens, namely bark beetle infestations, affected a much larger area (up to 6% of total forested area in U.S.) than both logging and fire, their effects do not always cause immediate tree mortality. It is arguable whether fire and LUCLM are the two most important global drivers of ecosystem age (Pan et al. 2011a), but nevertheless these are the drivers applied in a model framework in this study, in a manner that moves modeling one step forward to assess global age-class dynamics.

The overall aims of this study are to present new model developments that simulate the time-evolution of age-class distributions in a global ecosystem model and to determine if explicit representation of demography in this model influenced ecosystem stocks and fluxes at global scales or at the level of a grid cell. Technical details are presented for a module representing age-class dynamics, driven by fire feedbacks, land abandonment and wood harvesting in the LPJ-wsl v2.0 Dynamic Global Vegetation Model (DGVM). Prior versions of LPJ-wsl v2.0 that included early developments of the land-use change module and the age-class module have already contributed to previous studies (Arneth et a. 2017, Kondo et al. 2018, Pugh et al. 2019a). Analyses of model behavior, in terms of age-structure and age-functional patterns, the temporal evolution of age distributions and their causative drivers, and a statistical model of ecosystem production and respiration as a function of demography and climate are presented.

## 2 Methods

### 2.1 LPJ-wsl v2.0 General Model Description

#### 2.1.1 LPJ History

LPJ-wsl v2.0 has its legacy in the LPJ family of models, first developed by Sitch et al. (2003) in a Fortran coding environment [1]. In 2007, Bondeau et al. (2007) produced the LPJmL codebase, in C, which included the addition of 'managed lands'. The model known as LPJ-wsl v2.0 is based on LPJmL v3.0, but includes modifications to managed lands that now includes modeling gross land cover transitions, forest age cohorts, and also a modification that include permafrost and wetland methane; the permafrost and wetland modules were not used in this study. Many developments were made in the publicly-available LPJmL4 (version 4.0; Schaphoff et al. 2018) that are not present in LPJ-wsl v2.0.

---

1. LPJ and LPJmL History, https://www.pik-potsdam.de/research/projects/activities/biosphere-water-model-ling/lpjml/history-1)

The LPJ-wsl v2.0 model was branched off of LPJmL sometime around 2010 and continued to diverge. This research

paper represents a large effort toward this end, and the LPJ-wsl v2.0 code is now freely and publicly available (https://github.com/benpoulter/LPJ-wsl_v2.0) under a GNU Affero General Public License version 3. LPJ-wsl v2.0, excluding the version number unless an explicit reference is being made to prior versions or to clarify the version number.

### 2.1.2 LPJ-wsl v2.0 Overview

LPJ-wsl v2.0 simulates soil hydrology and vegetation dynamics in 0.5˚ grid cells, wherein climate, atmospheric $CO_2$, and soil texture is prescribed from driver datasets (Figure 1). Vegetation is categorized into Plant Functional Types (PFT; Box 1996). Plant populations compete for light, space, and soil water, depending on demand; nutrient cycles are not considered in this model version. LPJ-wsl v2.0 is a 'big-leaf' ecosystem model, whereby leaf-level photosynthesis and respiration (Haxeltine and Prentice 1996, Farquhar et al. 1980) occur at daily time-steps, accounting for the

photosynthetically active period (daytime), and are scaled to the stand-level using a mean-individual approximation, which assumes that important state variables (carbon stocks and fluxes) can be determined by using the average properties of a population. Plant populations are categorized using 10 PFTs in this study (phenology parameters and bioclimatic limits listed in SM Table 1); the same PFTs as in Sitch et al. (2003). Left unchanged are the PFT-specific bioclimatic limits, turnover rates, C:N tissue ratios, allometric ratios, and other parameters not explicitly commented

on here, but as described in Sitch et al. (2003). Mortality occurs as in the original version of LPJ, "...as a result of light competition, low growth efficiency, a negative annual carbon balance, heat stress, or when PFT bioclimatic limits are exceeded for a period of time" (Sitch et al. 2003). The fire module and the representation of land-use change and land management are described in detail in *Section 2.2.2*, as these modules require a greater number of modifications for integration with age classes.

## 2.2 Age-class Module

### 2.2.1 An age-based model of ecosystems – sub-grid cell dynamics

Age classes are represented as 'patches' within a grid cell (Figure 1). Every age class has the same climate, atmospheric $CO_2$, and soil texture, but the properties of the age class, such as available soil water and light availability, are determined by feedbacks from plant demand within an age class. Plant processes (competition, photosynthesis, respi-

ration) are simulated at the level of the age class for each PFT within the age class.

The age-class module has a fixed number of age classes that can be represented in a grid cell, but all age classes are not always represented. Age classes are classified into 12 age classes in fixed age width bins, defined as the *unequalbin* or the *10yr-equalbin* age-width setup (Table 1). Each age class contains *within* age class elements, which are simply a vector representation of areas for each age-unit in the age class. The within-ageclass elements are not independent

and every within-ageclass element has the same state variables, including the same soil water and light. As such, we only simulate processes at the ageclass level, and the within-ageclass elements are a simple method for a 'smooth' transition between ageclass. In theory, we can simulate processes independently for each within-ageclass element, but

this is not practical or necessary. The main benefit for using equal-bin or unequal-bin ageclasses is to independently simulate processes. The age widths of the age classes in the *10yr-equalbin* setup correspond to common age widths

of classes used in forest inventories; for contrast, JSBACH4 uses a 15-year age width in their equal-bin ageclass setup. Most ageclasses in this setup are represented by a vector of 10 elements, wherein each element represents an aerial fraction for each age-unit (Table1). The *10yr-equal*bin age setup is used for all simulations including the global simulation, whereas the *unequalbin* setup is used for regional and single grid cell simulations; simulation details in next section. The use of equal or unequal age class setups is more than just for reporting purposes. Resources available to

plants (space, light, soil water) differ between age classes but not within age classes, and we limit the model to represent a total of 12 ageclasses only. Also, there exists a greater range of forest ages at global scales and the equal-bin age class setup allows us to independently model resource dynamics for more of the terrestrial surface. If we had chosen the unequal-bin setup for global simulations, we would be independently modeling processes only for the youngest age classes and we would lose capacity to independently model processes at intermediate and older age

classes. A study by Nabel et al. (2020), using the demographically-enabled JSBACH4 DGVM, found that unequal binning of age widths had lower errors than equal age width binning but the largest reduction in model-observation error was achieved by simply adding more ageclasses at younger ages, regardless of the binning strategy employed.

Age classes are only created by disturbance and we only model the following disturbances: fire, wood harvest, or land abandonment, which initialize a new, youngest age class. The fraction of the age class that burns gets its age 'reset'

to the youngest age class, 1-10 yr. The same process occurs for the fractional area that undergoes wood harvest or when managed land is abandoned and allowed to regrow – the fractional area undergoing an age-transition is reclassified as a 1-10 yr age class. This process allows the model to accurately track the carbon stocks, fluxes and feedbacks associated with these state variables. For example, if a fire burns 50% of an age class, then 50% might have bare ground and 50% will have vegetation at pre-burn levels. If the probability of another fire is dependent on live vegeta-

tion, then feedbacks will result in a lower chance of fire on the bare-ground fraction versus the fully-vegetated fraction that was not previously burned.

The most novel advancement in this study is a new method of age class transition modeling, which we call 'vectortracking of fractional transitions' (VTFT), which improves the computational efficiency of modeling age classes in global models; this is a similar approach independently conceived by Nabel et al. (2019). The method is a transparent

and simple solution to the problem of dilution, which manifests as an advective process when state variables, such as carbon stocks or tree density, are made to merge by area-weighted averaging. The concept of merging two unique age classes on the basis of similarity is a computational solution to constrain the number of simulated age classes in accordance with computer resources, but can be considered ecologically unrealistic. For example, along what axis of similarity is an age class considered to be most similar to another age class – in terms of PFT composition, biomass

in plant organs, plant height, or stem density? Existing age class models (Medvigy et al. 2009, *Model*: ED2; Lawrence et al. 2019, *Model*: CLMv5.0; Yu et al. 2018, *Model*: ORCHIDEE-MICT) employ merging rules (although some do not – Lindeskog et al. 2013, *Model*: LPJ-GUESS) with varying thresholds to ensure that age classes are only merged if the difference among one state variable (biomass, tree height) is less than a fixed threshold. We also merge age

classes, but we do not employ merging rules along arbitrary axes of similarity. We fix the number of age classes *a*

*priori*, similar to LPJ-GUESS in that there is a maximum number of age classes. Instead of forced merging to reduce computational burden (as in ED2), a fraction of the age class always transitions to an older state, and a fractional area can transition and merge with the next oldest age class. By design, VTFT allows age classes to advance in a natural progression from young to old and ensures that age-class transitions always occur between the most similar age classes along multiple state variables.

The theoretical description of the VTFT approach is described as following in matrix notation. VTFT describes a matrix of size (*w* := *agewidths per ageclass*, *n* := ageclasses), where the elements $f_{i,j}$ are the within-ageclass fractional areas of the grid cell:

$$\mathbf{F} = \begin{bmatrix} f_{1,1} & f_{1,2} & \cdots & f_{1,n} \\ f_{2,1} & f_{2,2} & \cdots & f_{2,n} \\ \vdots & \vdots & \ddots & \vdots \\ f_{w,1} & f_{w,2} & \cdots & f_{w,n} \end{bmatrix} = \left( f_{i,j} \in \mathbb{R}^{w \times n} \right) \tag{1}$$

It is important to note here that within-ageclass fractional areas ($f_{i,j}$) are only used during age-class transitions – this is a key point. For almost all calculations in LPJ, processes operate on the total fractional area for each age class,

$$F_{total} = \sum_{j=1}^{n} \sum_{i=1}^{w} f_{i,j}$$

$$F\_total_j = \sum_{i=1}^{w} f_{i,j} \tag{2}$$

, where F_total is the sum of fractional areas for all grid cell age classes, defined as the sum of fractional areas for over age classes (n) and age widths (w). F_total$_j$ is the column sum of **F** for a given age class (*j*); the calculation can be vectorized for efficiency by computing the dot product between an 'all-ones' row vector of length (w) and **F**. In practice, when LPJ-wsl v2.0 simulates physical processes on an arbitrary carbon pool (C), for example, the calcula-tions are computed on a per-mass basis, which then requires conversion to a per-area basis by multiplying the total

carbon mass in an age class by the representative total fractional area:

$$C_j[\text{kg m}^{-2}] = C_j[\text{kg}] \times F\_total_j \tag{3}$$

, where $C_j$ [*units* := kg or km$^{-2}$] is the total carbon for a given age class (*j*). Again, the calculation can be computed via the Hadamard (element-wise) product, taking a vector ($\vec{C}$), where elements are the carbon pool totals for every age class and multiplying by vector F_total, with elements of the total fractional areas in each age class. In effect, all simulated processes in LPJ-wsl v2.0 act on an area-basis, based on the column sums of **F**.

In every year of simulation, an age-class transition always occurs, and this procedure is defined as an operation that increments the positions of the elements as,

$$\mathbf{F}^{(t+1)} := \begin{bmatrix} f_{1,1}^{(t+1)} \stackrel{\text{def}}{=} f_{new}^{(t+1)} & f_{1,2}^{(t+1)} \stackrel{\text{def}}{=} f_{w,1}^{(t)} & \cdots & f_{1,n}^{(t+1)} \stackrel{\text{def}}{=} f_{w,n-1}^{(t)} \\ f_{2,1}^{(t+1)} \stackrel{\text{def}}{=} f_{1,1}^{(t)} & f_{2,2}^{(t+1)} \stackrel{\text{def}}{=} f_{1,2}^{(t)} & \cdots & f_{2,n}^{(t+1)} \stackrel{\text{def}}{=} f_{1,n}^{(t)} \\ \vdots & \vdots & \ddots & \vdots \\ f_{w,1}^{(t+1)} \stackrel{\text{def}}{=} f_{w-1,1}^{(t)} & f_{w,2}^{(t+1)} \stackrel{\text{def}}{=} f_{w-1,2}^{(t)} & \cdots & f_{w,n}^{(t+1)} \stackrel{\text{def}}{=} f_{w,n}^{(t)} + f_{w-1,n}^{(t)} \end{bmatrix} \tag{4}$$

, where the superscripts are the time indices for the current timestep (t+1) and the previous timestep (t), subscripts are the matrix indices, $f_{new}^{(t+1)}$ is the fractional area of a newly created stand (by definition, it is the youngest age-class fraction), and $f_{w,n}$ is the oldest fractional age of the grid cell, which is incremented by an amount equal to fractional area ($f_{w-1,n}^{(t)}$). Of special importance is the bottom row of the **F** matrix, $F_{w,1\leq j\leq n}$, which are the fractional areas of each age class transitioning to the next oldest age class. The transitioning fractions ($F_{w*}$) become the incoming fractions in the next-oldest age class. Using an arbitrary carbon pool (C) as an example, the carbon pool for the next timestep (t+1) would be calculated via an area-weighted average between the carbon remaining in the age class and the carbon in the transitioning fraction,

$$C_j^{(t+1)} = \frac{\left(C_j^{(t)} \times F'\_total_j^{(t)}\right) + \left(C_{j-1}^{(t)} \times f_{w,j-1}^{(t)}\right)}{F'\_total_j^{(t)} + f_{w,j-1}^{(t)}} \tag{5}$$

, where $F'\_total_j$ is the total fractional area of age class ($j$) that remains in the age class, $f_{w,j-1}^{(t)}$ is the transitioning or 'incoming' fraction from the younger age class, and $C_{j-1}^{(t)}$ is the carbon pool (on area-basis, kg m$^{-2}$) in the younger age class, calculated at the end of the previous timestep. Equation 5 effectively converts the units of the carbon pools from an area-basis (km m$^{-2}$) to a total mass (kg), taking the sum of the carbon remaining and transitioning into the age class, and 'redistributes' the carbon mass by the new fractional area; during age-class transitions, these area-weighted averages are used to conserve mass across all state variables. In theory, VTFT minimizes the redistribution (or 'dilution') of mass across a larger area if the incoming fractional area is much smaller than the fractional area of the existing age class.

In a plain-language summary of the matrix representation, VTFT ensures that a vector of fractional areas is associated with every age class ($n$), of length ($w$), and where 'w' is equal to the age width of the age class, with elements ($f$) that are the fractional areas contributing to the total fractional area of the age class ($F\_total$). When a young age class ($a_1$) is first created, VTFT vectors are initialized to zero and the first element ($f_1$) is set to the *incoming* fractional area. The following is a description for within-class and between-class transitions. *Within-class Fractional Transitions*: For every simulation year, the position of each element ($f_x$) in the VTFT vector is incremented by the representative time of each element ($x$), which is simply 1. No changes occur to the state variables of the age class during within-class transitions. *Between-class Fractional Transitions*: Upon incrementing the position of each element in the VTFT vector, if the value at $f_w$ is non-zero then the corresponding fractional area $f_w$, defined as the outgoing fraction, is used in an area-weighted average between the state variables of $a_1 f_w$ and the next oldest age class $a_2 F\_total$. Upon incrementing the element position, if all elements in the VTFT vector of the preceding age class are zeros then the age class is simply deleted from computational memory.

Two hypothetical scenarios are provided in Figure 2 that demonstrate age-class transitions using the VTFT procedure when there is a young age class created, and when there are fractional age-class transitions between age classes. With VTFT, any number of age classes and age widths can be modeled, but it is demonstrated in this study that the age widths employed in this study are sufficient to minimize the dilution of state variables when area-weighted averaging is used to merge fractional age classes while also simulating stand-age patterns in state variables of carbon stocks,

stem density and fluxes.

**2.2.2 Integration with fire and land-use change and land management (LUCLM) modules**

The disturbance processes of simulated fire, land use change, and land management can occur on multiple age classes at a time. That is, these processes are related but independent. For instance, fire can occur independently on each age class, and each age class would have its own independent estimate of the probability of fire. Wood harvest occurs first

on the oldest age class and progressively harvests younger age classes until two demands are met (harvest area and harvest biomass); described in detail in the relevant section below. Clearly, each process influences the other as logging or fire both remove biomass that could be potential fuel for a future fire or biomass for a future harvest. These relationships are not evaluated here, but are noted for its potential importance. Below, we describe in detail the integration of the age-class module with those the two prominent forms of disturbance: fire, land use change, and land

management.

*Fire* – The fractional area burned initiates the creation of a youngest age class, or it gets merged with a youngest age class if one exists already. Fire simulation is based on the semi-empirical Glob-FIRM model by Thonicke et al. (2001), with implementation details described in Sitch et al (2003). In short, fire is dependent on the length of the fire season,

calculated as the number of dry days in a year above a threshold and a minimum fuel load, defined only as the mass of carbon in litter. When a fire occurs, PFT-specific fire resistances determine the fraction of the PFT population that gets burned. The biomass of burned PFTs, along with the aboveground litter in the age class, gets calculated as an immediate flux to the atmosphere. The fraction of the PFT population that does not burn maintains state variables (e.g., tree height, carbon in leaf and wood) at previous values. It is possible to have so called 'survivor' trees on the

youngest age class that then skews the age-height distribution of the age class. The model does not assume any structure of survivor trees. Instead, survivor trees occur as a function of the underlying process. For example, if a fire occurs on a stand, but the fire does not burn all the PFTs, then there will be survivor PFTs on the stand. Both fire and wood harvest (below) are simulated based on fractional area, and it is the fractional area, specifically, that gets reset to a young age class.


*LUCLM* – Age classes get created when managed land (i.e., crop/pasture) is abandoned and allowed to regrow into secondary forests, or when wood harvest occurs on forested lands and causes deforestation. In both cases, the fractional area abandoned/logged initiates the creation of a youngest age class, or it gets merged with a youngest age class if one exists already. To improve accounting of primary forests, defined here as natural land without a history of LUCLM,

and second-growth forests, defined as natural land with a history of LUCLM; transitions between these classes are

unidirectional from primary to secondary. In the LUCLM module, gross transitions between land uses are simulated (Pongratz et al. 2014, Stocker et al. 2014), such that if the fraction of abandoned land equals the fraction of land deforested in the same year (net zero land-use change) then the fluxes from the gross transitions are tracked independently and give an overall more accurate accounting (and higher magnitude) of emissions from LUC than if we only tracked net transitions (Arneth et al. 2017).

General rules distinguishing primary and secondary stands within the age class context stem from the Land Use Harmonization dataset version 2 (LUHv2; Hurtt et al. 2020), but with the following modifications so that the LUHv2 data can be used in LPJ-wsl v2.0: (1a) the primary grid cell fraction only decreases in size and never gets mixed with existing secondary forests or with abandoned managed land. Only fire creates young age classes on primary lands. (2a) secondary grid cell fractions can be mixed with other secondary forest fractions, recently abandoned land, fractions with wood harvest, and recently burned area. General priority rules for deforestation and wood harvest: (1b) For simplicity, deforestation (i.e., land-use change) always occurs in the ranking of oldest to youngest age class, proceeding to deforest each age class until the prescribed fractional area of deforestation is met. This rule will always result in greater land-to-atmosphere fluxes than if rules were employed that allowed younger age classes to be preferentially deforested. (2b) wood harvest (i.e., biomass harvest) also occurs in the ranking of oldest to youngest age class until two conditions are met. Timber harvest occurs on each age class until a prescribed harvest mass or harvest area is met.

Treatment of immediate emissions and residues: Deforestation results in 100% of heartwood biomass and 50% of sapwood biomass being stored for delay emission in product pools; root biomass is entirely part of belowground litter pools, while 100% leaf and 50% of sapwood biomass becomes part of aboveground litter pools. Grid cell fractions that underwent land-use change were not mixed with existing managed lands or secondary fractions until all land-use transitions had occurred. This avoids a computational sequence that results in a lower flux if deforestation and abandonment occur in the same year. For wood harvest, 100% of leaf biomass and 40% of the sapwood and heartwood enters the aboveground litter pools, and 100% of root biomass the belowground litter pools; 60% of sapwood and heartwood are assumed to go into a product pool for delayed emission.

Timber from deforestation and harvest in product pools for delayed emission (Earles et al. 2012): For deforestation, 60% of exported wood (i.e., not in litter) goes into a 2-yr product pool and 40% goes into a 25-yr product pool, following the 40:60 efficiency assumption from McGuire et al. (2001). For wood harvest, the model uses space-time explicit data on harvest fractions going into roundwood, fuelwood and biofuel product pools. We use three product pools and assume that 100% of the fuelwood and biofuel fraction goes into the 1-year product pool (emitted in the same year of wood harvest), 50% of the roundwood fraction goes into the 10-year product pool (emitted at rate 10% per year) and the remaining 50% of the roundwood fraction goes into the 100-year product pool (emitted at rate 1% per year).

**2.3 Experimental Design and Analysis**

### 2.3.1 Model inputs

Inputs to the model are gridded soil texture (sand, silt, clay fractions) from the USDA Harmonized World Soils Dataset v1.2 (Nachtergaele et al. 2008), annually-varying global-mean $[CO_2]$ (time series available in supplement), and monthly-varying air temperature, precipitation, precipitation frequency, and radiation from the Climate Research Unit (CRU, version TS3.26) data for 1901-2016. Land use, land-use change, and wood harvest were prescribed annually based on the Land Use Harmonization dataset version 2 (LUHv2; Hurtt et al. 2020), which is used as forcing land-use for the 6th Coupled Model Intercomparison Project (CMIP6; Eyring et al. 2016). The dataset includes fractional area of bi-directional (gross) land-use transitions between forested and managed lands, as well as the total biomass of wood harvest on a specified fractional area logged. In LPJ-wsl v2.0, managed lands (i.e., crop/pasture) are treated as grass-lands with no irrigation, no fire, and tree PFTs were not allowed to establish. Model representation of land management is an oversimplification to focus on effects of wood harvest.

### 2.3.2 Examining age dynamics: qualitative evaluation of regional simulations against U.S. Forest Inventory Analysis (FIA) data to assess simulated changes in stand structure and ecosystem function

*U.S. Forest Inventory and Analysis (FIA)* – The FIA dataset is freely available at the FIA DataMart web portal (FIADB version 1.6.0.0.2), accessed 2 February 2016. The FIA plot level data are composed of 4 circular sub-plot sample areas (168 m$^2$), wherein attributes of all trees with Diameter at Breast Height (DBH) $\geq$ 5.0 inches (12.7 cm) diameter are recorded. We extracted variables that capture two main axes of structural change as a function of forest age: stem density and tree height. Spatial coordinates of sample plots are 'fuzzed' with imposed error for privacy reasons (FIA User Guide v 6.02; O'Connell et al. 2015). For purposes of this analysis, plot data were aggregated to the spatial scale of U.S. Forest Service *Divisions* (SM Figure 2; USFS Divisions are delineated by regional-scale precipitation levels and patterns as well as temperature) minimizing co-location concerns between model-observation comparisons. We filtered the FIA data based on the following criteria. We only included plots that used the national standard plot design (DESIGNCD=1) and were located on forested land (COND_STATUS=1) with no history of major disturbance, stock-ing, or logging (DSTRBCD=0, TRTCD1=0), which could alter natural patterns of tree density versus age and con-found the comparison to simulated data. We also only included plots that had both sub-plot samples of live tree (STA-TUSCD=1) stem density and also circular micro-plot (13.5 m$^2$) stem density samples of seedling/sapling (defined as trees 1 to 5 inches [2.54 to 12.7 cm] in diameter), and where the sub-plot sampling design was the national standard (Tree Table SUBP = [1,4]); LPJ-wsl v2.0 implicitly includes sapling and adult trees in estimates of tree height and stem density. We assumed that the filtered plots were representative of the true density and distribution of tree species for the general vicinity of the plots and of the USFS Division. Although these requirements for selecting FIA plots reduce the total amount of data, we aimed to make evaluations in a fair manner, in both spatial scale and meaning.

*Regional Simulations:* The objectives of the regional simulations (Table 2) were to evaluate demographic patterns of stand structure and function when simulating age classes using different age-width binning. Two idealized simulations were conducted at a regional scale to sample simulated annual stem density, average tree height, and NEP. The first simulation used the *unequalbin* age-width setup, $S_{unequalbin}$, and another used the *10-yr-equalbin* age-width setup,

$S_{10yrbin}$ (Table 2). For both simulations, Fire and LUCLM were not simulated. Instead, 5% of the fractional area of age classes > 25 years were cleared of biomass annually; the fractional area cleared was re-classified and merged with the youngest age class. The intent of the setup was to ensure that each grid cell maintained fractional area in every age class for each year of the simulation and avoided situations in which age classes were only present in 'bad years', or when growing conditions were poor. Both simulations were conducted with a 1000-yr 'spinup' using fixed $CO_2$ (287 ppm, 'pre-industrial' values) and climate randomly sampled from 1901-1920 to ensure that age distributions were developed and state variables were in dynamic equilibrium (i.e., no trend). A transient simulation then used time-varying $CO_2$ and climate, as prescribed by model inputs. Stand structure data were analyzed for 1980-2016.

The idealized simulations were performed for the mixed deciduous and evergreen forests of Michigan, Minnesota and Wisconsin, U.S.A (bounding box defined by left: 97.00º W; right: 82.50º W, top: 49.50º N, bottom: 42.00º W). These forests are of moderate temperate climates, with total annual rainfall 815.0 mm/yr (average over 1980-2016, based on CRU TS3.26) with monthly minimum 21.0 mm/mo and maximums of 148.5 mm/mo. Mean annual temperature (1980-2016, CRU TS3.26) was 5.98º C with monthly minimum of −11.45º C and maximum 20.98º C.

Data were pooled for the region over the time period and by age class. Data were plotted in box plots to show median value, interquartile range and outliers. No attempt was made to de-trend data because there was enough between age class variation to evaluate general demographic patterns visually.

### 2.3.3 Examining resilience: idealized simulation of a single event of deforestation, abandonment, regrowth

The objective of the idealized simulation was to evaluate the effect of age classes on relaxation times following a single deforestation, abandonment and regrowth event within a single grid cell (Table 2). The relaxation time is defined as the time required for a variable to recover to previous state and is a direct measure of ecosystem resilience (*sensu* Pimm 1984). Two simulations were conducted, the first simulation used the *10-yr-equalbin* age-width setup, $S_{age\_event}$, and another did not simulate age classes, $S_{noage\_event}$ (Table 2). Both simulations were conducted with a 1000-yr 'spinup' using fixed $CO_2$ (287 ppm, pre-industrial value) and climate randomly sampled from 1901-1920 to ensure that state variables were in dynamic equilibrium. A transient simulation then used time-varying $CO_2$ and climate, as prescribed by model inputs. Fire and LUCLM were not simulated. Instead, 25% of the fractional area was deforested in year 1910 of the simulation and classified as managed land. Treatment of deforestation byproducts (i.e., carbon in dead wood left on-site) were the same in both simulations. In the following year (1911), the managed land fraction was abandoned and allowed to regrow. The following state variables were plotted over time and visually evaluated: Net Biome Production (NBP, defined as NBP = NEP – LUC_flux), NEP, NPP, Rh, carbon in biomass.

The idealized simulations were performed for a single grid cell in a mixed broadleaf and evergreen needleleaf forest in British Columbia, CAN (121.25º W 57.25º N). The grid cell is a boreal climate with total annual rainfall 473.7 mm/yr (average over 1980-2016, based on CRU TS3.26) with monthly minimum 9.11 mm/mo and maximums of 105.8 mm/mo. Mean annual temperature (1980-2016, CRU TS3.26) was 0.59º C with monthly minimum of −16.9º C

and maximum 14.7º C.

### 2.3.4 Global simulation objectives and setup

There were three main objectives for global simulations. The first objective was to evaluate the contribution of age class information to global stocks and fluxes. Here, a simulation with age classes ($S_{age}$) was compared to a simulation without age class representation ($S_{noage}$) (Table 2). The second objective was to determine the relative influence of fire and LUCLM on the spatial and temporal distribution of ecosystem ages. For this objective, a Fire-only simulation ($S_{Fire}$) only had age classes created by fire, whereas a LUCLM-only simulation ($S_{LU}$) had age classes only created by abandonment of managed land or by wood harvest (Table 2). A simulation with both Fire and LUCLM ($S_{FireLU}$) was used as the baseline for comparison against $S_{Fire}$ and $S_{LU}$. The third and fourth objectives used data from $S_{age}$ ($S_{age} = S_{FireLU}$) to determine where the effect of demography was greatest and to identify the relative influence of demography versus climate on simulated fluxes (NEP, NPP, and Rh).

For all global simulations, a spinup simulation was run for 1000 years using randomly sampled climate conditions from 1901-1920 and atmospheric $CO_2$ fixed at pre-industrial levels (287 ppm) and no land use or wood harvest; spinup ensured that age distributions and state variables were in dynamic equilibrium (i.e., no trend). For simulations with land use, a second 'land-use-spinup' procedure was run for 398 years to initialize land use fractions of crop/pasture to year 1860, resampling climate and fixing $CO_2$ as in the first spinup. After spinup procedures, climate and $CO_2$ were allowed to vary until simulation year 2016; in $S_{LU}$ and $S_{FireLU}$, land-use change and wood harvest varied annually as prescribed by the LUHv2 dataset.

In the first objective (as above), global values for stocks and fluxes include both natural and managed lands. These global estimates conform to typical presentation of global values (Le Quéré et al. 2018), in Petagrams ($10^{15}$) of carbon. Comparisons are made among simulation types and to values from the literature.

For the second objective, a time series of zonal mean ecosystem ages were analyzed to determine the relative importance of $S_{Fire}$ and $S_{LU}$ on the observed distributions in $S_{FireLU}$. The first assessment was made by visual inspection of zonally-averaged time-series (i.e., Hovmöller plots) for the entire period of transient simulation, years 1860-2016. In addition, for each of $S_{Fire}$ and $S_{FireLU}$, a simple linear regression model (age = $\beta_0 + \beta_1$*year, setting 1860 as the reference year and defined as 1) was applied to identify trends in ecosystem age by the following zonal bands: boreal (50˚ N to 90˚ N), temperate (23˚ N to 50˚ N), and tropics (23˚ S to 23˚ N). Trends in age distributions due to LUCLM are not prescribed by inputs per se; instead, the age module is a necessary model structure that allows full realization of the effect of forcing data on age distributions. Trends in age distribution due to Fire, which is a simulated process as opposed to prescribed, result from climate and fuel load feedbacks on fire simulation.

### 2.3.5 Statistical model to assess relative importance of demography and climate

For the third objective of global simulations – to reduce dimensionality of the data and to assess the relative influence

of demography and climate on simulated fluxes – annual flux data from $\mathbf{S_{age}}$ (Table 2) were analyzed from 2000-2016 using generalized linear regression model,

$$flux_{i,yr} = B1_i \times \text{total\_precipitation}_{i,yr} + B2_i \times \text{mean\_temperature}_{i,yr}$$
$$+ B3_{i,ageclass} \times ageclass_{i,yr} \tag{6}$$

, where *flux* was one of {NEP, NPP, Rh} in kg C m$^{-2}$ yr$^{-1}$, precipitation (mm) and temperature (Celsius) data from CRU TS3.26, and *ageclass* was categorical, defined by the age-class code (Table 1), and the beta coefficients (*B*) for subscripts of grid cells (*i*), years (*yr*) and age class. The beta coefficients are therefore unique to every grid cell, and the betas for age classes are estimated separately for each age class within the grid cell ($B3_{i,age}$). An initial test of the model attempted to estimate globally-consistent predictor effects, but the model was found to be a poor fit (*not shown*) and it was assumed that there was too much variation among grid cells to detect globally-consistent effects. Instead of adding additional gridded fields of predictor variables to account for gridcell-level variation, the same statistical model was applied and analyzed per-gridcell. This allowed coefficients of precipitation, temperature and *ageclass* to vary by grid cell, in essence, reducing the effect of variation in PFT composition, soil texture and hydrology that might otherwise reduce predictive power.

In all grid cell analyses, the intercept term was intentionally omitted from the data model by adding a '-1' term to the data model. The *ageclass* term in the statistical model (*B3$_{i,age}$*), as a categorical variable, effectively takes the place of the intercept term anyhow, so the outcome is that estimates are for the absolute effect of each age class on the predicted flux as opposed to estimates that were relative to the first age class; this had no impact on estimated coefficients but it did simplify analyses. In grid cells where only a single age class was present, the statistical model was defined as (*flux$_{i,yr}$ = B1$_i$* total_precipitation$_{i,yr}$ + *B2$_i$* mean_temperature$_{i,yr}$ + *B3$_i$*), leaving the intercept term, in this case – *B3$_i$,* to be estimated from the data and then re-classifying the intercept term by the age-class code for the grid cell.

The degrees of freedom (d.f.) of a model for a grid cell with a single age-class was d.f.=14, based on 17 annual data points to estimate coefficients of three predictors. The degrees of freedom for a grid cell that had a maximum of 12 age-classes was d.f.=190, based on 204 annual data points to estimate coefficients for 14 predictors. Because the analysis produced statistical results for every grid cell, the degrees of freedom are not presented elsewhere. Coefficients were only analyzed or mapped when significant at p=0.05.

## 3 Results

### 3.1 Model Stand Structure – comparison against inventory data

FIA data were not equally available for every age class, nor for every Division (Figure SM2), but there were enough inventory data across 8 Divisions, spanning subtropical to temperate steppe climates, to qualitatively suggest that LPJ-wsl v2.0 does capture the expected patterns of tree density and height per age in the different climates evaluated. There was a tendency for LPJ-wsl v2.0 to overestimate stem density in younger age classes and systematically underestimate tree heights among age classes (e.g., Figure SM3, Figure SM5), for which the greater number of small individuals

could cause the average tree heights to be dampened. However, LPJ-wsl v2.0 is a big-leaf, single-canopy model that include space-filling 'packing' constraints on stem density, based on allometric rules for size and height of PFTs. Also the model does not represent multiple PFT cohorts in an age class, or more simply, it does not represent vertical

heterogeneity such as understory growth that would otherwise increase stem density. As such, and under the current model architecture and associated assumptions, the exact cause of the mis-match is unclear. Even still, the more general pattern of modeled stem density and tree height tended to track FIA data, with stem density being maximal in the younger age classes and declining thereafter, whereas tree height patterns increased more linearly before stabilizing (Figure SM6 to SM9).

FIA data had greater variability among age classes, regardless of Division. FIA data are not aggregated by PFT, instead they are species-level data. Changes in species composition over time do occur and can add to the observed variability among age classes in tree density and tree height. LPJ-wsl v2.0 includes a limited set of PFTs, which most likely limits the model's capacity to represent similar levels of variation in tree density and tree height. It is beyond the scope of this study to disentangle these patterns further, but greater agreement between observed and simulated patterns of

forest structure might be acheived by including additional plant functional types that are representative of tree species for a given Division.

**3.2 Model Age Dynamics**

**3.2.1 Dynamics of stand structure and function – regional simulations**

Forest structural characteristics of stem density, height, and NEP followed the expected patterns with age with a few

exceptions. In $S_{unequalbin}$ (Table 2), stem density increased from near zero to maximum in the 21-25 yr age class, before declining non-linearly (Figure 3). By contrast, the gradual increase in stem density in the first age class in $S_{10-yrbin}$ (Table 1) was not readily apparent because this process, which is evident in $S_{unequalbin}$, occurs entirely within the youngest 1-10 yr age class in $S_{10-yrbin}$. Both simulation setups approach the same stem densities after age ~25; prior differences are due to binning of age widths.

For average tree height in $S_{unequalbin}$, there were large tree heights in the youngest age class, which results from so-called 'survivor' trees (Figure 3). Not all trees are killed-off when a disturbance occurs in LPJ. Although the age class is 'reset' to the youngest age class, the survivor trees skew the height distribution until the density of establishing saplings subsequently increases and brings down the average tree height to smaller values. This pattern is more akin

to what occurs during natural fires or selective harvesting, which can reduce the overall age but might not result in a complete removal of all trees. By contrast, the skewed age-height pattern is not apparent in $S_{10-yrbin}$ (Figure 3) only because the same process is effectively hidden. Both simulation types approach the same average tree heights after age ~25 (Figure 3).

NEP peaked at age class 5-6 in $S_{unequalbin}$, before declining non-linearly to the lowest average value in the oldest age class (Figure 3). Although the unimodal peak was not apparent in $S_{10-yrbin}$, the maximum NEP occurred in the youngest

age class and also declined non-linearly thereafter (Figure 3). The decline in NEP after a maximum at 5-6 years was driven mainly by an increase in Rh due to increases in turnover rather than a larger decline in NPP (Figure 4). The peak in NEP did not coincide with maximum tree density at ~20 years. Instead, model dynamics suggest that the total foliar projective cover of tree canopies reaches near maximum (80-95% cover, *not shown*) at 5-6 years, thereafter plant competition reduces NPP while biomass turnover increases, which together cause the apparent decline in NEP. The time period of canopy closure, at 5-6 years, in LPJ-wsl v2.0 is probably too early, in part due to advanced regeneration (saplings establish at 1.5 m height) and constant establishment rates. The age-class module qualitatively demonstrates NEP-age relationships consistent with field-based evidence (Ryan et al. 2004, Turner 2010).

Lastly, an emergent pattern was found in the declining portion of the NEP-age curve and approximately follows the functional form $NEPmax*0.70^{age-agemax}$, where *NEPmax* is the maximum NEP flux at the initial point of decline, *age* is the age of the patch, and *agemax* is the age of the patch where NEP is maximized. Thus, the non-linear decline in NEP is approximately 30% with increasing age. The functional equation holds between year 5-6 to year 25, after which NEP decreases only by 20% with increasing age and the functional form becomes $NEP_{25yr}*0.80^{age-25}$, where $NEP_{25yr}$ is the NEP at year 25. The functional form of the decline in NEP is consistent among climate regions when simulated data is analyzed separately for all U.S. States (*not shown*). **The binning strategy is likely not a determinant of this pattern between NEP and stand age, which is evident in Figure 3 for both age-class setups. In this regard, we care less about the binning strategy and more that the emergent pattern is reflective of simulated model dynamics. This emergent pattern could lend itself to observational constraints if similar emergent patterns can be derived from forest inventory data in the future.**

### 3.2.2 Time-series evolution of a deforestation, abandonment and regrow event

A single event of deforestation, abandonment and subsequent forest regrowth caused long-lasting effects on carbon balance and dynamics when omitting age-class dynamics. In the simulation without age classes, $S_{noage\_event}$ (Table 2), gridcell-level NEP takes ~30 years to recover to values prior the event, whereas the age-class simulation, $S_{age\_event}$, takes only 5-6 years to recover (Figure 5) – a 5-fold change in relaxation times. The quick recovery of gridcell-level NEP in $S_{age\_event}$ is due partly to the fact that the fraction of the grid cell (75%) that was *not* deforested maintained its state variables (carbon stocks in vegetation, soil, litter) unchanged from its prior state, which buffered NEP and dampened the effect of the smaller fraction (25% of grid cell) that was deforested. Age-class dynamics also contributed an elevated NEP (Figure 5) that quickens the recovery at the grid cell level. In $S_{age\_event}$, there is an elevated NEP in the secondary stand that is sustained for more than 30 years following the event.

In $S_{noage\_event}$, vegetation dynamics cause turnover to increase and causes an elevated gridcell-level Rh that is consistently higher than gridcell-level NPP for 30 years after the event. This pattern is striking because NPP recovers quicker than in $S_{age\_event}$ and maintains an elevated value for ~30 years. Following a disturbance event in LPJ, stem density and foliar projective cover is reduced but the state variables (carbon in plant organ pools of leaf, stem, root) maintain prior values; this is the reason gridcell-level NPP recovers quickly in the standard-no-age simulation. As stand density

increases again, canopy closure initiates competitive dynamics that result in mortality of individuals of the plant population that are generally larger than if the stand had progressed from small to large individuals (as in $S_{age\_event}$).

The VTFT module also uses the mean-individual approximation but stand dynamics are always allowed to occur in natural progression and the relatively small age widths (10-years) ensure that stand age dynamics (NEP-age trajectories in Figures 3 and 4) most evident in the first 50 years are discretely modeled. To reiterate, we think that the simulated flux dynamics in the no-age simulation is a pure model artefact. What we mean by that is that a patch-based (age

class) model is more like reality, where the full 'grid' of space is an explicit representation of unique patches of ecosystem. Whether or not the recovery times themselves are accurate (30-years vs 5-years) is less concerning at this point. The growth rates and recovery trajectories will likely have to be optimized, ideally, to observed patterns, but this is beyond the scope of this paper.

### 3.3 Global Stocks, Fluxes, and Age Distribution

**3.3.1 Stocks and fluxes – $S_{noage}$ versus $S_{FireLU}$ and convergence in global NEP.**

Carbon stocks in biomass are lower in $S_{age}$ than in $S_{noage}$ by ~40 Pg C globally (Figure 6). Lower global biomass in $S_{age}$ can be explained by feedbacks from LUC and Fire that create younger age classes that have lower overall biomass than in older stands. In addition, age dynamics cause turnover to increase (as in Figures 3 and 4), causing soil carbon to be greater by ~35 Pg C and litter carbon to be greater by 5 Pg C. Taken together, age-class dynamics cause 40 Pg

C to be re-allocated from the living biomass pool to the soil-detrital pool, which compounds to alter the magnitude of fluxes from heterotrophic respiration. Demographic changes in turnover, such as these, are already known to be a large source of uncertainty among projections by global ecosystem models (Friend et al. 2014). What these numbers emphasize, however, is that uncertainty among models could be reduced by explicitly modeling age dynamics.

Net Ecosystem Exchange (NEE; positive fluxes to atmosphere) is only marginally different between $S_{noage}$ and $S_{age}$ simulations (mean difference of 0.25 Pg C yr$^{-1}$ over 2000-2010). Compensatory fluxes in Fire and Rh explain the small difference in NEE at global scales. Fire fluxes in $S_{age}$ are lower by 0.92 Pg C yr$^{-1}$ in the 2000s than in the $S_{noage}$, but fluxes from Rh are greater in $S_{age}$ by 1.61 Pg C yr$^{-1}$ and NPP also greater by 0.55 Pg C yr$^{-1}$. The fluxes in Fire, Rh and NPP largely offset to minimize differences in NEE from age dynamics.

The question still remains – should there be an expectation for greater differences in NEE? Consider that deforestation (areal changes prescribed the same in $S_{noage}$ and $S_{age}$) occurs from the oldest to youngest age class in $S_{age}$, following greater to lower overall biomass, respectively. The deforestation flux is greater in the $S_{age}$ by only 0.04 Pg C yr$^{-1}$ in 2000s compared to deforestation fluxes in $S_{noage}$, which makes sense given that low-biomass age classes are not pref-

540 erentially deforested or harvested. By contrast, fire is not prescribed in LPJ-wsl v2.0 but it is simulated based on soil moisture and a minimum fuel load. It is not clear outright how age-dynamics affect soil moisture, but fluxes from fire would need to be proportional to the biomass in an age class. By definition in $S_{age}$, there is explicit representation of lower-biomass age classes (i.e., younger) than in $S_{noage}$, and a series of fires or disturbances within the grid cell would

drive the age distribution towards younger states, exacerbating differences in downstream fluxes as well. That global NEE only changed marginally when simulating global age dynamics was a surprise, but explained by shifts in the carbon pools and compensatory fluxes, then the patterns appear to make sense. In light of these compensation effects, however, there is a great need to benchmark fluxes from critical feedbacks, particularly from fire in this case. It is beyond the scope of this paper to do so, and best available datasets, such as the Global Fire Emission Database (GFEDv4s; van der Werf et al. 2017) do not lend themselves to direct comparison with fire fluxes from LPJ. GFED includes fires from deforestation and land management that are tracked differently in LPJ-wsl v2.0 – as a land-use change flux, which cannot simply be added to the fire flux for direct comparison to GFED without double counting. In any manner, this issue is stated as a suggestion for future development and refinement.

**3.3.2 Global age-class distribution – contribution of fire and LUCLM to age distributions**

Average ecosystem age, generated by the model, differed greatly among continents (Figure 7), with large areas of old-growth forests in Asia, Europe, North and South America skewing the distribution towards older ages. The largest area of young ecosystems was located in Africa and Australia (Figure 1), wherein age classes comprised an ~1:1 age to fractional area ratio of vegetated land (age-classes < 20 years comprise ~20% of the vegetated land area in Africa and Australia and age-classes < 40 years ~ 40% of vegetated land area; Figure 7).

Ecosystem age by zonal band was oldest at boreal latitudes, followed by temperature latitudes, and youngest in tropical latitudes, which was primarily the results of frequent fires in simulated grassland ecosystems. The primary driver of zonal age distributions was Fire (Figure 8). According to results from the statistical model (Table 3), the average age difference due to fire among zonal bands in 1860 was 23 years between Boreal (older) and Temperature (younger) latitudes, and it was 32 years between temperature (older) and tropical (younger) latitudes. The difference in ecosystem age among zonal bands increased to 60 years in simulation year 2016 between boreal and temperate latitudes, while the difference in ages between temperature and tropical latitudes remained similar (31 yr age difference). There was a statistically significant decrease in zonal ecosystem age over time due to fire (Table 3), most likely from feedbacks due to enhanced fuel (biomass) production from $CO_2$ fertilization. The causes were not explored further because feedbacks between fire-climate-$CO_2$ are largely constrained by the fire module itself. The emphasis here is simply that fire was a major driver of age distributions and fire-age relationships had an apparent trend over time. Between simulation years 1860 and 2016, fire caused a total change in ecosystem age, integrated over the time period, by -1.5 years in boreal zones (negative values for a decrease in age), whereas the change was greater in temperate (-6.7 years) and tropical (-8.24 years) zonal bands (Table 3). The larger trend in temperate and tropical latitudes might be due to increasing warming temperatures in contemporary times, causing drier conditions more suitable for fire, or from increases in fuel loads from $CO_2$ fertilization. A more convincing argument would require support from additional factorial experiments to identify to the casual driver of the trend differences.

After accounting for the effects of fire, LUCLM caused a much greater change over time in the zonal ecosystem age (Figure 9). Integrating from 1860 to 2016, LUCLM caused a zonal change in ecosystem age by -6.1 years in boreal

zones, whereas the change in ecosystem age from LUCLM in temperate and tropical zones was -21.6 years, with no significant difference in the trend due to LUCLM among these zonal bands (Table 3). These patterns are consistent with the concentration of deforestation in the tropics and land-use change in temperate latitudes, as described by the forcing data (Hurtt et al. 2011, Hurtt et al. 2020).

**3.4 Global Demographic Effects on NPP and Rh**

**3.4.1 Simplification of LPJ-wsl v2.0 via a statistical model**

The statistical model (flux = *B1* precipitation + *B2* temperature + *B3_{age}* age-class; *See Sect. 2.3.5* for details) was able to estimate simulated NPP and Rh with great predictive power, with $R^2$ values between 0.95-0.98 (Figure 10). The predicted fluxes were at annual time scales, with annual variation being mainly driven by total annual precipitation and mean annual temperature, whereas the mean state (intercept) being predicted by the age class. The predictive

power for a model of NEP was worse ($R^2$ between 0.60-0.65; SM Figure 1). The effect of precipitation, temperature and age-class on NEP was not consistent enough for robust predictions, but more specifically, the predictors had different effects on NPP versus Rh leading to poorer model fit. As it is, NEP is better derived as predictions of NPP minus predictions of Rh rather than having a standalone model for NEP.

**3.4.2** *The Effective Range of Predictors – assessing relative importance of demography on predicted fluxes*

The "Effective Range of the Predictors" were mapped to visualize spatial patterns of the range of effects, given observed values for the predictors (Figure 11). In essence, the effective range of the predictor is a measure of the dynamic range in the predicted flux due to changes in precipitation, temperature or demography. It is calculated as the gridcell-

specific beta coefficient multiplied by the observed range of the predictor for a given grid cell, which helps constrain the effect of the predictor on the predicted flux to realistic values. For example, for the LPJ-wsl v2.0 grid cell at location [110.75 W 50.25 N], the ß estimate for the effect of precipitation on NPP was 0.0028, and the range of observed precipitation (based on CRU TS36) was 282 mm, then the effective range of the predictor on the flux was calculated as 0.0028*282 = 0.79 kg C m$^{-2}$ yr$^{-1}$.


The effect of precipitation on NPP was clearly greater in the central USA, central South America, and Eastern Australia (range of effect ~ 0.70 kg C m$^{-2}$ yr$^{-1}$ due to precipitation) than in other locations, and overall, precipitation had a stronger (positive) effect on NPP than on Rh (Figure 11). It was also clear from the maps that the direction of the effect of temperature on NPP was more spatially varied in the direction of effect (both positive and negative) than

other predictors (Figure 11). The effects of precipitation and temperature displayed similar spatial patterns in both primary and secondary stands, which was a good indicator that the model was performing as expected because, within the LPJ-wsl v2.0 model, the distinction between primary and secondary stands is mainly to track land use histories and there was no reason, *a priori*, that climate effects should differ substantially between the two stand types.

The effective range of demography on fluxes was generally lower than the effective range of precipitation and

temperature, but there were regions where the range of demographic effects were just as important as, or greater than, the climate predictors. The demographic effect on NPP ranged between 0.30-0.60 kg C m$^{-2}$ yr$^{-1}$ in Eastern North America, Western Europe, Central Africa, Eastern China, Tropical Asia, and distributed smaller areas of South America (Figure 11), whereas it was at maximum ~0.10 kg C m$^{-2}$ yr$^{-1}$ in other regions. The higher demographic effect was predominately on secondary stands (Figure 12), but there was also a distinct absence of primary stands in these same areas (Figure 11) so it could not be said definitively if the higher demographic effect was due to a wider age distribution, and therefore a greater demographic effect, or simply due to the productivity of these locations.

**3.4.3 Frequency distribution of demographic effects**

The global mean demographic effect on NPP on primary stands was 0.078 ± 0.063 [0, 1.37] kg C m$^{-2}$ yr$^{-1}$ (μ ± stdev. [min, max]), whereas on secondary stands it was 0.160 ± 0.141 [0, 1.33] kg C m$^{-2}$ yr$^{-1}$. There were differences in the spatial distribution of primary and secondary stands that led to the disparity in global mean values of the demographic effect. On primary stands, the distribution of age classes with maximum NPP flux was skewed towards the second (11-20 years) age class having the maximum NPP flux, whereas on secondary stands, the maximum NPP flux was in the first (1-10 years) and also in the second age class (Figure 12). The first class was categorized as 1-10 years, but in the presence of constant renewal, an age class can effectively be younger than an equivalent age class without such recurrent disturbance. Furthermore, on primary stands, fire is the only mechanism that creates young age classes, whereas land management also creates young age classes on secondary stands. It is possible for wood harvest, a form of simulated land management, to result in advanced regeneration of younger stands if harvest demand is met without 'clear-cutting' the prescribed fractional area under harvest. Currently, the model structure does not lend itself to say definitively the cause of the difference in the age class of maximum flux, but the only process that differs between primary and secondary stands is land management, so it is reasonable to assume that land management is the cause of the difference. In any manner, global values for age-effects for NPP on primary and secondary stands were also skewed towards greater values on secondary stands, but more due to the absence of primary stands in productive areas where secondary stands dominated (e.g., Eastern U.S.A.).

Following a similar pattern, the demographic effects on Rh were greater on secondary stands than on primary stands (Figures 11 and 12), which could be partly explained by the differential coverage of secondary and primary stands, but also by historical land use. LUCLM leads to overall greater inputs to soil and litter carbon pools than does fire, and the latter is simulated in the same manner on secondary stands as on primary stands. In LPJ, wood harvest is only 60% efficient, leaving dead biomass 'residue' as a legacy flux. An increase of carbon in the litter and soil pools would add additional mass that can be respired during heterotrophic respiration, and which manifests as a larger demographic effect on Rh, ranging from 0.25 to 0.70 kg C m$^{-2}$ yr$^{-1}$ on the high-end (Figure 12).

**4 Discussion**

**4.1 Distribution of Ecosystem Age on Earth**

The LPJ-wsl v2.0 age-module simulates the age-class distributions on Earth resulting from fire, land use change, and

wood harvest (Figure 13), while also simulating important demographics effects on NPP and Rh. Simulations demonstrate that fire and LUCLM have been driving the latitudinal age distribution towards younger states in contemporary times (Figure 8), suggesting an increasing role of age dynamics on global ecosystem functioning. Whereas time is the only mechanism that increases ecosystem age, any additional disturbance not explicitly modeled in this study will

decrease age.

The simulations omit widespread disturbances of windstorms, flood, pest and disease outbreak, selective logging, and other processes that would modify stand structure and function. For instance, small-scale logging activity is a dominant disturbance in South Eastern U.S.A. (Williams et al. 2016) but it is underestimated by the LUCLM driver data in this

study ('LUHv2', Hurtt et al. 2020); otherwise the simulated age of secondary forests in this region (~100 years) would be lower and closer to inventory-based age estimates of these forests (< 50 years; *Figure 4 in* Pan et al. 2011b). In some geographic locations, it is certainly possible that our wood harvest priority rules (defined by harvesting oldest age class first) might lead to simulated stand ages that are younger than observed stand ages if other harvest rules were applied in real life. For example, if there are a mandates to preserve old-growth forests, then logging might preferen-

tially occur on young or mid-aged forests, leaving older age class forests unharvested. We evaluated the age distribution by continent simulated by LPJ-wsl v2.0 to the Global Forest Age Database (GFAD v1.0, Poulter et al. 2018), which is derived from country-level inventory data. The comparison shows that the simulated ages are consistently older than the GFAD dataset (SM Figure 11). Furthermore, the fire module has been well evaluated at global scale (Thonicke et al. 2001) but it needs improvement because it is overly simplistic and underestimates global burned area

(Hantson et al. 2020). It is more likely that effects of fire are much greater than simulated in this study. This study likely underestimates disturbances rather than overestimates them, and as such, these simulations overestimate ecosystem age. But again, additional disturbances would only lead to younger age classes, enhancing the role of age dynamics in regional and global carbon cycles.

Our model developments are not optimized to match observations, although we are working toward this end. Future goals are to assimilate stand-age related data, such as remotely-sensed canopy data and stand index growth curves, to align model processes with observations. Even with these caveats in mind, the findings presented retain utility as insight into the way age-class dynamics integrate into our broader understanding of global carbon dynamics. Ecosystem demographics likely play a larger role than suggested here, and on regional scales, demographic effects on NPP

and Rh are already identified by this study as more important in East Asia, Tropical Asia, Europe, Central Africa, Eastern North America, and Tropical South America than they are in other regions, where average ecosystem ages are much older.

**4.2 Age Dynamics Increase Turnover**

In an analysis by Friend et al. (2014), it was determined that demographic processes (age-dependent mortality and

turnover) influence carbon residence time (1/Turnover), which was found to be a major source of uncertainty in future projections by global ecosystem models. In this study, it was demonstrated that simulation of age classes led to a ~40

Pg C shift from live vegetation to the soil-litter pool, effectively an increase in biomass turnover. That turnover increases when explicitly simulating ageclasses is a natural expectation, but the magnitude of the simulated turnover between carbon pools less certain until detailed benchmarking is conducted. Further, relaxation times, or the time to

return to a previous state, were up to 30 years in the no-age simulation ($\mathbf{S_{noage\_event}}$; Figure 5) but relaxation times were less than 10 years when simulating age classes, suggesting that uncertainty in carbon residence time could potentially be reduced by improving representation of demographics in models. Omitting age class representation in models can leave long-lasting patterns in simulated fluxes that could inflate land-use change fluxes at global scales when considering legacy fluxes from past land-use change (Pongratz et al. 2014). The current state of knowledge is that fluxes

from gross land-use change and land management cause greater-than-expected land use fluxes (Arneth et al. 2017), but existing models that estimate the global land use flux (Arneth et al. 2017, Le Quéré et al. 2018) do not include age dynamics. If resiliency is inversely proportional to relaxation times (a quicker return to previous states is represented by shorter relaxation times, therefore greater resiliency; Pimm 1984, Tilman and Downing 1994), then instead of land-use change fluxes being 'greater than assumed' (Arneth et al. 2017), we might rethink the land as being 'more resilient

than expected' when demographic effects are considered at large scales.

### 4.3 Forecasting Demographic Effects with a Simplified Statistical Model

The modeling community has made increasing effort to simplify complex models using a traceability framework (Friedlingstein et al. 2006, Xia et al. 2013). Statistical emulators, from matrix models (Huang et al. 2018) to accounting-type statistical models, which track individual carbon pools (Xia et al. 2013, Ahlström et al. 2015), have been

developed to reduce the dimensionality of simulated state variables. However, statistical modeling by linear regression can be a more straightforward approach, as long as the statistical model shows promise.

We found that LPJ-wsl v2.0 fluxes of NPP and Rh could be predicted at annual timescales by three terms, precipitation, temperature and age-class. Part of the success of the data model came from allowing coefficients to vary by grid cell.

This allowed the intercept (age-class) term to effectively capture grid cell level variation in soil texture (which influences soil hydrology and plant available water), PFT composition and cloud cover. Another insight was that climate and age-class had differential effects on NPP versus Rh, which makes sense and ultimately led to poorer fit of the NEP model (NEP = NPP – Rh). It might have been possible to improve upon the NPP model further by separately modeling GPP and Autotrophic Respiration (NPP = GPP – Ra) because climate might also have differential effects

on GPP than on Ra, but suffice to say that the NPP statistical model was robust.

Although unexplored in this study, the spatial datasets of predictor coefficients could be used within an emulator (Xia et al. 2013, Ahlström et al. 2015) to forecast NPP and Rh, while exploring the effects of extreme climate scenarios (Reichstein et al. 2013) and changes in ecosystem demography from land-use change and land management. Such

application would allow for a much quicker exploration of scenarios and could include a more explicit treatment of uncertainty that would otherwise be too costly for the simulation model in terms of computing time. With regards to climate, the spatial dataset of precipitation coefficients has an equivalent meaning to spatial maps of climatic

sensitivity. In fact, the maps of the effective range of precipitation on NPP (Figure 11) show areas where the precipitation effect is largest, notably in semi-arid biomes – a biome that is known to be highly sensitive to precipitation and has been shown to play an important role in the inter-annual variability of global-scale fluxes (Poulter et al. 2014, Ahlström et al. 2015). But what if, in a given year, semi-arid biomes received their maximum annual precipitation, while every other biome received its lowest annual precipitation – can anomalously high annual precipitation and high productivity events in some regions overcome anomalously low precipitation and low productivity events in other regions? Are the effects of different climate scenarios dependent on demography? These types of question are best suited for exploration within a simplified statistical model that maintains fidelity to the process-based model because effects of climate on fluxes can be explored quicker, easier, and with a better treatment of statistical uncertainty.

A last note on emulators. Useful statistical emulators have fidelity to the underlying process model, but such emulators often cannot address uncertainty from parameter values that are often fixed in the underlying process model or uncertainty in process representation. In an ideal world, the statistical parameters for climate sensitivity and stand age, for instance, would be constrained by uncertainty simulations that are themselves bounded to a realistic range of parameter values in the process model (Zaehle et al. 2005) and alternate representations of ecosystem processes (Forkel et al. 2016).

**4.2 Vector Tracking of Fractional Transitions (VTFT) – modeling age classes in global models**

The VTFT approach simulated classic demographic responses in NPP and Rh (Figure 4), a differential in younger age classes that led to a larger carbon sink in the youngest stands. These demographic responses are inherent within the original formulation in LPJ; that is, establishment rates and the process of self-thinning of stand density over time as plants grow and compete (for space, light, water resources) have been unchanged. In the original formulation of LPJ-wsl v2.0 (prior to this study), and under a hypothetical scenario where a disturbance clears the biomass from the entire grid cell ($0.5° \sim 2,500$ km$^2$), the resultant evolution of stand structure and fluxes would produce the same pattern as in the age-module, such as the age-NPP pattern from Figure 4. It is often the case, however, that smaller disturbances ($<< 2,500$ km$^2$) occur regularly as opposed to a much larger disturbance the size of the entire grid cell. As such, in the original formulation of LPJ, the potential benefits of demographic responses are often masked (as demonstrated in *Section 3.2.2*; Figure 5). One can then say that the VTFT age-module reveals intrinsic demographic responses and model behavior that would rarely emerge otherwise.

Total runtime for global age-class simulations ($S_{age}$) was ~8 hrs on 32 Intel Xeon CPUs, including spinup to transient simulations, whereas the total runtime for the no-age simulations ($S_{noage}$) was ~3 hrs. On a limited sample of single grid cell simulations, there was a 4- to 6-fold increase in runtimes, but not all grid cells require simultaneous tracking of every age-class so the increase in runtime of global simulations was lower than expected from per-gridcell estimates.

**4.3 Opportunities for Improving Modelled Age-dynamics**

In order of priority for improvement of the age-module: 1) improve age class growth rates to align with observations, 2) improve representation of disturbances, 3) improve representation of early- and late-successional plant species and add vertical structural complexity such as understory/overstory canopy. Below, we provide suggestions and examples from the literature as how these improvements might be accomplished.

Inventory data or remotely-sensed observations of canopy height provide a potential means for constructing age-height curves (Croft et al. 2014, Yue et al. 2016) to inform growth rates by age class. Alternately, Hiltner et al. (2020) recently optimized mortality rates in an individual-based model at different forest successional stages by using satellite-derived proxies of tree mortality (Hiltner et al. 2020); their optimized model was shown to improve representation of forest states during post-disturbance regrowth. Another LPJ variant, the LPJmL4 DGVM, also underwent parameter optimization to improve spatial patterns of tree cover and forest turnover (Forkel et al. 2019). Different solutions are possible, and not all of them require parameter optimization, but the aim should be to align simulated forest structure and function with observations.

Our comparison of simulated versus inventory forest age distributions by continent (SM Figure 11) clearly show that LPJ-wsl v2.0 overestimates stand age. A potential solution to this discrepancy is to incorporate additional disturbances within the model to help simulate age distributions more consistent with inventory (Pan et al. 2011a) and satellite (Pugh et al. 2019b) data and contribute to more scientifically relevant questions. Firstly, general improvement of the fire module in LPJ-wsl v2.0 is necessary to match burned area observations. One such solution is to use a suite of satellite data products to prescribe burned area instead of simulating fire, which could help improve simulated stand age distributions in areas where fire is observed but not well simulated mechanistically (Poulter et al. 2015). Modeled disturbances need not be complex to explore their effects on age distributions, they only need to reset a fractional area to the youngest age class. For example, windstorms from Hurricanes are known to be a large disturbance of Eastern North American forests (Dale et al. 2001). Data on Hurricane return intervals and locations of landfall in Eastern North America have been available for some time (Keim et al. 2007), and could be used to prescribe a periodic resetting of age classes to assess the demographic effect of Hurricanes on ecosystem function. In another example, forest gaps represent areas of high production because of high resource abundance relative to the surrounding areas. The distribution of forest gaps also has a predictable power-law relationship with size of the gap (Asner et al. 2013), which can be allowed to vary across and within regions (Asner et al. 2013, Espírito-Santo et al. 2014), and this fact lends itself well for representing gaps within the framework of the current age-module. Many disturbances can be prescribed based on observed forest disturbance rates (Pugh et al. 2019), but prescribed disturbance patterns typically sacrifice capacity to simulate under novel conditions so there are tradeoffs to consider.

There are limitations to the current framework of the model, which are more difficult to overcome and will require more effort in model development. In this version of the model, plant composition and competitive dynamics in young age classes are not representative of early successional dynamics because there is a lack of plant trait variation in the current set of PFTs that could otherwise represent a wider range of growth strategies, turnover, and production (Pütz

et al. 2011, Fischer et al. 2016, Miller et al. 2016). There is also no height variation within an age class, for lack of a radiative transfer model; each age class in LPJ-wsl v2.0 is an even-height stand. Demographic patterns in this study (age-NPP, age-Rh, relaxation times by age class) will inevitably differ when, and if, additional trait and height variation is incorporated into the model. Recent model developments in JSBACH4 (Nabel et al. 2019) and ED-2.2 (Longo et al. 2019) could point the way forward for incorporating a greater amount of vertical heterogeneity in LPJ-wsl v2.0,

as well as in other models.

## 5. Code and Data Availability

LPJ-wsl v2.0 model code, in its entirety, is freely available at <https://github.com/benpoulter/LPJ-wsl_v2.0>, and a permanent version of the model code is deposited at Zenodo <DOI: 10.5281/zenodo.4409331>. Code used for analyses and figure production are available at <https://github.com/lcalle/VTFT_demography>. Associated data necessary to reproduce the analyses and figures, as well as a copy of the analysis code is permanently archived at the

Dryad Digital Repository <https://doi.org/10.5061/dryad.k6djh9w4x>.

## 6. Author Contributions

LC and BP designed the model experiments and LC carried them out. LC developed the code for the age-class module and performed the simulations. LC and BP prepared the manuscript.

## 7. Acknowledgements

Computational efforts were performed on the Hyalite High Performance Computing System, operated and supported by University Information Technology Research Cyberinfrastructure at Montana State University. LC was supported by a National Aeronautics and Space Administration Earth and Space Science Fellowship (NASA ESSF 2016-2019, Grant# NNX16AP86H). This work contributed to the 3DSI Project (NASA proposal #16-CARBON16-0124). Thanks

to Paul Montesano, Bryce Currey, and Tom Pugh for comments on the manuscript.

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

**Table 1. Age class widths corresponding to two different simulation age class setups in LPJ-wsl v2.0. The age-class codes are referenced in Figures.**

| | *Age Widths (years)* | |
|---|---|---|
| **Code** | **Unequal Bins** | **10-yr Equal Bins** |
| 1 | 1-2 | 1-10 |
| 2 | 3-4 | 11-20 |
| 3 | 5-6 | 21-30 |
| 4 | 7-8 | 31-40 |
| 5 | 9-10 | 41-50 |
| 6 | 11-15 | 51-60 |
| 7 | 16-20 | 61-70 |
| 8 | 21-25 | 71-80 |
| 9 | 26-50 | 81-90 |
| 10 | 51-75 | 91-100 |
| 11 | 76-100 | 101-150 |
| 12 | +101 | +151 |


**Table 2. Description of LPJ-wsl v2.0 simulations in this study, corresponding objectives and related science questions. Land-Use Change and Land Management (LUCLM, LU).**

| Simulation | Description | Objective and Questions | Structure/Processes Included | | |
| --- | --- | --- | --- | --- | --- |
| | | | Age classes | Fire | LU-CLM |
| *Single-cell* | | | | | |
| S$_{age\_event}$ | Idealized simulations of a deforest, abandon, and regrow event in British Columbia, CAN [121.25W 57.25N] | Evaluate recovery dynamics of a single regrow event. Do age dynamics influence relaxation times? | ✓ | ✓ | ✓ |
| S$_{noage\_event}$ | | | x | ✓ | ✓ |
| *Regional* | | | | | |
| S$_{unequalbin}$* | Idealized simulation with 5% of grid cell cleared annually to create a wide age-class distribution in mixed broadleaf and evergreen temperate forests of Michigan (MI), Minnesota, and Wisconsin (WI) of U.S.A. | Does the model capture 'classic' demographic patterns in stand structure (tree density and height) and function (NEP, NPP, Rh)? | ✓* | x | x |
| S$_{10yrbin}$‡ | | | ✓‡ | x | x |
| *Global* | | | | | |
| S$_{noage}$ | Standard-forcing factorial simulations at global scale. | Do age dynamics influence global stocks and fluxes? | x | ✓ | ✓ |
| S$_{Fire}$ | | What is the relative contribution of Fire and LU to ecosystem age? | ✓ | ✓ | x |
| S$_{LU}$ | | Are demographic effects evident in fluxes, and where is the effect greatest? | ✓ | x | ✓ |
| S$_{FireLU}$ (S$_{age}$) | | What is the relative contribution of climate versus demography on fluxes? | ✓ | ✓ | ✓ |

* unequal age width simulation. Age widths as described in Table 1

‡ 10-yr interval age width simulation. Age widths as described in Table 1

**Table 3. Linear trend statistics by zonal band from LPJ-wsl v2.0 simulations, based on model (age = $\beta0$ + $\beta1$*year) where year at 1860 is indexed at 1. Coefficients listed as $\mu$ ± S.E. All d.f. are 113 and p < 0.001.**

| Zonal Band | Simulation | $\beta0$ | $\beta1$ | $R^2$ |
|---|---|---|---|---|
| Boreal | Fire Only ($S_{Fire}$) | 141.7 ± 0.01 | -0.0098 ± 0.0002 | 0.95 |
| | Fire and LUCLM ($S_{FireLU}$) | 139.7 ± 0.13 | -0.0388 ± 0.0019 | 0.78 |
| Temperate | Fire Only ($S_{Fire}$) | 118.5 ± 0.05 | -0.0525 ± 0.0008 | 0.98 |
| | Fire and LUCLM ($S_{FireLU}$) | 112.6 ± 0.21 | -0.1383 ± 0.0032 | 0.94 |
| Tropics | Fire Only ($S_{Fire}$) | 95.9 ± 0.06 | -0.0429 ± 0.0009 | 0.95 |
| | Fire and LUCLM ($S_{FireLU}$) | 88.9 ± 0.16 | -0.1382 ± 0.0024 | 0.97 |


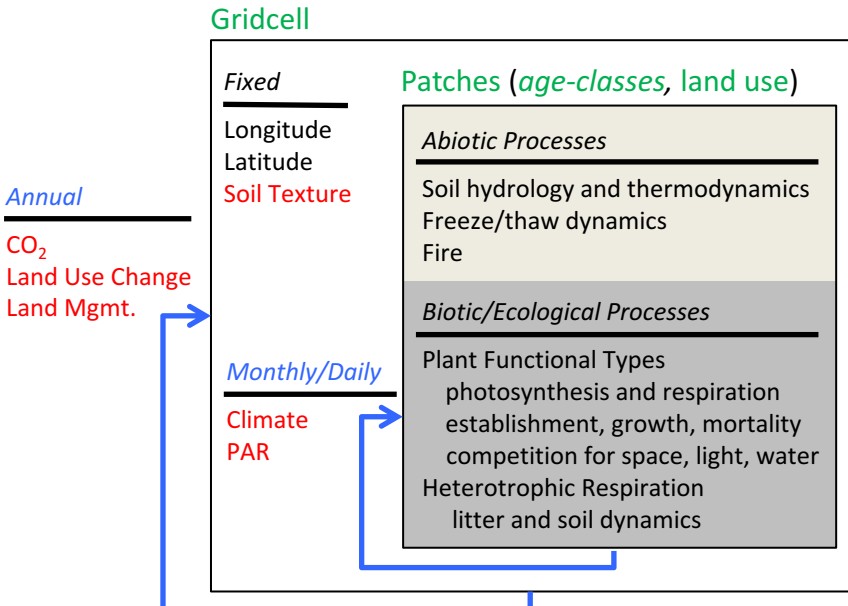

**Figure 1. LPJ-wsl v2.0 model structure of inputs (red), time-steps (blue) and the level at which state variables are tracked within grid cells and sub-gridcell age classes (green), such as age classes or land uses. Simulation of abiotic, biotic and ecological processes occurs at the scale of an age class.**


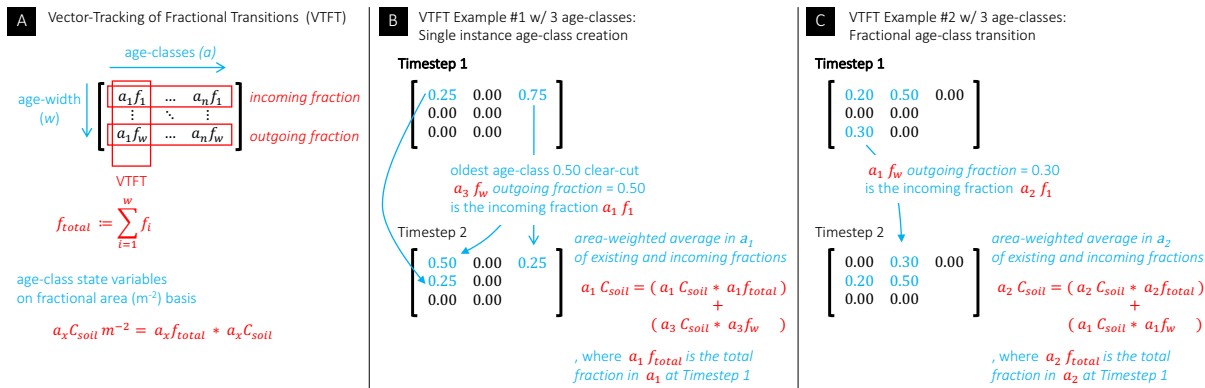

**Figure 2. Methodological examples of the matrix based method called Vector-Tracking of Fractional Transitions for computationally-efficient simulation of age classes in large-scale models. (a) Hypothetical matrix of VTFT vectors of fractional areas (*f*). The total area of the age class is the sum of the fractional areas in the corresponding VTFT vector. State variables are calculated on area basis by accounting for the fractional area of the age class, in this example $C_{soil}$ is the carbon in soil. (b) An example of the VTFT method for a newly created age class by clear-cut wood harvest. An area-weighted average updates age-class state variables in the youngest age class using the preceding total fractional area of the age class and the incoming fraction. (c) A VTFT example for a fractional age-class transition. An area-weighted average updates state variables in an age class using the preceding total fractional area of the age class and the incoming fraction from the younger age class.**

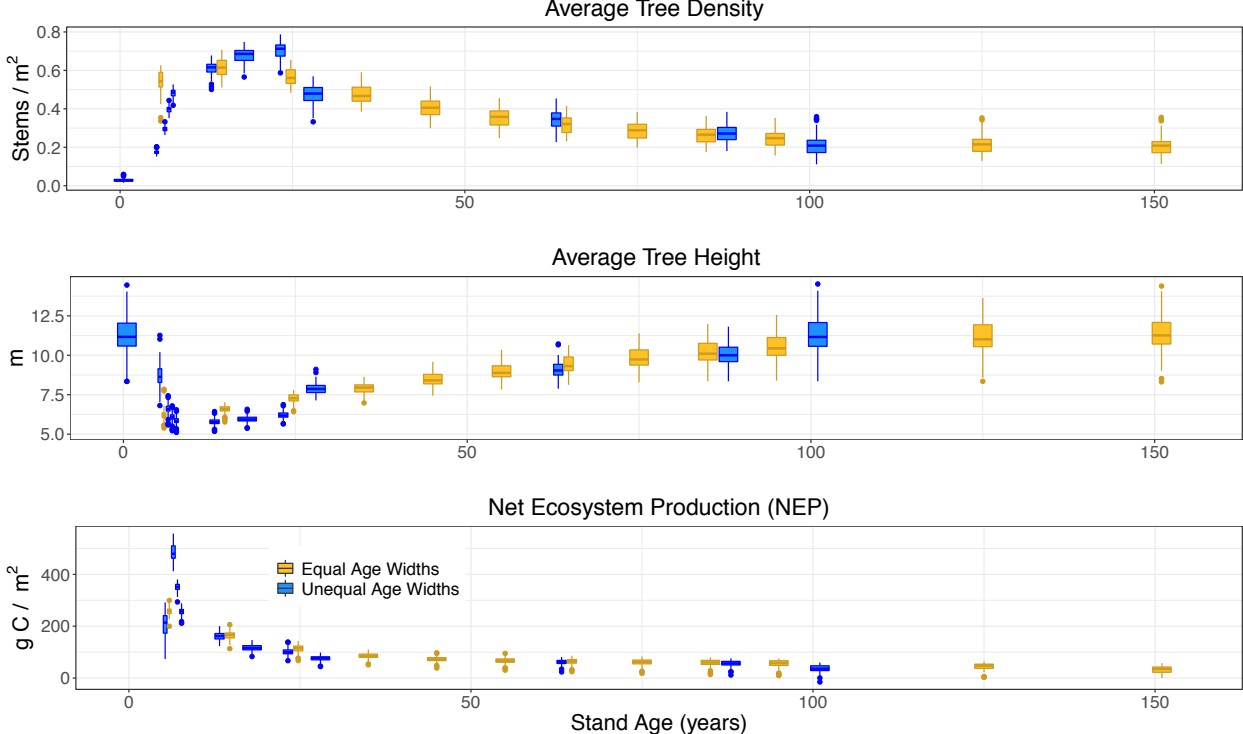

**Figure 3. Boxplots by age classes (x-axis, *in years*) from LPJ-wsl v2.0 simulations for U.S. States of MI, MN, WI. Data are from (yellow) simulations using *equal* age widths and (blue) simulations using and *unequal* age widths (*see* Table 1). Data are plotted on the x-axis at the middle age of the age class bin; note that the last age class bin for either simulation is defined as +151 yrs (equal bin) and +101 years (unequal bin). For average tree height (middle row), large tree heights in the youngest age class of the unequal bin simulation (blue) represents**
**so-called 'survivor' trees.**

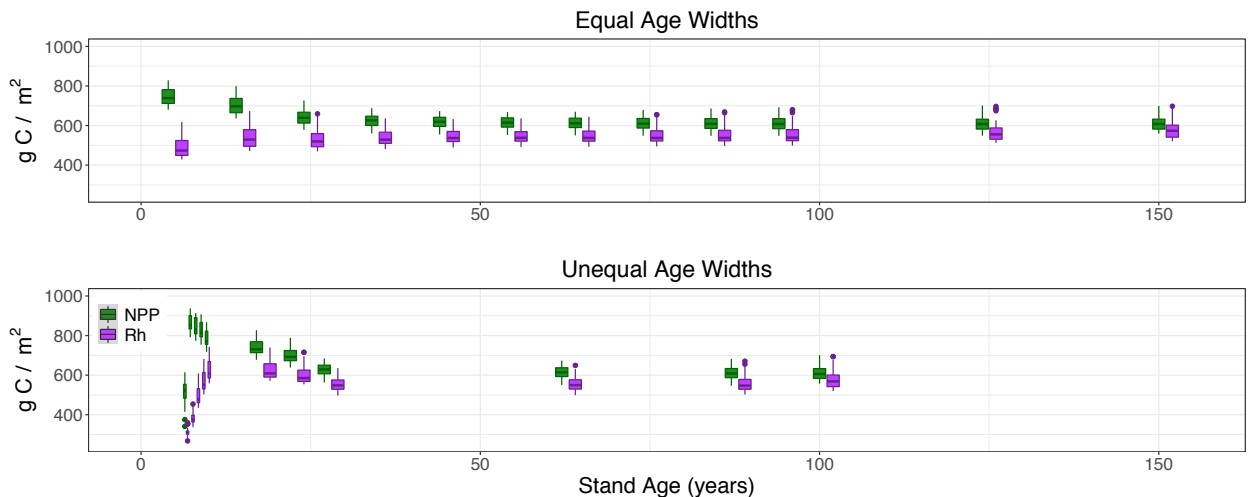

**Figure 4. Boxplots of NPP and Rh by age classes (x-axis, *in years*) from LPJ-wsl v2.0 simulations for U.S. States MI, MN, WI, for simulations using (top) equal-bin age widths and (bottom) unequal-bin age widths.**


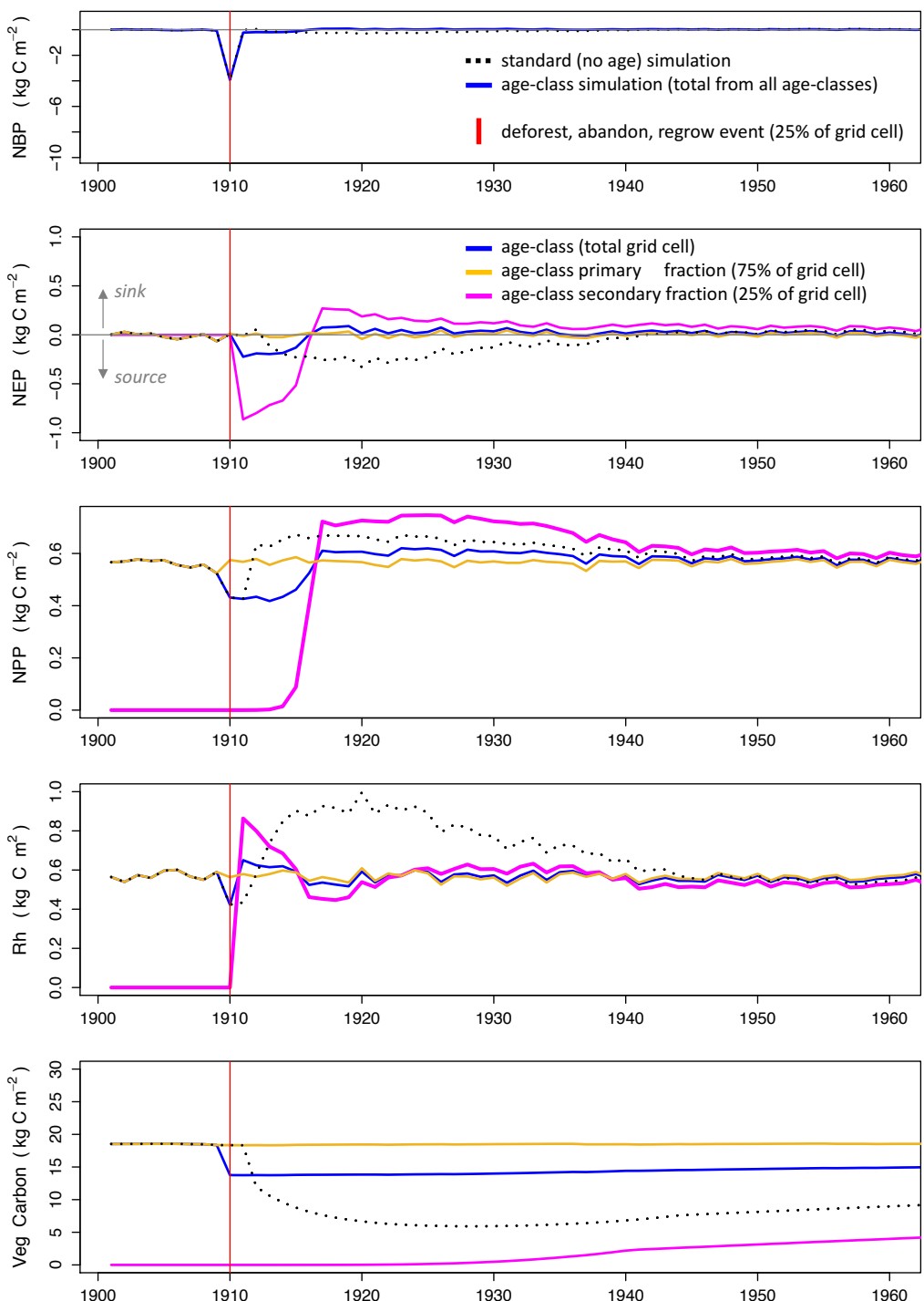

**Figure 5. A time-series comparison between the standard LPJ-wsl v2.0 simulation ($S_{noage\_event}$) and the age-class approach ($S_{age\_event}$) in an idealized single-cell simulation of a deforestation, abandonment, and subsequent re-grow event. x-axis is the simulation year. See Table 2 for simulation details.**


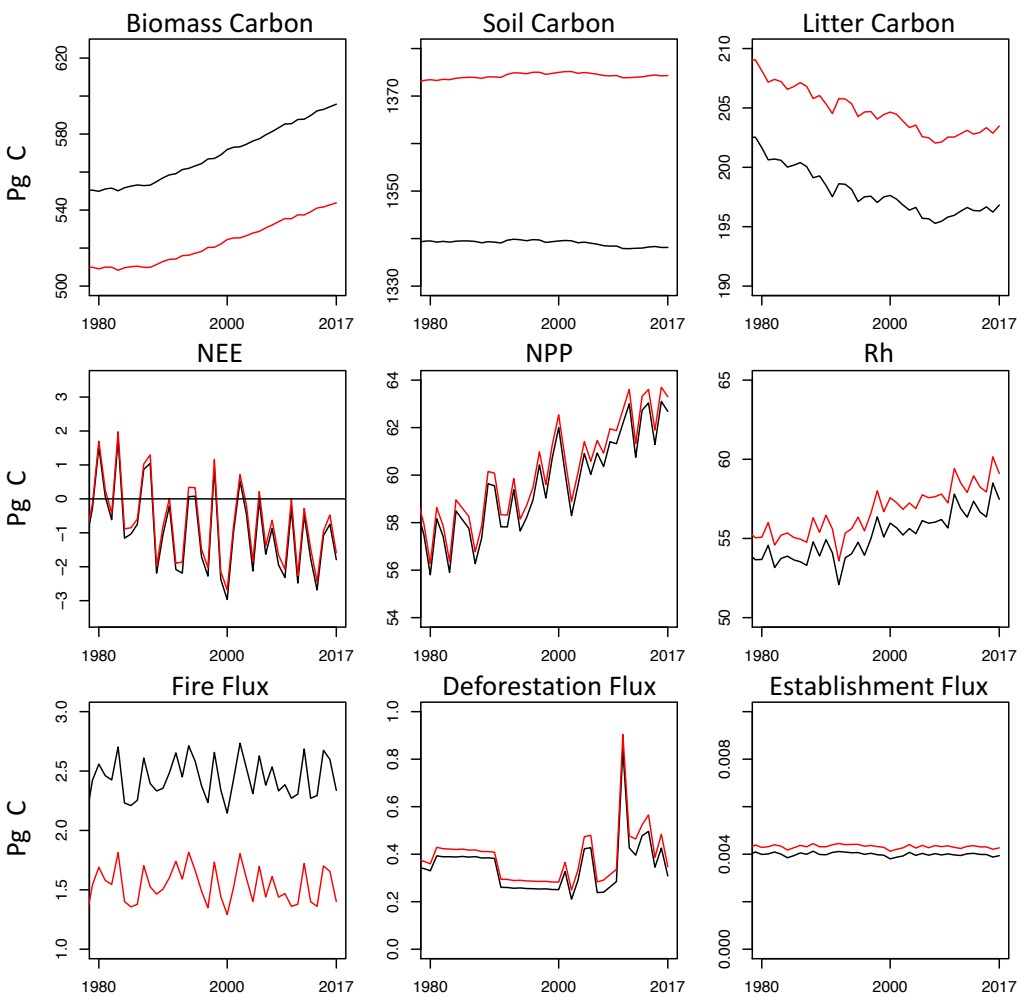

**Figure 6. Time-series of global carbon stocks and fluxes from LPJ-wsl v2.0 simulation *without* age classes (black lines) compared against simulations *with* age classes (red).**

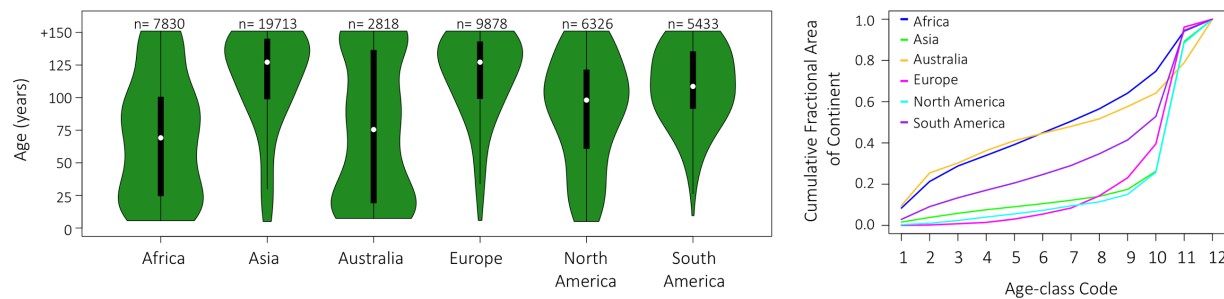


**Figure 7. Age class distributions by Continent. (left) Violin plots of ecosystem age by continent averaged over 2000-2010, based on LPJ-wsl v2.0 simulations. Violin plots show the distribution of data points (green), inter-quartile range (black box) and the median value (white circle). The number of vegetated 0.5° grid cells in each continent are above plot. (right) Cumulative fractional area in continent by age classes. Age-class codes, lowest**

**(youngest) to greatest (oldest), correspond to the *10-yr-equalbin* age-class setup (Table 1).**

Hovmöller Plots of Ecosystem Age

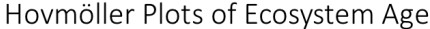

**Figure 8. Zonal ecosystem age versus year based on LPJ-wsl v2.0 simulations using full forcing (top), only fire (middle), or only land use and land cover change (bottom).**


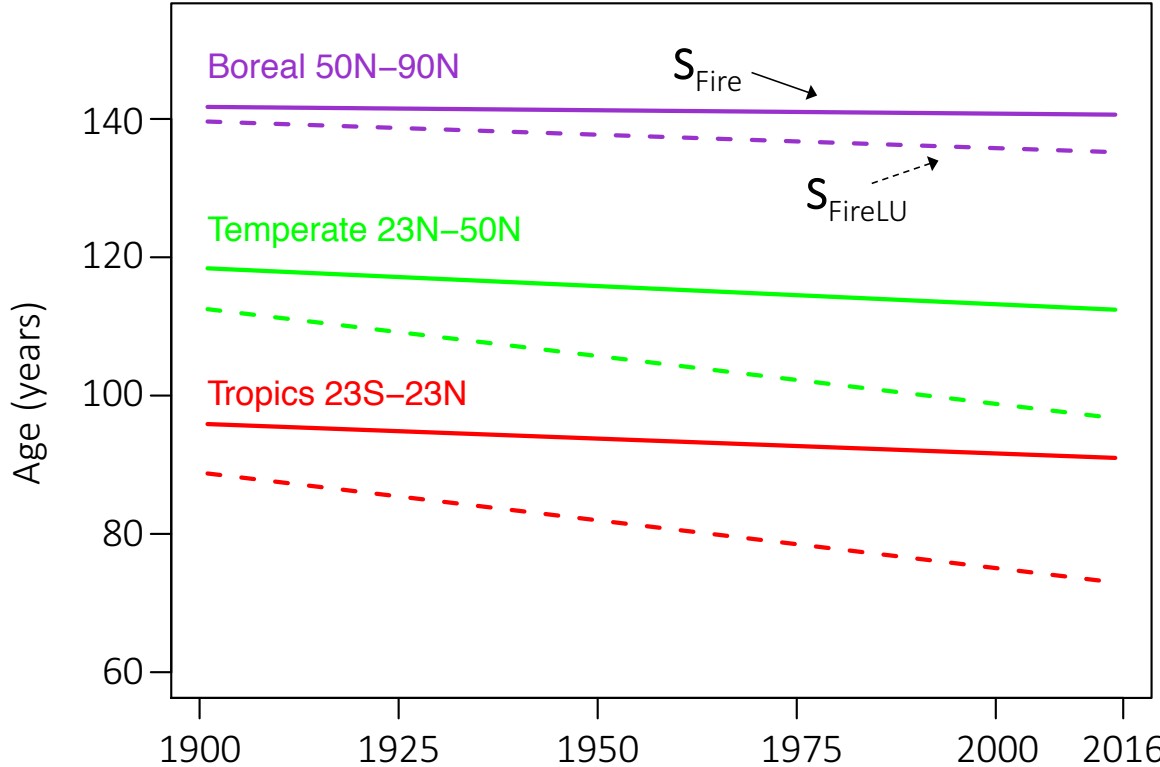

**Figure 9. Trend in ecosystem age by zonal band for LPJ-wsl v2.0 simulation with only fire ($S_{Fire}$, solid lines) and with both fire and LUCLM ($S_{FireLU}$, dashed lines). Fire causes zonal bands to differ in ecosystem age by ~23 years, and decreases the average age by 0.009 to 0.054/yr. LUCLM decreased ecosystem age at rates up to 3-times the rate of fire, from 0.038/yr in boreal zones to 0.138/yr in temperate and tropical zones.**


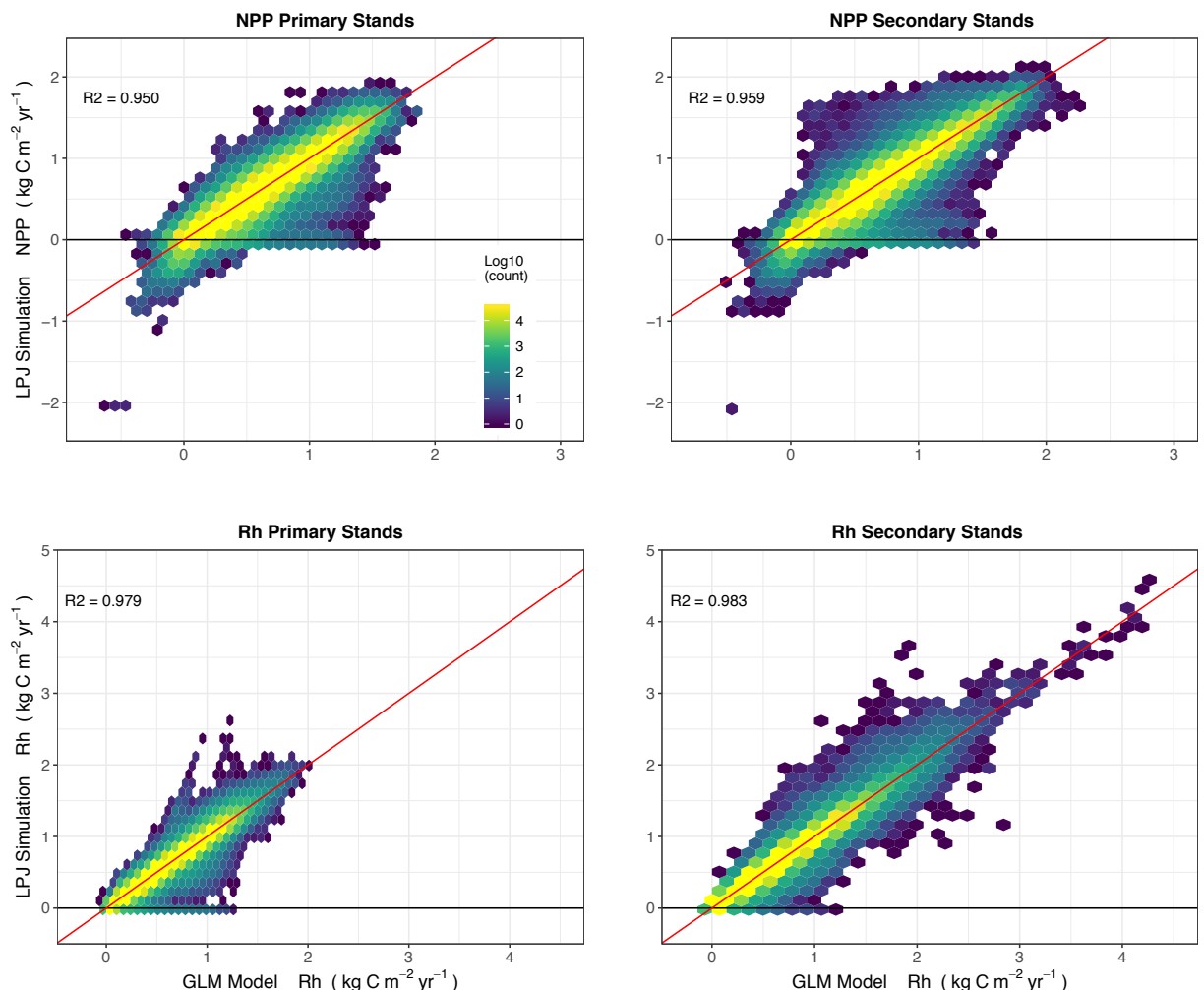

**Figure 10. Annual fluxes (NPP, Rh) (2000-2017) from LPJ-wsl v2.0 simulations versus predictions of LPJ-wsl v2.0 fluxes based on a generalized linear model (flux = precipitation + temperature + age-class); coefficients were allowed to vary by grid cell, in essence, reducing the effect of variation in plant composition, soil texture and hydrology. Coloring is by density of grid cells on a log scale; diagonal red line is the 1:1 correspondence line. The simplified statistical model can simplify the dynamics in the global vegetation model, with coefficients from the GLM helping to determine the relative importance of a small set of predictors.**


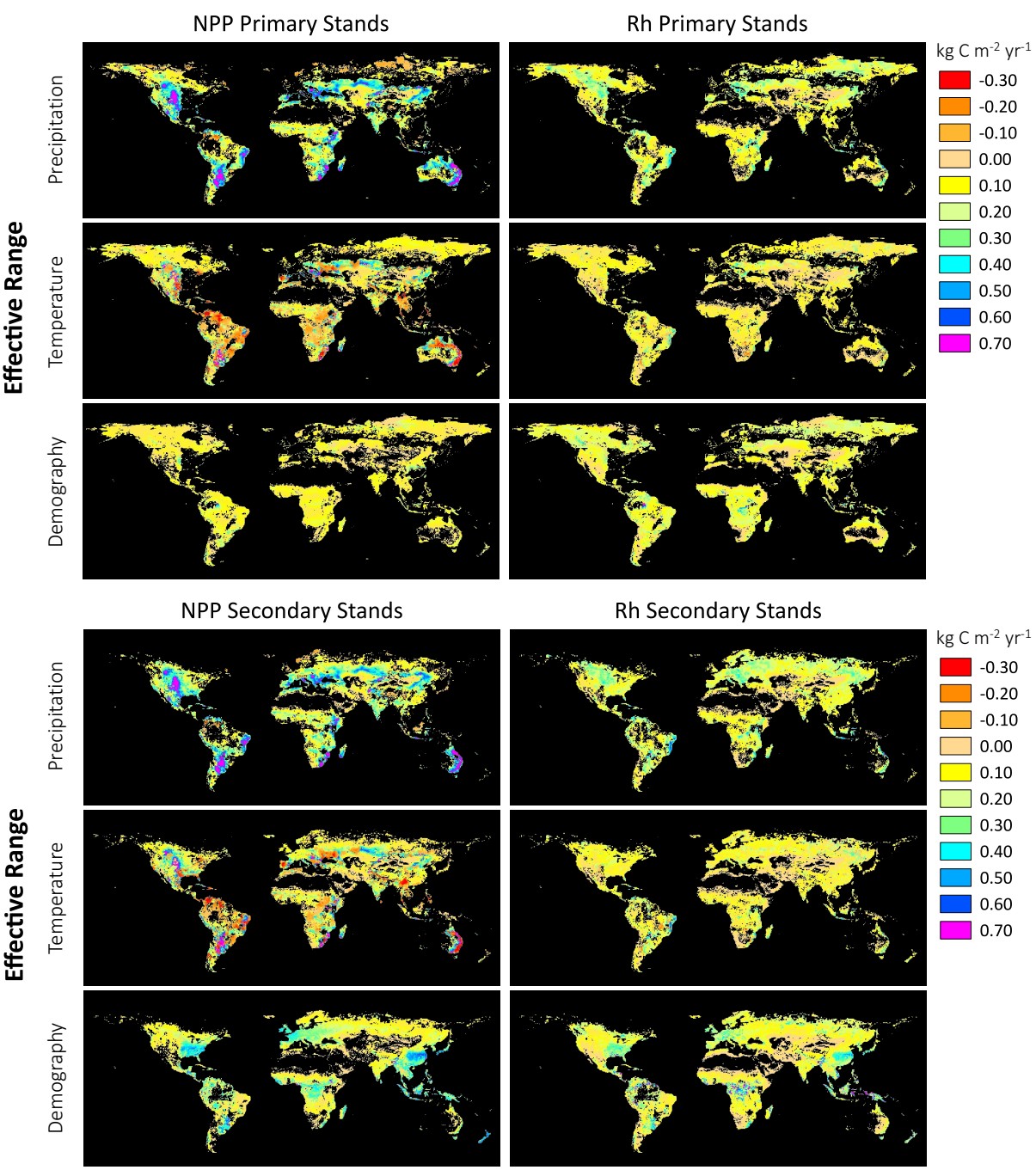

**Figure 11. Global maps of the Effective Range of the Predictors (precipitation, temperature, demography) on LPJ-wsl v2.0 fluxes (NPP, Rh); black is zero values or no-data. The Effective Range of the predictor is calculated as the gridcell-specific beta (ß) coefficient multiplied by the observed range of the predictor variable for the grid cell, for years 2000-2017. Units are on the scale of the predicted flux (kg C m$^{-2}$ yr$^{-1}$). In these maps, an emphasis is placed on the effective range of the predictor rather than the absolute value of the coefficient, although these too can be mapped for forecasting purposes. See *Sect 2.3.5* and *Sect 3.4* for additional details.**

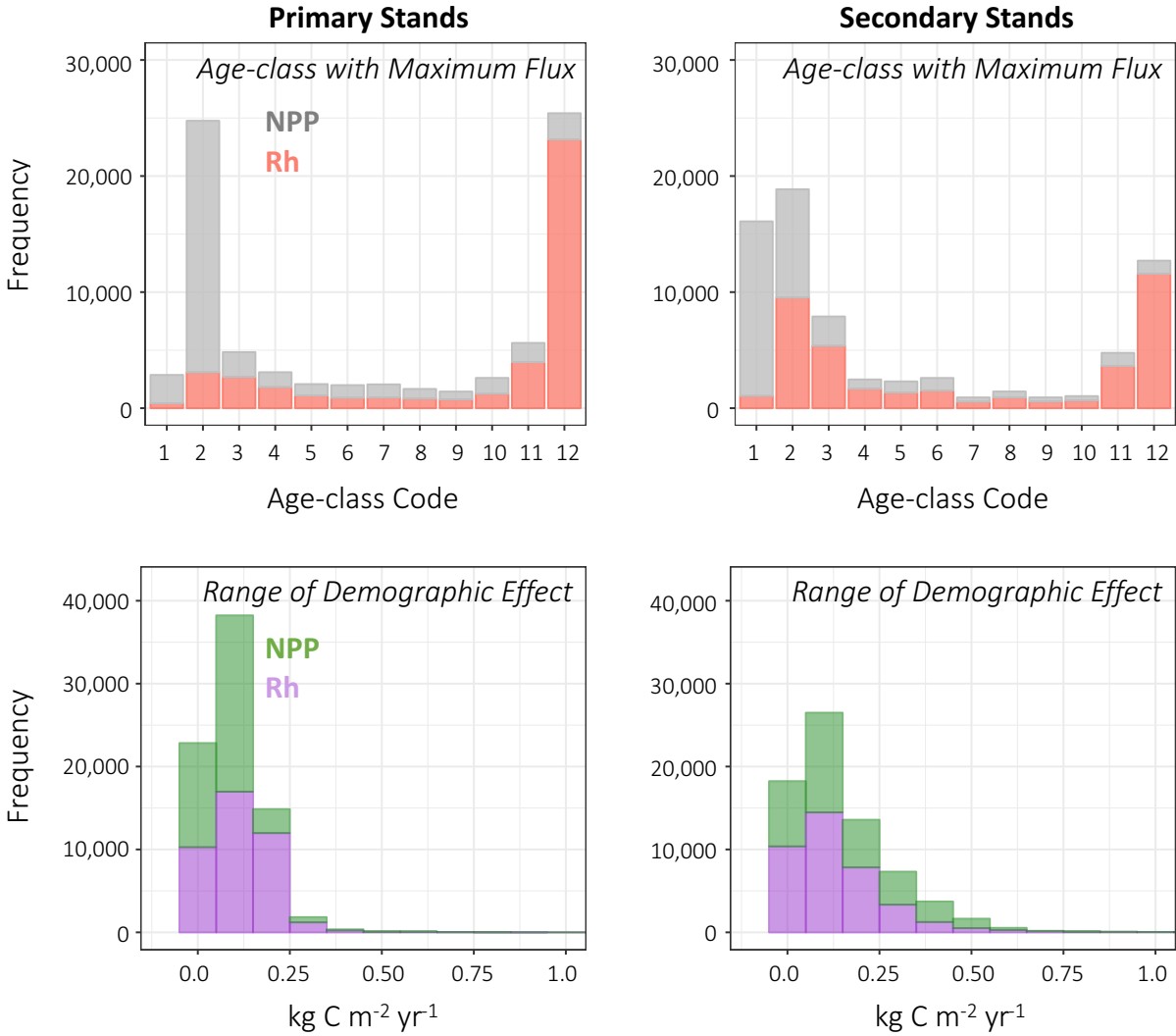

**Figure 12. Stacked frequency plots for NPP, Rh on primary and secondary stands. (top row) Global frequency of age classes with the largest flux (NPP, Rh), relative to other age classes in the grid cell. Age-class codes, lowest (youngest) to greatest (oldest), correspond to the *10-yr-equalbin* age-class setup (Table 1). (bottom row) Global frequency of the range of the demographic effect on fluxes, bin width is 0.10 kg C m$^{-2}$ yr$^{-1}$. An example interpretation, on primary stands, (top left) NPP is greatest in the second age class and (bottom left) the demographic effect on NPP is < 0.25 kg C m$^{-2}$ yr$^{-1}$.**

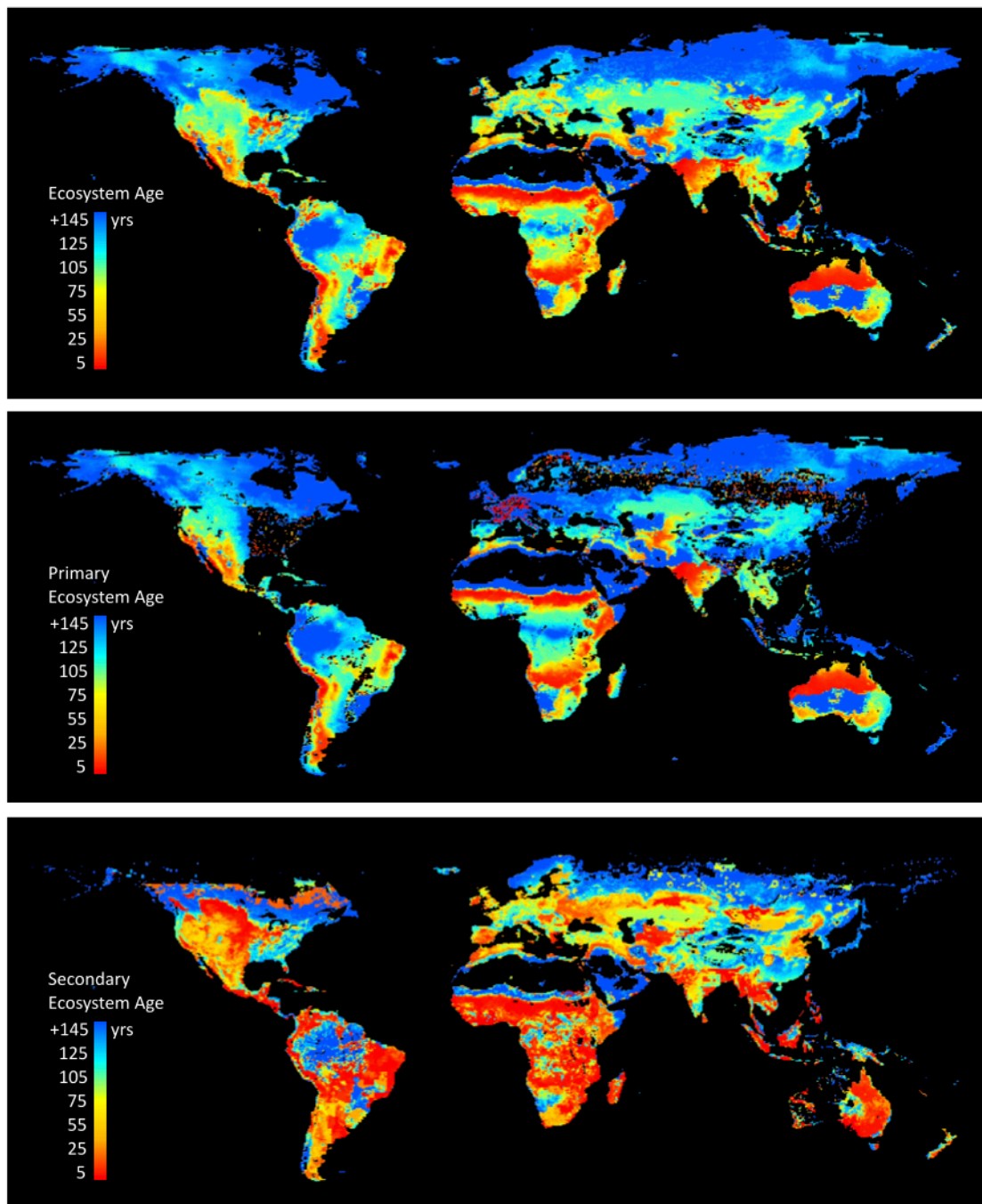


**Figure 13. LPJ-wsl v2.0 simulated global distribution of ecosystem ages, defined as the time since disturbance by fire and/or land-use change and land management (LUCLM) in year 2016. (top) Average age of the natural ecosystem, scaled to the area of natural lands within 0.5° grid cells. (middle) Average age of primary Ecosystems only, wherein only fire creates age structure, scaled to the area of primary lands. (bottom) Average age of** 1130 **secondary Ecosystems only, wherein fire and LUCLM creates age structure, scaled to the area of secondary lands.**