# Peer review of "Ecosystem age-class dynamics and distribution in the LPJ-wsl v2.0 global ecosystem model"

_Geoscientific Model Development, 2020_

## Referee Comment (RC1) · Anonymous Referee #1 · 18 Oct 2020

**General comments**

Disturbance history and disturbance regime are important drivers of terrestrial biosphere dynamics and ecosystem function, but they are rarely represented in dynamic global vegetation models. Here Calle and Poulter describe their age-class implementation in the LPJ model (LPJ-wsl v2.0), and present a series of simulations seeking to highlight the effects of disturbance history on vegetation structure and the carbon cycle, as well as the global patterns of ecosystem age when accounting for fire and land cover and land use disturbances. This work provides an important model development and can become an important contribution to the modelling community, once

some issues, which I describe below, are addressed by the authors.

The current model description provides an overview of the age structure in LPJ-wsl and includes some examples on how this module works (Figures 1 and 2). However, some mechanisms are not sufficiently described and deserve attention, especially in a journal like GMD. For example, in section 2.2.1, I could not tell how each within age-class element ($f_{i,j}$) is represented in the model: are they treated as "independent" components (i.e., available soil water and light computed independently for each within age-class element), or do all the elements in the same age class share the resources? Also, how do the age-width transitions work in the case of unequal age classes, considering that the age class transitions occur once a year? Does that mean that young age classes have fewer elements, or are multiple elements allowed to transition to another age class at the annual time step? These are mostly points for clarification and should be straightforward to address in a revised version.

The authors compare the effect of some model settings (e.g., enabling vs. disabling the age structure module), but no benchmarking is provided other than the comparison of the predicted forest structures with FIA plots. Consequently, several processes were not truly evaluated against observations or at least reported values in the literature. For example, when the authors compare the simulations with and without age-class dynamics (Figures 5, 6 and text referring to them), it is implied that the age-structure simulations are more reasonable, but the authors do not provide any reference to observations. Although the simulations are idealized, some values from literature could at least indicate whether the time scale for recovery is at least in the right order of magnitude at different biomes.

Finally, the fire disturbance is presented as the critical determinant of forest age distribution, but no assessment of the fire module is provided. I understand and agree with the authors that fire datasets such as GFED will include fire types currently not represented in LPJ-wsl and a comparison of carbon emissions is not possible due to the risk of double counting, but they could be still useful for verifying whether spatial

distribution and the inter-annual variability of fire disturbance predicted by LPJ-wsl is reasonable or not.

**Specific comments**

L58. Re-write this sentence, so it is clear that some models do account for demographic effects, including a few that were cited in the previous sentence.

L94. The authors mention permafrost and wetland methane but these features are not described anywhere. Considering that these are features in the new code, shouldn't they be described somewhere?

L132. This is a good and clear explanation, but I wonder if the authors could also highlight the consequences of adding age-classes to the representation of the micro-environments in LPJ-wsl (light, water and perhaps nutrient availability). Also, was there any reason why natural disturbances (e.g., tree fall) cannot create new age classes?

L140–155. This is not entirely accurate. In some cohort-based models, a patch represents a collection of gaps with similar forest structure. In such models, fusing patches that have similar structure simply means that the structures of patch A and patch B are sufficiently similar so that the merged patch can represent all gaps in A and B (and thus representative of a larger area). At least for ED2, the patch fusion is not determined by one state variable as implied in the text, but by the vertical LAI profile (Fisher et al. 2018).

Section 2.2.2. I understand that the fire model has been previously described, but more detail would help here, as fires are critical for the results shown later in the paper. Instead of describing the model qualitatively, the authors could provide the basic equations and also a table with the PFT-specific fire resistances (SI text and table would be fine).

L219–221. Presumably the fractional area abandoned/logged goes entirely to the youngest element within the youngest age class ($f_{0,0}$, following your notation in Eq. 4), is this correct? Clarify. Also, does it mean that the model assumes that all recently disturbed areas have similar structure of survivors? This may be fine for abandoned and clear-cut logging, but not very appropriate for fires and selective logging.

Section 3.1. Are there allometric equations that relates carbon stocks, vegetation height and stem number density in LPJ? I wonder if this could explain the consistently lower stem densities, and if the biomass distribution across size would look more/less similar to the plot data.

Section 3.1.2. I may be missing something here, but I cannot see which ecological processes are affected by choosing equal or unequal age classes. It almost reads like the only difference between the two simulations is how results are reported, please clarify the mechanistic differences between the two approaches. Also, as a point for discussion, it would be nice if the authors provided some insight of which approach is recommended.

L436–440. These results are a bit expected because recently disturbed patches are more dynamic, so having finer bins for young age-classes makes sense to me. But it is also unclear is the effect of different binning strategies on the final results.

Section 3.2.2. Is a recovery of NEP in 5–6 years more reasonable than 30 years? I don't see why, this needs some independent evidence from observations. Also, some clarification is needed to explain why Rh is consistently higher in the no-age simulation. Shouldn't the stand-scale mortality (and turnover) be the same in both cases, and the only difference be how mortality (and turnover) are applied?

L518–519. I agree with the authors on the need of more targeted simulation experiments, but if some of the variables mentioned are available from the LPJ-wsl output, then the authors could check the results to see if some hypothesized mechanisms could be ruled out.

L647. This would account for only part of the uncertainty. Parameter and process uncertainty in most models can be quite large.

L688–690. It may be worth mentioning that this size distribution may vary across regions (e.g., Espírito-Santo et al., 2014) and even within region depending on abiotic factors (e.g., Asner et al. 2013 which the authors already cite).

L700. It makes sense to end the text with a paragraph about future developments, but the current one is vague. Which specific features could be implemented and which ones should be priority?

**Minor comments**

L23. Explicitly say which latitudinal band has the lower age.

L24. Land use change and land management were. . .

L25. Does $-21$ yr correspond to both temperate and tropical areas? Clarify.

L81. "is" instead of "was"?

L98. This sentence could be dropped, considering that version control software has been around for a very long time.

L125. I don't see a strong reason to use both patch and age-class throughout the text. It makes sense to keep the explanation here but use a single term thereafter.

Eq. 4. Isn't the $f_{w,n}(t+1)$ term a form of fusion? I guess this depends on how independent the different elements within age-class are.

L175–187. Is there any reason why some of the fractional areas are fw,n and others are Fw,n? If not, then use a single notation. Also, in Eq. (5), is it correct to say that $F'_{\text{total}_j}(t) = F\text{total}_j(t) - f_{w,j}(t)$?

L202. Rewrite this sentence. Conceptually yes, the approach does seem to avoid dilution, but no example from actual model simulations was provided. Also showing that this approach works in LPJ-wsl is different than saying that the age-class/age-width approach solves the dilution issue. I am not even sure this is an issue with other models or the default LPJ, are there examples of this happening from the literature or in other LPJ simulations that the authors carried out?

L223. "to" instead of "->"

L233–235. This assumption seems counter-intuitive at least in the tropics, where young secondary forests have high deforestation rates (e.g., Nunes et al. 2020; Wang et al. 2020).

L235. At least for me, this seems the opposite of a conservative estimate.

L262. "were" instead of "was"

L275. Because readers may not be familiar with FIA plots, include the total plot area and the minimum DBH measured over the entire plot area. Also add the metric equivalents for all diameter references.

L293. Is the $5\%$ based on any real mechanism?

L306. "Data" instead of "Date"

L375. This seems a software-specific remark, mention and cite the software.

L434. Clarify this text. What is the field-based evidence, and whether the results are consistent with the evidence in a quantitative or qualitative manner (from reading the text it looks like it is the latter).

L477. What are the differences in GPP?

L484. "(?), perhaps not" is confusing.

L489. Isn't it possible to retrieve the soil moisture as a function of age from the LPJ-wsl

output? I had understood that soil moisture was solved independently for each age class.

L493. True, but the apparent large difference for other terms may be just because the scales for most variables do not go to zero in Figure 6. In relative terms they may be comparable to the changes in NEE.

L518. "drier" instead of "dryer".

L549. The central South America looks as strong as the central USA.

L593. Figure 13 could be described in the Results section, and in more detail.

L610. Including age dynamics is important, but this is not a novel concept, so it would be nice to put this paragraph into perspective with previous efforts.

Fig. 2. In case B, shouldn't 0.25 be in the 2nd row of the 3rd column, with a zero at the 1st row? Also, can logging be applied to other age-classes or just the last one? If multiple classes can be disturbed, then it may be worth showing such example too (or replacing the single-patch disturbance with a multi-patch disturbance example).

Fig. 4. It would be interesting to compare these trajectories for the two age-class approaches (equal bins, unequal bins).

Fig. 9. These results are a bit surprising given that boreal forests burn frequently. Could this be caused by the zonal averaging, which puts drylands and savannas together with low-disturbance forests in tropical and temperate zones (but not so much in the boreal zone)?

**References**

Asner G. P., Kellner J. R., Kennedy-Bowdoin T, et al. 2013. Forest canopy gap distributions in the southern Peruvian Amazon. PLoS One, 8: 1–10. doi:10.1371/journal.pone.0060875.

Espírito-Santo F. D. B., Gloor M., et al. 2014. Size and frequency of natural forest disturbances and the Amazon forest carbon balance. Nat. Commun., 5: 3434. doi:10.1038/ncomms4434.

Fisher, R. A., Koven, C. D., et al.: Vegetation demographics in Earth system models: a review of progress and priorities, Glob. Change Biol., 24, 35–54, doi:10.1111/gcb.13910, 2018.

Nunes, S., Oliveira, L., Siqueira, J., Morton, D. C., and Souza, C. M.: Unmasking secondary vegetation dynamics in the Brazilian Amazon, Environ. Res. Lett., 15, 034 057, doi:10.1088/1748-9326/ab76db, 2020.

Wang, Y., Ziv, G., Adami, M., Almeida, C. A., Antunes, J. F. G., Coutinho, A. C., Esquerdo, J. C. D. M., Gomes, A. R., and Galbraith, D.: Upturn in secondary forest clearing buffers primary forest loss in the Brazilian Amazon, Nat. Sustain., 3, 290–295, doi:10.1038/s41893-019-0470-4, 2020.

---

## Referee Comment (RC2) · Anonymous Referee #2 · 20 Oct 2020

The manuscript by Calle and Poulter investigates age-class dynamics as simulated with a dynamic global vegetation model (DGVM) called LPJ-wsl 2.0, a model developed based on the DGVM LPJ. Some aspects of this model are described in the methods, including those which were newly introduced to work with age-classes. The core of the paper seems to be a set of factorial simulations on different spatial scales used to investigate age-class dynamics together with their effect on the simulated carbon fluxes. In addition, the authors assess the contribution of the two types of modelled disturbances (fire vs land use) on forest age structure and derive a generalised linear model to predict carbon fluxes from temperature, precipitation and age-class. The latter is then used to map the "effective range" of each of the predictors to identify regions

with significant contribution of demography. I find the manuscript interesting and timely, because forest age structures are an important aspect of the (anthropogenically) disturbed terrestrial biosphere, particularly with respect to the role of land use in climate change mitigation scenarios, and since forest age structures are still underrepresented in DGVMs. In my opinion, however, several aspects of the paper need careful revisions. In particular, the main aim of the paper did not become apparent to me (see general comments below).

––

General comments:

––

1. The main aim of the paper is unclear to me and so is what the new aspects are (i.e. the gap mentioned in the abstract l.15 and in the last paragraph of the introduction ∼l.81). Is the paper supposed to be a) a model development paper, i.e. describing LPJ-wsl v2.0 or describing the implementation of age-classes in LPJ-wsl v2.0? Or is the paper b) the investigation of the simulated demographic effects? While I find detailed descriptions of models / new model development important and a legitimate scientific contribution, in my opinion, there would still be quite a bit information missing if a) would be the purpose of the paper. To me it especially did not get clear, what has been the new development and what was there before (particularly in subsection 2.2.2 – is this all new or are parts from LPJ-wsl v1.0 or even LPJ?). One aspect that could help to clarify this would be a consequent use of "LPJ-wsl v1.0" vs "LPJ-wsl v2.0" (vs LPJ) highlighting the "modifications for integration with age-classes" (l.118). (Furthermore, there are currently several occurrences of only "LPJ" which probably should be called LPJ-wsl 2.0 (e.g. Table 1, Supplementary, results section)). In addition to having clear model version references, some reordering could help, e.g. moving LPJ-wsl v1.0/LPJ aspects to 2.1.2, such as probably most aspects of fire, primary and secondary/managed forest, LUH2 driver, emissions and residues, product pools, etc.

This could, by the way, also solve the sudden occurrence of primary and secondary tiles (l.215) and the unexplained "land use" in Figure 1. From the current structure of the paper I tend to assume that b) is the main purpose / the new aspect. In this case – but to some degree this also holds for case a) – I would expect some form of comparison to observational based data, particularly for the global simulation for which the authors derive the role of demography in the global carbon cycle. On one hand, I would expect some kind of comparison of the global simulation with and without age-classes to e.g. a GPP or better AGB dataset to get a feeling for the relevance of the finding of a 40 PgC increase in turnover, and, on the other hand a comparison to a global age map, especially since one of the authors recently published such a map (Poulter et al., 2018). The comparison to a global age map could particularly be instructive to learn where the model fails to reproduce age-structures from the observational based dataset and to discuss why this might be the case (e.g. missing disturbances vs. issues with the fire algorithm or as I expect also issues with the LUH2 data – could be included e.g. in 3.3.2 and 4.1).

2. I had some problems with the way the matrix notation is presented. In general, I found the matrix description a good idea, since it quite nicely visualizes what happens upon ageing and particularly which fractions are merged into the next age-class. My critique, however, is that this is not what has been done in the code and that it also does not suit any of the two age-class setups applied in the study (Table 1). I would therefore recommend to clearly state that this is the theoretical idea, which neither suits the applied age-class setups (because they both contain unequal age widths) nor is what has been implemented in the code. Furthermore, I would appreciate a paragraph on how the age tracking is actually realised in the code.

3. The authors state that the simulated age structures are an "upper limit of age-class distributions" due to not represented disturbances (e.g. l.38, l.593) and that the study overestimates ecosystem age (l.606). However, couldn't the simulated disturbances (fire, harvest and land-use changes) also be too strong in some places? Especially with l.233-239 stating that "deforestation always occurs in the ranking of oldest to youngest age-classes… typically resulting in greater land-to-atmosphere fluxes"? Could this lead to too young forests as well as to an overestimation of historical fluxes? For the conclusions drawn in this manuscript, two additional scenarios could maybe be valuable: a) a simulation with deforestation occurring in the ranking from younger to older age-classes and b) a "best guess" simulation using the LUH2 harvest categories "harvest from secondary young forest" and "harvest from secondary mature forest". Another thought: Have the authors considered conducting the fully fledged global simulation also with the unequal bin setup? Would this lead to different results?

4. Literature work: In some of the sections I had the feeling that more references / locating the paper in context of the existing literature would be appropriate.

a) First of all I wondered if LPJ-wsl v2.0 is the model on which the publication of Pugh et al. (2019a) is based? In this case this should in my opinion clearly be stated in the paper.

b) Looking in the literature for occurrences of LPJ-wsl I found several publications that had at least short model descriptions and I wonder why none of these is referred to in the manuscript (e.g. Poulter et al, 2015; Zhang et al. 2017,2018)?

c) In section 2.2.1 l.142 when introducing the VTFT approach, the authors point to the paper by Nabel et al. (2019) having a similar independently conceived approach. Indeed, it seems as if many of the aspects described in 2.2.1 are similar to those described in Nabel et al. (2019), including the tracking of fractions per year and the merging process: merging of disturbed areas into the youngest age-class and merging of aging fractions exceeding the width of the age-class into the next age-class. Each with subsequent area-weighted averaging of carbon with the transitioning fractions. To a certain degree similarities seem to also hold for the applied age-class setups. While I truly believe that this approach has been independently conceived, I would still recommend relating to the existing approach, e.g. pointing out similarities and in

particular also differences.

d) In the discussion section I would highly recommend to conduct a bit more literature work. Several studies have already been conducted with global models that do include age-classes and I would strongly recommend having a look what is out there to compare with and refer to. For example, regarding the statements around l.625 please have a look into Yue et al. (2018) (but also e.g. Shevliakova et al., 2009). Overall, l.55-57 include quite a lot of references to models that account for demography in one way or the other but hardly any of these models occur in the discussion section and particularly e.g. statements in ∼671 are probably true for some of these models.

5. I would recommend clearly stating when simulation output is referred to as opposed to observational based data (e.g. l.20, l.32-33).

6. It did not become clear to me what exactly is compared in 2.3.2 and 3.1: Are these simulation results from a global simulation? From which? Sage? But if from Sage, why are the FIA data with disturbance, stocking or longing excluded?

7. Figure 3 and 4: I would appreciate to also have Figure 4 for the 10-year age-widths, since this is what is used in the global simulations. Also, could for ease of readability maybe all panels with unequal age-widths start with the youngest age-class? Furthermore, it might increase comparability when changing the x-axis to show linearly increasing years instead of the classes and then to place the boxes for the different age-classes at age-class mean ages. This would particularly underline the differences in the NEP dynamics among the different age-class setups. Even more so, if the two age-class setups would be integrated in one plot/panel for each of the depicted variables instead of having separate panels with differing x-axis.

8. Is there a recommendation/conclusion on what age-class setup to use based on the studied simulations? I.e. when would a simulation with unequal bins be preferable, when with equal bins or the like?

——

Specific comments/ Technical corrections:

——

- Mixed usage of hyphens: grid cell, grid-cell, gridcell; age-widths, age widths; age class, age-class; land use change -> land-use change; land use transitions -> land-use transitions; land-use -> land use, . . .

l.13 "most global ecosystem models" – consider changing to "many" acknowledging the considerable list in l.55-57.

l.15 Could you specify which gap exactly?

l.18 Could you maybe make this sentence a bit more precise? Could it also be fractions of an age-class which experience a stand-clearing disturbance? The simulated stand-clearing disturbance is fire, and the prescribed ones are harvest and abandonment of agricultural area?

l.20 "that patterns of ecosystem function" -> simulated patterns? Patterns resulting in/from model simulations?

l.24 land-use change

l.25 "an additional" –in the sentence before, with regards to fire, only the difference between boreal and tropical lats is given, maybe you could give the absolute effect there, too?

l.25 "-21 years in temperate (23N-50N) and tropical latitudes" are these analysed together or independently but experience both the same decrease in age through land-use change and land management?

l.32-33 please clarify what kind of "Spatial datasets are provided for global ecosystem age" simulated ones? Do these stem from the 'fully-fledged' LPJ-wsl v2.0 simulations?

l.35 "40-Pg C" -> "40 Pg C"

l.35-36 A 40 Pg C increase over which time period?

l.38 "upper limit" – what do you mean with upper limit? That the forest will not be younger? Please consider rephrasing/explaining. And couldn't the modelled disturbances (fire, harvest and land-use changes) also be too strong in some grid-cells leading to forest which is too young (particularly due to the applied old forest first rule; see also general comments)?

l.41-45 this seems to rather be an enumeration than a sentence and pretty long, could it maybe be taken apart and rephrased?

l.49-52 This sentence seems imprecise to me: From which of the publication exactly do the ∼60% total sink stem from? Over which time period? What are the time periods for which Pan et al. 2011b and Pugh et al. 2019a report/estimate the specified sinks, respectively? Is this in combination with changes in environmental forcings?

l.50 Pan et al. 2011b not 2011a according to the references?

l.51 really 0.3 to 1.1 PgCyr-1?

l.51 When I understood it correctly than the findings in Pugh et al. 2019a are mainly build on exactly the model being described in this study? In this case I would find the line of argumentation circular, in-transparent and therefore somehow scientifically concerning.

l.54 why is fire listed separately of "disturbances"?

l.60 but have a look at e.g. Zaehle et al. (2006) or Bellassen et al. (2010)

l.64-65: Unfortunately, I cannot find this order in Frolking et al. (2009). In section 3.1 in Frolking et al. (2009) globally disturbed fire area is largest (∼3 × 106 km2 a−1) but only 1 × 105 km2 a−1 in forest – which is equal to that estimated for wind (∼1 × 105 km2 a−1), while global estimates for wood harvest and shifting cultivation are larger

−each ∼1–2 × 105 km2 a−1 of forest area.

l.76-79: Please clarify: why forest management here – elsewhere land-use change and land management?

l.81 Could you specify which gap exactly? Else maybe omit this phrase?

l.81-83: Note: Several of the studies listed in l.55-57 have demonstrated that a representation of demography influences ecosystem stocks and/or fluxes.

l.85 is there any more recent reference than Sitch et al. 2003 (maybe Poulter et al, 2015; Zhang et al. 2017,2018)? Or maybe rephrase e.g. "a model building/based on the Lund..."?

l.110 are?

l.115 before and elsewhere in the text I understood that fire is also implemented as a stand replacing disturbance/ burned fraction moves to youngest age-class?

l.130 "unequalbin setup is applied to explore model dynamics at the level of a single grid-cell;" according to Table2 its not a single grid-cell but region, which is also suggested by e.g. Fig.4.

l.127-131: I would appreciate a bit more information on and explanation of the choices that drove the separation in age-classes. Particularly, why is the cut off with 151years in the 10-yr equal bins and why is it with 101years in the unequal bins? Why is the age range of the pre-last class (code 11) in the 10-yr equal bin larger – making it an "unequal bin", too. Maybe also the motivation for the 2, 5 and 25 year ranges as well as the switches between these ranges could shortly be outlined? If this resulted e.g. from preliminary tests, the experiences of the authors could maybe be instructive to the readers.

l.146 . . . number "of" simulated . . .

l.161 I would recommend to introduce a j on the w to indicate that the age-classes (can)

have different widths.

EQ4 and l.173 personally I find f0,0 an unlucky choice and would prefer an extra term, such as fdis or the like

EQ5 and l.179 why a capital F in $F(t)\_w,j-1$, isn't this just one entry?

l.192-199: is this an enumeration? If so, could it maybe be separated with newlines? Else I would appreciate complete sentences.

l.202 to which age-widths does this refer to, those from the unequal setup or both setups? Is there a specific section of the manuscript where "it is demonstrated" or is this a more general statement as "in this study"?

l.206 and 220 "merged with a youngest" -> the? Or can there be several youngest?

l.213 I do not understand this, why can't the not burning fraction stay in the current age-class/patch and only the burned fraction move to the youngest age-class?

l.215 This is the first time primary and secondary forest are mentioned. Also, the term tile has only been mentioned one time before ("Age-classes are represented as sub-tiles within a grid-cell"). Maybe it would help to already introduce these aspects in 2.1.1?

l.217 Does managed land refers to crop/pasture here (i.e. not forest management)?

l.225-226 mix of singular and plural?

l.224 I assume this is not a only "if the". Consider rephrasing such that it gets apparent that net zero land-use change is just one example?

l.229, l.263, l.527, l.602 consider updating to Hurtt et al. 2020

l.228 lost "and"?

l.229: I do not understand what you mean with modifications 1a (and 2a) seem not to be modified with respect to LUH2?

l.233: LUH2 offers a separation of harvest to mature and young forests. Consider shortly stating why this separation is not used in LPJ-wsl 2.0?

l.233-237: But wouldn't e.g. shifting cultivation rather make use of younger forests?

l.237-238: LUH2 offers both, harvested area and harvested biomass. Here it is stated: "until two conditions are met" and in the next sentence: "until a prescribed harvest mass or harvested area is met". This requires clarification when which of these criteria is applied.

l.244-245: I wonder if this would really be the case, I would assume that the ranking from old to younger age-classes decouples deforestation and abandonment?

l.240-... Is this new in LPJ-wsl 2.0 or is this as it has been done already before? Noticing Earles et al (2012)and McGuire et al. (2001) in 249/251 I wondered if the authors could also give the reference for the % ratios in l.240-247?

l.249-251: could you clarify which numbers are from Earles?

l.251-256: "product pool" is used twice here – with different meanings?

l.253 "dataset described further in Sect 2.3.3" – I cannot find such a description there?

l.265 "managed lands" = agricultural managed lands (since forests can also be managed?)

l.271 "accessed . . ." consider moving to references.

l.272 "fuzzed" is this relevant for this study?

l.275 Refer to SM2 here.

l.275 "model-observation comparisons" – isn't the model resolution anyway 0.5° in the compared simulations?

l.310 and regrowth?

l.318 info does not match Table 2.

l.320 it is unclear to me which of the deforestation rules from Section 2.2.2. also applies for the Snoage_event simulation, could you please give a bit more detail?

l.322 NBP so far not introduced (NPP and Rh only in the abstract).

l.330 Table lists 4 objectives/questions.

l.336 Maybe already add here for clarification that Sage = SFireLU.

l.339 "all three simulations" presumably refers to SFire, SLU and SFireLU? But what about Snoage? What was the spin-up procedure for this simulation?

l.339 does the first spin-up also has "land use values" or does it assumes only natural vegetation?

l.341 Could you please specify what you mean by 'natural conditions' – fire?

l.342 please clarify "land use values" does that mean managed agricultural land distribution? What about harvest?

l.342 please clarify: was the second spin-up procedure subsequently or alternatively for different simulations? Do all four simulations start from the same values in 1860?

l.356-359 I found this sentence a bit difficult to read since the "By contrast, fire ..." seems to refer to the "Trends in LULCM are ... prescribed" – please clarify by e.g. rephrasing.

EQ6 I wonder if the last factor should be written as a sum with age classes as index?

l.393 "age-structure patterns" – maybe "patterns of tree density and height per age"?

l.397 what does stand refers to –patch?

l.404 I do not understand this part of the sentence: "data be taken on every species; although species-level data are available".

l.417 These survivor trees make me think if a classification as "time since disturbance" would make more sense than a classification as age-classes?

l.417 LPJ? LPJ-wsl 1.0? LPJ-wsl2.0? All of them?

l.442 all "three" U.S. States.

l.452 Figure 5?

l.457 missing t in event.

l.457 LPJ? LPJ-wsl 1.0? LPJ-wsl2.0? All of them?

l.484 "(?)"?

l.487 Snosge -> Snoage

l.498 LPJ? LPJ-wsl 1.0? LPJ-wsl2.0? All of them?

l.508-509 the 23 years are not directly evident from Table 3, nor is it the decrease in zonal ecosystem age, could you help your readers specifying which of the values in Table 3 show these? This also holds for the rest of the paragraph; maybe consider extending Table 3 or adding another table showing integrated values?

l.519 grammar issue?

l.529 also here sum over B3age?

l.530 simulated NPP and Rh.

l.533 consider to delete "slightly"!

l.587 LPJ? LPJ-wsl 1.0? LPJ-wsl2.0? All of them?

l.593: again I would recommend clarifying "upper limit" and again I am not sure if this is correct, due to the oldest age-classes first principle for harvesting and deforestation (l. 233-239).

l.606 same here with the underestimation – given the oldest age-classes first principle I am not fully persuaded that it underestimation is granted.

l.622 But isn't this model dependent? Maybe consider rephrasing, e.g. "suggesting that uncertainty in carbon residence time could potentially be reduced" or the like

l.624-627: I do not agree that this is "the current state of knowledge", nor that "existing models that estimate the global land-use flux... do not include age dynamics". For the former and the latter please e.g. refer to findings of Yue et al. 2018, in addition, for the latter, the authors might have a look into other studies conducted with some of the models listed around l.55-57.

l.631: consider adapting the subsection header since this subsection seems to be more about precipitation than demographic effects?

l.662: is this only the case if using the unequal age-class setup?

l.664: LPJ? LPJ-wsl 1.0? Both?

l.671: I assume this is the case in several of the models listed around l.55-57.

l.675: consider adding "on the same machine" (if this is correct).

l.680: The first 2-3 sentences seem to be incomplete?

l.700: To my understanding JSBACH4 does not represent much vertical heterogeneity. You might want to have a look into e.g. ORCHIDEE-CAN (Naudts et al. 2015) or in individual based models (in addition to ED), e.g. LPJ-Guess (Bayer et al., 2017).

Table 1: LPJ-wsl v2.0?!

Table 2:

* single-cell: included processes might not match the description in 2.3.4, l.318.

* global: Initially I tried to associate each of the four questions with one of the simulations, due to the visual structuring of the rows of the table. Maybe merge cells and

number questions?

Figure 1: please explain what you mean with "land use" in this context

Figure 2:

* Initially I was confused by the last explanation located next to timestep 2 in panel B. Maybe it would help to place this explanation in-between timestep 1 and 2?

* fw in a3 in panel B confused me, maybe this is because a3fw = a3f1 since w=1 for a3?

* I am not sure about using ftotal in the formulas in panel B and C, would this also be correct if at the same timestep a fraction would leave an age-class and another fraction would enter the age-class (which might be the case in a large grid-cell quite often)? Instead of showing two examples for the simple merging please consider replacing example 2 (Panel C) with an example with incoming and outgoing fractions.

Figure 3: explain MI,MN & WI again.

Figure 5: consider increasing visibility by changing the y-axis of the first panel (max of -5/-6 kgCm-2).

Figure 6:

*consider adding simulation names (Sage and Snoage if I understood it correctly).

* could you show the simulation starting from the spin-up, i.e. starting 1860? Is the difference between the simulations due to the spin-up or evolving in the course of the simulation?

Figure 7:

* consider using (a) and (b) instead of left and right.

* "LPJ-wsl simulations" consider adding simulation name from Table 2. Figure 9: Since SFireLU is more complete than SFire, consider using the solid line for this more complete set-up?

Figure 10:

* consider using the same y-axis for better comparability (same SMFig1).

* l.988 model is can -> model can

Figure 11:

* l.993 "black is zero"? On the colour map it is yellow?

* red (-0.3) and pink (0.7) are difficult to distinguish, maybe consider a change in the colour map.

Figure 12: consider labelling panels (a)-(d) instead of using top row, bottom row, top left and bottom left.

——

References (only added below if not already in the manuscript):

——

Bellassen, V. et al.: Modelling forest management within a global vegetation model – Part 1: Model structure and general behaviour, Ecol. Model., 221, 2458–2474, https://doi.org/10.1016/j.ecolmodel.2010.07.008, 2010.

Hurtt et al. 2020: Harmonization of global land-use change and management for the period 850–2100 (LUH2) for CMIP6. Geosci. Model Dev., https://doi.org/10.5194/gmd-2019-360, in press.

Naudts et al. 2015: A vertically discretised canopy description for ORCHIDEE (SVN r2290) and the modifications to the energy, water and carbon fluxes, Geoscientific Model Development, 8, 2035–2065, https://doi.org/10.5194/gmd-8-2035-2015, 2015.

Poulter, B. et al. "Sensitivity of global terrestrial carbon cycle dynamics to variability in

satellite-observed burned area." Global Biogeochemical Cycles 29.2, 207-222, 2015.

Poulter, B. et al. "The global forest age dataset (GFADv1.0)", https://doi.org/10.1594/PANGAEA.889943, 2018.

Yue, C. et al.: Smaller global and regional carbon emissions from gross land use change when considering sub-grid secondary land cohorts in a global dynamic vegetation model, Biogeosciences, 15, 1185–1201, https://doi.org/10.5194/bg-15-1185-2018, 2018.

Zaehle, S. et al.: The importance of age-related decline in forest NPP for modeling regional carbon balances, Ecol. Appl., 16, 1555–1574, 2006.

Zhang, Z. et al. "Enhanced response of global wetland methane emissions to the 2015–2016 El Niño-Southern oscillation event." Environmental Research Letters 13.7, 074009, 2018.

Zhang, Z., et al. "Emerging role of wetland methane emissions in driving 21st century climate change." Proceedings of the National Academy of Sciences 114.36, 9647-9652, 2017.

---

## Short Comment (SC1) · 27 Oct 2020

Dear authors,

in my role as Executive editor of GMD, I would like to bring to your attention our Editorial version 1.2:

https://www.geosci-model-dev.net/12/2215/2019/

This highlights some requirements of papers published in GMD, which is also available on the GMD website in the 'Manuscript Types' section: http://www.geoscientific-model-development.net/submission/manuscript_types.html

In particular, please note that for your paper, the following requirement has not been met in the Discussions paper:

- Code must be published on a persistent public archive with a unique identifier for the exact model version described in the paper or uploaded to the supplement, unless this is impossible for reasons beyond the control of authors. All papers must include a section, at the end of the paper, entitled "Code availability". Here, either instructions for obtaining the code, or the reasons why the code is not available should be clearly stated. It is preferred for the code to be uploaded as a supplement or to be made available at a data repository with an associated DOI (digital object identifier) for the exact model version described in the paper. Alternatively, for established models, there may be an existing means of accessing the code through a particular system. In this case, there must exist a means of permanently accessing the precise model version described in the paper. In some cases, authors may prefer to put models on their own website, or to act as a point of contact for obtaining the code. Given the impermanence of websites and email addresses, this is not encouraged, and authors should consider improving the availability with a more permanent arrangement. Making code available through personal websites or via email contact to the authors is not sufficient. After the paper is accepted the model archive should be updated to include a link to the GMD paper.

As explained in https://www.geoscientific-model-development.net/about/manuscript_types.html the preferred reference to the code used for the ppublication is through the use of a DOI which then can be cited in the paper. For projects in GitHub a DOI for a released code version can easily be created using Zenodo, see https://guides.github.com/activities/citable-code/ for details.

Yours, Astrid Kerkweg

---

## Author Comment (AC1) · 18 Jan 2021

We have archived a permanent version of the model code, in its entirety, to Zenodo <DOI: 10.5281/zenodo.4409331>.

---

## Author Comment (AC2) · 18 Jan 2021

**Reviewer_1_ Main_Comment_001:** Disturbance history and disturbance regime are important drivers of terrestrial biosphere dynamics and ecosystem function, but they are rarely represented in dynamic global vegetation models. Here Calle and Poulter describe their age-class implementation in the LPJ model (LPJ-wsl v2.0), and present a series of simulations seeking to highlight the effects of disturbance history on vegetation structure and the carbon cycle, as well as the global patterns of ecosystem age when accounting for fire and land cover and land use disturbances. This work provides an important model development and can become an important contribution to the modelling community, once some issues, which I describe below, are addressed by the authors.

The current model description provides an overview of the age structure in LPJ-wsl and includes some examples on how this module works (Figures 1 and 2). However, some mechanisms are not sufficiently described and deserve attention, especially in a journal like GMD. For example, in section 2.2.1, I could not tell how each within ageclass element ($f_{i,j}$) is represented in the model: are they treated as "independent" components (i.e., available soil water and light computed independently for each within age-class element), or do all the elements in the same age class share the resources?

> Agreed, this could be clearer. The hierarchical structure of the model is described on L120. All ageclasses share the same gridcell inputs (climate, co2, radiation). The state variables of plant available soil water and light can differ among ageclasses, which is mainly controlled by plant water demand and plant cover, respectively.
>
> The *within* ageclass elements are not independent and every *within* ageclass element has the same exact state variables, including the same soil water and light. The *within* ageclass elements are simply a vector representation of areas for each age-unit in the ageclass. As such, we only simulate processes at the ageclass level, and the *within* ageclass elements are a simple method for a 'smooth' transition between ageclasses (i.e., no big jumps in state variables when ageclasses transition). In theory, we can simulate processes independently for each within ageclass element, but this is not practical or necessary.

Also, how do the age-width transitions work in the case of unequal age classes, considering that the age class transitions occur once a year? Does that mean that young age classes have fewer elements, or are multiple elements allowed to transition to another age class at the annual time step? These are mostly points for clarification and should be straightforward to address in a revised version.

> Correct, ageclass transitions occur only once per year. In the unequal-bin setup, young age-classes have fewer elements. Each *within* ageclass element represents the areal fraction of a single age-unit, for either setup (equal-bin and unequal-bin ageclasses). Every year, all elements increment, but each element can only increment its position once per year (rate of change is 1). Per response above, the main benefit for using unequal-bin ageclasses is to independently simulate processes and track state variables separately.

We added the text below Section 2.2.1 "An age-based model of ecosystems – sub-grid-cell patch dynamics. (bold is for emphasis, here only).

... The age-class module has a fixed number of age-classes that can be represented in a grid cell, but all age-classes are not always represented. **Age-classes are classified into 12 age-classes (patches) in fixed age-width bins, defined as the *unequalbin* or the *10yr-equalbin* age-width setup (Table 1). Each ageclass contains *within* ageclass elements, which are simply a vector representation of areas for each age-unit in the ageclass. The *within* ageclass elements are not independent and every *within* ageclass element has the same state variables, including the same soil water and light. As such, we only simulate processes at the ageclass level, and the *within* ageclass elements are a simple method for a 'smooth' transition between ageclass. In theory, we can simulate processes independently for each within ageclass element, but this is not practical or necessary. The main benefit for using equal-bin or unequal-bin ageclasses is to independently simulate processes.** The age-widths of the age-classes in the *10yr-equalbin* setup correspond to common age-widths of classes used in forest inventories. The *10yr-equal*bin age setup is used for all global simulations, whereas the *unequalbin* setup is applied to explore model dynamics at the level of a single grid-cell; simulation details in next section.

The authors compare the effect of some model settings (e.g., enabling vs. disabling the age structure module), but no benchmarking is provided other than the comparison of the predicted forest structures with FIA plots. Consequently, several processes were not truly evaluated against observations or at least reported values in the literature. For example, when the authors compare the simulations with and without age-class dynamics (Figures 5, 6 and text referring to them), it is implied that the age-structure simulations are more reasonable, but the authors do not provide any reference to observations. Although the simulations are idealized, some values from literature could at least indicate whether the time scale for recovery is at least in the right order of magnitude at different biomes.

There is value in improving modeled forest structure. The comparisons to FIA plot data are intended to provide confidence in the model's capacity to reproduce forest structural properties – a form of benchmarking. We provide a new comparison of the age distribution by continent simulated by LPJ-wsl v2.0 and compared to the Global Forest Age Database (GFAD v1.0, Poulter et al. 2018), which is derived from country-level inventory data (SM Figure 11). The comparison shows that the simulated ages are consistently older than the GFAD dataset.

This work is not intended to be a benchmarking/optimization paper, although we intend to do this in the future. Benchmarking, optimizing model parameters, identifying and improving model processes is no small task. Throughout the Discussion sections we use phrasing throughout that accounts for our uncertainty in our simulation results. We make this clear in the first section of the Discussion and we add comments to clarify our uncertain position, as below.

"(4.1) Distribution of Ecosystem Age on Earth

... Our model developments are not optimized to match observations, although we are working toward this end. Future goals are to assimilate stand-age related data, such as remotely-sense canopy data and stand index growth curves, to align model processes with observations. ..."

"(4.2 Age Dynamics Increase Turnover) ... That turnover increases when explicitly simulating ageclasses is a natural expectation, but the magnitude of the simulated turnover between carbon pools is less certain until detailed benchmarking is conducted. ..."

Finally, the fire disturbance is presented as the critical determinant of forest age distribution, but no assessment of the fire module is provided. I understand and agree with the authors that fire datasets such as GFED will include fire types currently not represented in LPJ-wsl and a comparison of carbon emissions is not possible due to the risk of double counting, but they could be still useful for verifying whether spatial distribution and the inter-annual variability of fire disturbance predicted by LPJ-wsl is reasonable or not.

The fire module was left unchanged from prior versions. The Glob-FIRM fire module has been previously evaluated in great detail by Thonicke et al. 2001, and Hanston et al. 2016. There are better efforts at answering the utility or realistic representation of simulated fire dynamics than we can do justice in this paper. This was the major aim of the Fire Model Intercomparison Project (FireMIP). We make it clear that the fire module needs improvement, it underestimates burned area, and that the resultant effect is older ecosystem ages.

In the second paragraph of the Discussion, we added clarifying remarks as below. (bold for emphasis, here only)

"...Furthermore, the fire module has been well evaluated at global scale (Thonicke et al. 2001) but **it needs improvement because it is overly simplistic and underestimates global burned area** (Hantson et al. 2020), so it is more likely that effects of fire are much greater than simulated in this study. It is clear then that this study underestimates disturbances rather than overestimates them, and as such, these simulations overestimate ecosystem age. But again, additional disturbances would only lead to younger age-classes, enhancing the role of age dynamics in regional and global carbon cycles."

Specific Comments

**Reviewer_1_Specific_Comment_001**: L58. Re-write this sentence, so it is clear that some models do account for demographic effects, including a few that were cited in the previous sentence.

The text was changed accordingly, as below, to clarify that some models already account for demographic effects.

"Following a call to the science community to improve demographic representation in models (Fisher et al. 2015), there is now a growing list of global models that are capable

of simulating global ecosystem demographics (Gitz and Ciais 2003, *Model*: OSCAR; Shevliakova et al. 2009, *Model*: LM3V; Haverd et al. 2014, *Model*: CABLE-POP; Lindeskog et al. 2013, *Model*: LPJ-GUESS; Yue et al. 2018, *Model*: ORCHIDEE MICT; Nabel et al. 2019, *Model*: JSBACH4), although more models need the capability to represent landscape heterogeneity in forest structure and function."

Note that few models that simulate the global terrestrial surface account for demography. CABLE, LPJ-GUESS and now JSBACH are the few exceptions that now include sub-grid-cell heterogeneity in ecosystem demography. ED2 does have demographic capabilities, as do many other regional and landscape-scale simulation models, but such models have not been run globally. The lack of global simulations demographic models is primarily due to the computational burden of ageclass representation, which we overcome with our methodological approach.

**Reviewer_1_Specific_Comment_002**: L94. The authors mention permafrost and wetland methane but these features are not described anywhere. Considering that these are features in the new code, shouldn't they be described somewhere?

> The LPJ-wsl v2.0 model is written as a fully modular program. Compiler flags are used to turn on/off modules. In this paper, we did not use the wetland methane or permafrost compiler flags.

**Reviewer_1_Specific_Comment_003**: L132. This is a good and clear explanation, but I wonder if the authors could also highlight the consequences of adding age-classes to the representation of the microenvironments in LPJ-wsl (light, water and perhaps nutrient availability). Also, was there any reason why natural disturbances (e.g., tree fall) cannot create new age classes?

> The ageclass module doesn't model microenvironments per se, rather it is intended to represent landscape heterogeneity, but the remark is well taken and it is a good point. The ageclass or patch size is at minimum ~2.5 km2, with a maximum of ~50 km2 (0.5 degree grid cell). Resource availability (space, light, water, *no* nutrients) is implicitly modeled as a function of a mean-individual 'big-leaf' plant functional type (PFT), with each PFT having properties of stem density, fractional plant cover, tree height, and other attributes that govern water demand and space filling properties.

> Other disturbances such as tree fall can create new age classes, yes. Our model only includes the disturbances of fire and land use and land management, but other disturbances can certainly be added. The main text has similar phrasing in the Discussion section 4.3 Opportunities for Improving Modelled Age-dynamics."

> "..There a number of opportunities for refining the age-module. Incorporating additional disturbances within the model, which will help simulate age distributions more consistent with inventory (Pan et al. 2011a) and satellite (Pugh et 680 al. 2019b) data and contribute to more scientifically relevant questions. Modeled disturbances need not be complex to explore their effects on age distributions, they only need to reset a fractional area to the youngest age-class. ..."

**Reviewer_1_Specific_Comment_004**: L140–155. This is not entirely accurate. In some cohort-based models, a patch represents a collection of gaps with similar forest structure. In such models, fusing patches that have similar structure simply means that the structures of patch A and patch B are sufficiently similar so that the merged patch can represent all gaps in A and B (and thus representative of a larger area). At least for ED2, the patch fusion is not determined by one state variable as implied in the text, but by the vertical LAI profile (Fisher et al. 2018).

> In ED2, the vertical LAI profile can still be considered a state variable of the patch, even if it is emergent from the underlying PFT cohorts. The point we make is three-fold, (1) some models do not have fixed patch size (LPJ-GUESS has a fixed patch size and patches do not merge); (2) models that have variable patch size require merging similar patches otherwise the patches could be created every year and computation will slow to a crawl. Merging is a computational solution to patch creation. (3) merging patches based on a limited set of state variables, or even a single state variable, is an arbitrary decision along a single axis of similarity between patches. We clarify as below in Section "2.2.1 An age-based model of ecosystems – sub-grid-cell patch dynamics":

> "We also employ merge age-classes (patches), but we do not employ merging rules along arbitrary axes of similarity. We fix the number of age-classes *a priori*, similar to LPJ-GUESS in that there is a maximum number of age-classes. Instead of forced merging to reduce computational burden (as in ED2), a fraction of the age-class always transitions to an older state, and a fractional area can transition and merge with the next oldest age-class."

**Reviewer_1_Specific_Comment_005**: Section 2.2.2. I understand that the fire model has been previously described, but more detail would help here, as fires are critical for the results shown later in the paper. Instead of describing the model qualitatively, the authors could provide the basic equations and also a table with the PFT-specific fire resistances (SI text and table would be fine).

> The fire module is described in greater detail in other papers (Thonicke et al. 2001, Sitch et al. 2003). Yes, the fire module is important for simulating disturbances, but we do not modify parameters in the fire module or alter the process representation in this paper. The GlobFIRM module requires much needed improvements or replacement with another fire module. The GlobFIRM module clearly underestimates burned area, both regionally and globally. The assessment of GlobFIRM, relative to other fire modules and datasets, are already reported elsewhere (Poulter et al. 2015, doi:10.1002/2013GB004655; Hantson et al. 2020, doi: 10.5194/gmd-13-3299-2020).

**Reviewer_1_Specific_Comment_006**: L219–221. Presumably the fractional area abandoned/logged goes entirely to the youngest element within the youngest age class (f0,0, following your notation in Eq. 4), is this correct?

> Yes, correct.

Clarify. Also, does it mean that the model assumes that all recently disturbed areas have similar structure of survivors? This may be fine for abandoned and clear-cut logging, but not very appropriate for fires and selective logging.

> The model does not assume the structure of survivor trees. The structure of the abandoned/logged/burned area that goes into the youngest element is determined by the underlying process. For example, if wood harvest is prescribed to an area, but the demand for harvest biomass is satisfied before all biomass is removed, then there will be 'survivor' trees on the youngest element stand. If a fire occurs on a stand, but the fire does not burn all the PFTs, then there will be survivor PFTs on the stand.

**Reviewer_1_Specific_Comment_007**: Section 3.1. Are there allometric equations that relates carbon stocks, vegetation height and stem number density in LPJ? I wonder if this could explain the consistently lower stem densities, and if the biomass distribution across size would look more/less similar to the plot data.

> Yes, there are space filling 'packing' constraints on stem density, based on allometric rules for size/height of PFTs. Yes, it could help explain the lower densities in LPJ-wsl v2.0 relative to the FIA plot data. Moreover, LPJ-wsl v2.0 does not represent vertical complexity, such as understory growth, which would increase stem density.

**Reviewer_1_Specific_Comment_008**: Section 3.1.2. I may be missing something here, but I cannot see which ecological processes are affected by choosing equal or unequal age classes. It almost reads like the only difference between the two simulations is how results are reported, please clarify the mechanistic differences between the two approaches. Also, as a point for discussion, it would be nice if the authors provided some insight of which approach is recommended.

> We clarified in Section "2.2.1 An age-based model of ecosystems – sub-grid-cell patch dynamics" as below
>
> ".. The within-ageclass elements are not independent and every within-ageclass element has the same state variables, including the same soil water and light. As such, we only simulate processes at the ageclass level, and the within-ageclass elements are a simple method for a 'smooth' transition between ageclass. In theory, we can simulate processes independently for each within-ageclass element, but this is not practical or necessary. The main benefit for using equal-bin or unequal-bin ageclasses is to independently simulate processes. .."
>
> ".. The use of equal or unequal age class setups is more than just for reporting purposes. Resources available to plants (space, light, soil water) differ between age-classes but not within age-classes, and we limit the model to represent a total of 12 ageclasses. Also, there exists a greater range of forest ages at global scales and the equal age-class setup allows us to independently model resource dynamics for more of the terrestrial surface. If we had chosen the unequal-bin setup for global simulations, we would be independently

modeling processes only for the youngest age-classes and we would lose capacity to independently model processes at intermediate and older age-classes."

**Reviewer_1_Specific_Comment_009**: L436–440. These results are a bit expected because recently disturbed patches are more dynamic, so having finer bins for young age-classes makes sense to me. But it is also unclear is the effect of different binning strategies on the final results.

> The line reference (L 436-440) was in regards to the emergent pattern in the decline in NEP with age of stand. It is generally expected that NEP declines with increasing age, yes. However, we did not expect to find such consistent patterns between NEP and stand age. We clarify as below,

> "The binning strategy is likely not a determinant of this pattern between NEP and stand age, which is evident in Figure 3 for both age-class setups. In this regard, we care less about the binning strategy and more that the emergent pattern is reflective of simulated model dynamics. This emergent pattern could lend itself to observational constraints if similar emergent patterns can be derived from forest inventory data in the future."

**Reviewer_1_Specific_Comment_010**: Section 3.2.2. Is a recovery of NEP in 5–6 years more reasonable than 30 years? I don't see why, this needs some independent evidence from observations. Also, some clarification is needed to explain why Rh is consistently higher in the no-age simulation. Shouldn't the stand-scale mortality (and turnover) be the same in both cases, and the only difference be how mortality (and turnover) are applied?

> Agreed, we state throughout that future work requires additional benchmarking or data assimilation to align model processes with observational patterns.

> After a disturbance event, Rh is consistently higher in the no-age simulation, yes. We try to explain the mechanisms that results in this model artefact in the aforementioned Section 3.2.2. Note that mortality and turnover are left unchanged in the model; these processes are the same for all model setups (no-age, equal-bin and unequal-bin setup). The processes of mortality and turnover, among all other processes, act on the state variables themselves.

**Reviewer_1_Specific_Comment_011**: L518–519. I agree with the authors on the need of more targeted simulation experiments, but if some of the variables mentioned are available from the LPJ-wsl output, then the authors could check the results to see if some hypothesized mechanisms could be ruled out.

> More simulations could help explain the fire-age zonal patterns, yes. Ideally, we first would want to make sure that the fire module aligns with burned area observations. We think such investigation is beyond the scope of the current work, and leave it simply as an open question for future investigation.

**Reviewer_1_Specific_Comment_012**: L647. This would account for only part of the uncertainty. Parameter and process uncertainty in most models can be quite large.

> Correct. The statistical model would be emulating a model defined by a specific set of parameters and processes. In an ideal world, the statistical parameters for climate sensitivity and stand age would be constrained by uncertainty simulations, bounded to realistic parameter values.

**Reviewer_1_Specific_Comment_013**: L688–690. It may be worth mentioning that this size distribution may vary across regions (e.g., Espírito-Santo et al., 2014) and even within region depending on abiotic factors (e.g., Asner et al. 2013 which the authors already cite).

> We agree with the recommendation and rephrased as below. (bold for emphasis, here only).

> "The distribution of forest gaps also has a predictable power-law relationship with size of the gap (Asner et al. 2013), **which can be allowed to vary across and within regions (Asner et al. 2013, Espírito-Santo et al. 2014),** and this fact lends itself well for representing gaps within the framework of the current age-module."

**Reviewer_1_Specific_Comment_014**: L700. It makes sense to end the text with a paragraph about future developments, but the current one is vague. Which specific features could be implemented and which ones should be priority?

> We agreed that we could do better to prioritize model improvements for the readers. The text in Section 4.3 has been updated accordingly. The beginning of the section now starts as below, with added text to support the suggestions.

> "In order of priority for improvement of the age-module: 1) improve age-class growth rates to align with observations, 2) improve representation of disturbances, 3) improve representation of early- and late-successional plant species and add vertical structural complexity such as understory/overstory canopy. Below, we provide suggestions and examples from the literature as how these improvements might be accomplished. ..."

Minor comments

**Reviewer_1_Minor_Comment_001**: L23. Explicitly say which latitudinal band has the lower age.

> Edited accordingly.

**Reviewer_1_Minor_Comment_002**: L24. Land use change and land management were. . .

> Edited accordingly.

**Reviewer_1_Minor_Comment_003**: L25. Does −21 yr correspond to both temperate and tropical areas? Clarify.

Yes, the difference (-21 yr) corresponds to both temperate and tropical zonal bands. Edited accordingly.

**Reviewer_1_Minor_Comment_004**: L81. "is" instead of "was"?

Edited accordingly.

**Reviewer_1_Minor_Comment_005**: L98. This sentence could be dropped, considering that version control software has been around for a very long time.

We agree. Edited accordingly.

**Reviewer_1_Minor_Comment_006**: L125. I don't see a strong reason to use both patch and age-class throughout the text. It makes sense to keep the explanation here but use a single term thereafter.

We agreed, we now think use of the term 'patch' causes unnecessary confusion. We replaced all instances of 'patch' with 'age-class' throughout.

**Reviewer_1_Minor_Comment_007**: Eq. 4. Isn't the $f_{w,n}(t + 1)$ term a form of fusion? I guess this depends on how independent the different elements within age-class are.

Yes, this is fusion or 'merging'. We added clarifying text to the Section 2.2.1 to explicitly say that we also merge patches, but we do not merge along axes of similarity.

"... We also merge age-classes, but we do not employ merging rules along arbitrary axes of similarity. We fix the number of age-classes *a priori*, similar to LPJ-GUESS in that there is a maximum number of age-classes. Instead of forced merging to reduce computational burden (as in ED2), a fraction of the age-class always transitions to an older state, and a fractional area can transition and merge with the next oldest age-class. ..."

**Reviewer_1_Minor_Comment_008**: L175−187. Is there any reason why some of the fractional areas are $f_{w,n}$ and others are $F_{w,n}$? If not, then use a single notation.

The text was changed to reflect single elements, $f_{w,n}$

**Reviewer_1_Minor_Comment_009**: Also, in Eq. (5), is it correct to say that $F^0\,totalj\,(t) = F_{totalj}(t) − f_{w,j}(t)$?

The meaning of the Reviewer's comment is unclear.

We edited Eq #5 to show that the sum of fractional areas for all age classes and age widths equals F_total

**Reviewer_1_Minor_Comment_010**: L202. Rewrite this sentence. Conceptually yes, the approach does seem to avoid dilution, but no example from actual model simulations was provided. Also showing that this approach works in LPJ-wsl is different than saying that the age-class/agewidth approach solves the dilution issue. I am not even sure this is an issue with other models or the default LPJ, are there examples of this happening from the literature or in other LPJ simulations that the authors carried out?

There are no other examples showing this issue in the literature. We conduct a single-pixel idealized simulation to show this effect directly. In the Panel for Veg Carbon in Figure 5, the post-disturbance biomass in the no_age simulation is diluted. This is the extreme scenario for a single stand, which can be thought of as a simulation within only 1 age class.

When averaging two numbers, the mean will always be less than the maximum value, by definition. The average over a vector of carbon densities (C m$^{-2}$), which takes into account the contributing fractional areas, will give a mean carbon density that will always be lower than the maximum carbon density in the vector. Hence a dilution of the densities will always occur. The VTFT method tries to reduce this effect. Absent computational constraints, we could represent every land fraction separately and avoid dilution.

**Reviewer_1_Minor_Comment_011**: L223. "to" instead of "->"

Edited accordingly.

**Reviewer_1_Minor_Comment_012**: L233–235. This assumption seems counter-intuitive at least in the tropics, where young secondary forests have high deforestation rates (e.g., Nunes et al. 2020; Wang et al. 2020).

We agreed. We changed the text to read "This rule will always result in greater land-to-atmosphere fluxes than if rules were employed that allowed younger age-classes to be preferentially deforested."

**Reviewer_1_Minor_Comment_013**: L235. At least for me, this seems the opposite of a conservative estimate.

Agreed, we corrected the text as above.

**Reviewer_1_Minor_Comment_014**: L262. "were" instead of "was"

Edited accordingly.

**Reviewer_1_Minor_Comment_015**: L275. Because readers may not be familiar with FIA plots, include the total plot area and the minimum DBH measured over the entire plot area. Also add the metric equivalents for all diameter references.

We edited and reword the text accordingly. For clarity, it now reads as below:

"... The FIA plot level data are composed of 4 circular sub-plot sample areas (168 m$^2$), wherein attributes of all trees with Diameter at Breast Height (DBH) ≥ 5.0 inches (12.7 cm) diameter are recorded. ..."

**Reviewer_1_Minor_Comment_016**: L293. Is the 5% based on any real mechanism?

No, it is a simple way of maintaining fractional areas in every age-class for every year of the simulation. If we did not prescribe disturbance (5% annual clearing), then might not have a distribution of age-classes within a grid-cell. Alternately, we might have a situation where young age-classes are only present once during the simulation, which could occur during dry or wet years.

**Reviewer_1_Minor_Comment_017**: L306. "Data" instead of "Date"

Edited accordingly.

**Reviewer_1_Minor_Comment_018**: L375. This seems a software-specific remark, mention and cite the software.

If the line reference above is correct (L 375), then the text refers to statistical modeling, which is not software-specific.

**Reviewer_1_Minor_Comment_019**: L434. Clarify this text. What is the field-based evidence, and whether the results are consistent with the evidence in a quantitative or qualitative manner (from reading the text it looks like it is the latter).

We edited the sentence as below for clarity. (bold is for emphasis)

"... The age-class module **qualitatively** demonstrates NEP-age relationships..."

**Reviewer_1_Minor_Comment_020**: L477. What are the differences in GPP?

Within the paper, we focus on differences in NPP as opposed to GPP, which is less certain. NPP is much more easily constrained by observations of changes in biomass.

**Reviewer_1_Minor_Comment_021**: L484. "(?), perhaps not" is confusing.

We agreed and removed the referenced text.

**Reviewer_1_Minor_Comment_022**: L489. Isn't it possible to retrieve the soil moisture as a function of age from the LPJ-wsl output? I had understood that soil moisture was solved independently for each age class.

Soil moisture is solved independently for each age-class, yes. Although we output many state variables by age-class, we currently do not have soil moisture as an output by ageclass. We think we understand the Reviewer's point. Such output could be beneficial to a focal analysis or further development of the fire module.

Regarding the context where soil moisture is mentioned in the text, the point we make is that the difference in fire fluxes between the $S_{no\_age}$ and $S_{age}$ simulations are probably less to do with soil moisture and more to do with simulating biomass heterogeneity within a grid-cell. After all, each age-class within a grid-cell receives the same exact climate inputs (precipitation, temperature). If it is hot and dry in one age-class, it will typically be hot and dry in all age-classes within a grid-cell.

**Reviewer_1_Minor_Comment_023**: L493. True, but the apparent large difference for other terms may be just because the scales for most variables do not go to zero in Figure 6. In relative terms they may be comparable to the changes in NEE.

The y-axes are all the same units. Although they are displayed on different scales, the fact that values do not go to zero does not play a role in our interpretation, nor does the relative difference among the state variables. The absolute difference is what matters in this context. It is relevant that there are compensating fluxes from Fire and Rh in the $S_{no\_age}$ and $S_{age}$ simulations which contribute to give a similar NEE value.

For clarification -- The compensating fluxes are driven by differences in the distribution of carbon among pools. When we include age-classes in the simulation and see little to no change in global NEE, someone might conclude that there is no important effect of demography. Arguably, carbon stocks in different pools (live vegetation, litter, soil) is easier to benchmark than carbon fluxes from fire or heterotrophic respiration. The differences in the component fluxes and corresponding source stocks are indeed large.

**Reviewer_1_Minor_Comment_024**: L518. "drier" instead of "dryer".

Edited accordingly.

**Reviewer_1_Minor_Comment_025**: L549. The central South America looks as strong as the central USA.

Edited accordingly. The precipitation effect generally tracks semi-arid regions, which was a good sanity check.

**Reviewer_1_Minor_Comment_026**: L610. Including age dynamics is important, but this is not a novel concept, so it would be nice to put this paragraph into perspective with previous efforts.

We revised sentences in the introduction that puts our work into better context, stating that there are existing models that simulate ecosystem demography.

**Reviewer_1_Minor_Comment_027**: Fig. 2. In case B, shouldn't 0.25 be in the 2nd row of the 3rd column, with a zero at the 1st row? Also, can logging be applied to other age-classes or just the

last one? If multiple classes can be disturbed, then it may be worth showing such example too (or replacing the single-patch disturbance with a multi-patch disturbance example).

**Reviewer_1_Minor_Comment_028:** Fig. 4. It would be interesting to compare these trajectories for the two age-class approaches (equal bins, unequal bins).

**Reviewer_1_Minor_Comment_029**: Fig. 9. These results are a bit surprising given that boreal forests burn frequently. Could this be caused by the zonal averaging, which puts drylands and savannas together with low-disturbance forests in tropical and temperate zones (but not so much in the boreal zone)?

References

Asner G. P., Kellner J. R., Kennedy-Bowdoin T, et al. 2013. Forest canopy gap distributions in the southern Peruvian Amazon. PLoS One, 8: 1–10. doi:10.1371/journal.pone.0060875.

Espírito-Santo F. D. B., Gloor M., et al. 2014. Size and frequency of natural forest disturbances and the Amazon forest carbon balance. Nat. Commun., 5: 3434. doi:10.1038/ncomms4434.

Fisher, R. A., Koven, C. D., et al.: Vegetation demographics in Earth system models: a review of progress and priorities, Glob. Change Biol., 24, 35–54, doi:10.1111/gcb.13910, 2018. Nunes, S., Oliveira, L., Siqueira, J., Morton, D. C., and Souza, C. M.: Unmasking secondary vegetation dynamics in the Brazilian Amazon, Environ. Res. Lett., 15, 034 057, doi:10.1088/1748-9326/ab76db, 2020.

Wang, Y., Ziv, G., Adami, M., Almeida, C. A., Antunes, J. F. G., Coutinho, A. C., Esquerdo, J. C. D. M., Gomes, A. R., and Galbraith, D.: Upturn in secondary forest clearing buffers primary forest loss in the Brazilian Amazon, Nat. Sustain., 3, 290–295, doi:10.1038/s41893-019-0470-4, 2020.

---

## Author Comment (AC3) · 18 Jan 2021

**Reviewer_2_ Main_Comment_001:** The manuscript by Calle and Poulter investigates age-class dynamics as simulated with a dynamic global vegetation model (DGVM) called LPJ-wsl 2.0, a model developed based on the DGVM LPJ. Some aspects of this model are described in the methods, including those which were newly introduced to work with age-classes. The core of the paper seems to be a set of factorial simulations on different spatial scales used to investigate age-class dynamics together with their effect on the simulated carbon fluxes. In addition, the authors assess the contribution of the two types of modelled disturbances (fire vs land use) on forest age structure and derive a generalised linear model to predict carbon fluxes from temperature, precipitation and age-class. The latter is then used to map the "effective range" of each of the predictors to identify regions with significant contribution of demography. I find the manuscript interesting and timely, because forest age structures are an important aspect of the (anthropogenically) disturbed terrestrial biosphere, particularly with respect to the role of land use in climate change mitigation scenarios, and since forest age structures are still underrepresented in DGVMs. In my opinion, however, several aspects of the paper need careful revisions. In particular, the main aim of the paper did not become apparent to me (see general comments below).

**Reviewer_2_General_Comment_001**: The main aim of the paper is unclear to me and so is what the new aspects are (i.e. the gap mentioned in the abstract l.15 and in the last paragraph of the introduction ~l.81). Is the paper supposed to be a) a model development paper, i.e. describing LPJ-wsl v2.0 or describing the implementation of age-classes in LPJ-wsl v2.0? Or is the paper b) the investigation of the simulated demographic effects? While I find detailed descriptions of models / new model development important and a legitimate scientific contribution, in my opinion, there would still be quite a bit information missing if a) would be the purpose of the paper. To me it especially did not get clear, what has been the new development and what was there before (particularly in subsection 2.2.2 – is this all new or are parts from LPJ-wsl v1.0 or even LPJ?). One aspect that could help to clarify this would be a consequent use of "LPJ-wsl v1.0" vs "LPJ-wsl v2.0" (vs LPJ) highlighting the "modifications for integration with age-classes" (l.118). (Furthermore, there are currently several occurrences of only "LPJ" which probably should be called LPJ-wsl 2.0 (e.g. Table 1, Supplementary, results section)). In addition to having clear model version references, some reordering could help, e.g. moving LPJ-wsl v1.0/LPJ aspects to 2.1.2, such as probably most aspects of fire, primary and secondary/managed forest, LUH2 driver, emissions and residues, product pools, etc. C2 This could, by the way, also solve the sudden occurrence of primary and secondary tiles (l.215) and the unexplained "land use" in Figure 1.

> We outline the main aims of the paper in the first sentence of the last paragraph of the Introduction,
>
>> "The overall aims of this study were to present new model developments that simulate the time-evolution of age-class distributions in a global ecosystem model and to determine if explicit representation of demography influenced ecosystem stocks and fluxes at global scales or at the level of a grid-cell."

There are two main aims of the paper. The first aim, (a) to present technical model development details. The ageclass developments are the new developments, which we tried to make clear in the title and abstract. Although the Fire and Land Use modules have not been changed, we described them in detail for completeness because these modules are integral modeled disturbances that initiate ageclasses; detail knowledge of these processes is deemed important, especially if it helps readers identify points of improvement.

We agree that the naming conventions used varied and this is confusing. We replaced all instances of model version to "LPJ-wsl v2.0"

From the current structure of the paper I tend to assume that b) is the main purpose / the new aspect. In this case – but to some degree this also holds for case a) – I would expect some form of comparison to observational based data, particularly for the global simulation for which the authors derive the role of demography in the global carbon cycle.

The second aim of the paper is to (b) demonstrate the effect of ageclass model developments on global scale dynamic vegetation simulations. Aside from showcasing the FIA comparisons to provide confidence that LPJ-wsl v2.0 can reasonably represent forest structure attributes among difference ageclasses, we tried to avoid benchmarking.

Firstly, we demonstrate that ageclass improves structural representation, and that this, in turn affects function, which we show via idealized single-cell simulations. One of the main points of the paper, however, is that the model does not simulate every disturbance. And the disturbances we do simulate (fire, land use change, wood harvest) and need improvement. As such, the results we present underestimates ecosystem ages, and therefore the results underestimate the demographic effect.

It is unclear to us as to how other state variables such as global NPP, GPP, Rh, Fire Flux, etc., would change with a realistic representation of forest ages. A benchmarking effort is beyond the scope of this paper, although we are working toward this end to improve our confidence in the flux estimates.

On one hand, I would expect some kind of comparison of the global simulation with and without age-classes to e.g. a GPP or better AGB dataset to get a feeling for the relevance of the finding of a 40 PgC increase in turnover, and, on the other hand a comparison to a global age map, especially since one of the authors recently published such a map (Poulter et al., 2018). The comparison to a global age map could particularly be instructive to learn where the model fails to reproduce age-structures from the observational based dataset and to discuss why this might be the case (e.g. missing disturbances vs. issues with the fire algorithm or as I expect also issues with the LUH2 data – could be included e.g. in 3.3.2 and 4.1).

We provide a comparison of ecosystem ages in map form and violin plots of ecosystem ages by continent for the GFAD v1.0 age map (Poulter et al. 2018). Much can be learned, even without benchmarking. We know the model underestimates ages because we lack representation of all types of disturbances, from windthrow to beetle kill to small fires.

FireMIP results (Hanston et al. 2020 GMDD) clearly demonstrate that the GlobFIRM module we use in LPJ-wsl v2.0 underestimates burned area.

**Reviewer_2_General_Comment_002**: I had some problems with the way the matrix notation is presented. In general, I found the matrix description a good idea, since it quite nicely visualizes what happens upon ageing and particularly which fractions are merged into the next age-class. My critique, however, is that this is not what has been done in the code and that it also does not suit any of the two age-class setups applied in the study (Table 1). I would therefore recommend to clearly state that this is the theoretical idea, which neither suits the applied age-class setups (because they both contain unequal age widths) nor is what has been implemented in the code. Furthermore, I would appreciate a paragraph on how the age tracking is actually realised in the code.

> We added text to clarify that the matrix formulation is the theoretical basis for the approach. In our paper, we offer four different ways of explaining the VTFT method: 1) the mathematical theoretical description, 2) a plain-language summary of the method, 3) a visual description of hypothetical examples in figure form, and 4) we provide the full model code. We understand that our approach is hard to translate so we tried four different ways of presenting the same procedure in an effort to reach the most people. As a programmer, I find it always easiest for me to view the actual code to understand the implementation more completely. In case you are interested, the main block of code is freely available (also on Zenodo):
> <https://github.com/benpoulter/LPJwsl_v2.0/blob/master/src/tools/ageclass_transition.c>

**Reviewer_2_General_Comment_003**: The authors state that the simulated age structures are an "upper limit of ageclass distributions" due to not represented disturbances (e.g. l.38, l.593) and that the study overestimates ecosystem age (l.606). However, couldn't the simulated disturbances (fire, harvest and land-use changes) also be too strong in some places? Especially with l.233-239 stating that "deforestation always occurs in the ranking of oldest to youngest age-classes. . . typically resulting in greater land-to-atmosphere fluxes"? Could this lead to too young forests as well as to an overestimation of historical fluxes?

> We removed the term 'upper limit' throughout. We added clarifying text as below.
>
> "In some locations, it is possible that our wood harvest priority rules (harvest oldest age-class first) might lead to simulated stand ages that are younger than observed stand ages if other harvest rules were applied in practice, such as preferentially logging forests of intermediate age with a goal of preserving the oldest forests from harvest. We evaluated the age distribution by continent simulated by LPJ-wsl v2.0 to the Global Forest Age Database (GFAD v1.0, Poulter et al. 2018), which is derived from country-level inventory data (SM Figure 11). The comparison shows that the simulated ages are consistently older than the GFAD dataset."
>
> The GlobFIRM fire module clearly underestimates burned area, see results from fire-model intercomparisons from FireMIP (Hanston et al. 2020); we have updated the Hanston et al. 2017 reference to the 2020 paper throughout. The FireMIP results confirms similar findings about GlobFIRM.

For the conclusions drawn in this manuscript, two additional scenarios could maybe be valuable: a) a simulation with deforestation occurring in the ranking from younger to older age-classes and b) a "best guess" simulation using the LUH2 harvest categories "harvest from secondary young forest" and "harvest from secondary mature forest".

> Certainly, there are scenarios where we will want to identify regional logging practices for which these encoded rules can be modified. That would be interesting, especially for a focal analysis on full cycle sustainable harvest practices. For simplicity and to reduce the number of 'moving targets', we chose to stay with the old-to-young harvest assumption.

Another thought: Have the authors considered conducting the fully fledged global simulation also with the unequal bin setup? Would this lead to different results?

> It is possible this would lead to different results, but most likely only in an extreme 'very young world' scenario, or in the 'fire band' latitudinal zones, where the ecosystems are relatively young. That's an interesting point, however, and a case to be made for a more flexible setup, such that the ageclass setup could be flexible to accommodate more frequent disturbances. In any case, the largest differences in ecosystem function (NPP, Rh) between the two ageclass setups (equal/unequal binning) are seen in the youngest ageclasses.

**Reviewer_2_General_Comment_004**: Literature work: In some of the sections I had the feeling that more references / locating the paper in context of the existing literature would be appropriate.

a) First of all I wondered if LPJ-wsl v2.0 is the model on which the publication of Pugh et al. (2019a) is based? In this case this should in my opinion clearly be stated in the paper.

> Agreed. We added clarifying text as below for transparency.

> "Technical details are presented for a module representing age-class dynamics, driven by fire feedbacks, land abandonment and wood harvesting in the LPJ-wsl v2.0 Dynamic Global Vegetation Model (DGVM). Prior versions of LPJ-wsl v2.0 that included early technical developments of the land use change module and the age-class module have already contributed to prior studies (Arneth et a. 2017, Kondo et al. 2018, Pugh et al. 2019a).

b) Looking in the literature for occurrences of LPJ-wsl I found several publications that had at least short model descriptions and I wonder why none of these is referred to in the manuscript (e.g. Poulter et al, 2015; Zhang et al. 2017,2018)?

> We understand the confusion. The unique feature of LPJ-wsl v2.0, including earlier versions of LPJ-wsl (as referenced above), is that it is programmed as a fully modular model. Each module can be run independently using compiler flags. This is slightly different than other DGVM models. We have maintained a practice of preserving old

code (bug-free) and adding modular updates to process representation, such that we can revert to older versions of the code. Poulter et al. 2015 does not present model developments. Modular developments for permafrost and wetland methane by Zhang et al. 2017,2018 are not 'turned on' and do not influence our simulation results. At some point, our goal is to conduct a full factorial experiment with all the modular developments, but this is not the aim of this paper.

c) In section 2.2.1 l.142 when introducing the VTFT approach, the authors point to the paper by Nabel et al. (2019) having a similar independently conceived approach. Indeed, it seems as if many of the aspects described in 2.2.1 are similar to those described in Nabel et al. (2019), including the tracking of fractions per year and the merging process: merging of disturbed areas into the youngest age-class and merging of aging fractions exceeding the width of the age-class into the next age-class. Each with subsequent area-weighted averaging of carbon with the transitioning fractions. To a certain degree similarities seem to also hold for the applied age-class setups. While I truly believe that this approach has been independently conceived, I would still recommend relating to the existing approach, e.g. pointing out similarities and in particular also differences.

> We added the following text below to the corresponding Section 2.2.1. We clearly state in the text that the VTFT method is similar to that described in Nabel et al. 2020. "The most novel advancement in this study is a new method of age-class transition modeling, which we call 'vector-tracking of fractional transitions' (VTFT), which improves the computational efficiency of modeling age-classes in global models; this is a similar approach independently conceived by Nabel et al. (2019). ". Their paper only provide a brief description of their method. Their focus appears to be on the implications of different age width binning in age class simulations. We add the following text that draws on their findings.

> "The age widths of the age-classes in the *10yr-equalbin* setup correspond to common age widths of classes used in forest inventories; for contrast, JSBACH4 uses a 15-year age width in their equal-bin ageclass setup. Most ageclasses in this setup are represented by a vector of 10 elements, wherein each element represents an aerial fraction for each age-unit (Table1)."

> A study by Nabel et al. (2020), using the demographically-enabled JSBACH4 DGVM, found that unequal binning of age widths had lower errors than equal age width binning but the largest reduction in model-observation error was achieved by simply adding more ageclasses at younger ages, regardless of the binning strategy employed.

**Reviewer_2_General_Comment_005**: I would recommend clearly stating when simulation output is referred to as opposed to observational based data (e.g. l.20, l.32-33).

> Edited accordingly.

**Reviewer_2_General_Comment_006**:  It did not become clear to me what exactly is compared in 2.3.2 and 3.1: Are these simulation results from a global simulation? From which? Sage? But if from Sage, why are the FIA data with disturbance, stocking or longing excluded?

**Reviewer_2_General_Comment_007**: Figure 3 and 4: I would appreciate to also have Figure 4 for the 10-year age-widths, since this is what is used in the global simulations. Also, could for ease of readability maybe all panels with unequal age-widths start with the youngest age-class? Furthermore, it might increase comparability when changing the x-axis to show linearly increasing years instead of the classes and then to place the boxes for the different age-classes at age-class mean ages. This would particularly underline the differences in the NEP dynamics among the different age-class setups. Even more so, if the two age-class setups would be integrated in one plot/panel for each of the depicted variables instead of having separate panels with differing x-axis.

**Reviewer_2_General_Comment_008**:  Is there a recommendation/conclusion on what age-class setup to use based on the studied simulations? I.e. when would a simulation with unequal bins be preferable, when with equal bins or the like?

> We clarified in Section "2.2.1 An age-based model of ecosystems – sub-grid-cell dynamics" as below. We think this provides recommendation to use equal-bin setup for global simulations.
>
> "... The within-ageclass elements are not independent and every within-ageclass element has the same state variables, including the same soil water and light. As such, we only simulate processes at the ageclass level, and the within-ageclass elements are a simple method for a 'smooth' transition between ageclass. In theory, we can simulate processes independently for each within-ageclass element, but this is not practical or necessary. The main benefit for using equal-bin or unequal-bin ageclasses is to independently simulate processes. ..."
>
> ".. The use of equal or unequal age class setups is more than just for reporting purposes. There resources differ between age-classes but not within age-classes, and we limit the model to represent a total of 12 ageclasses. Also, there exists a greater range of forest ages at global scales and the equal age-class setup allows us to independently model resource dynamics (space, light, water availability) for more of the terrestrial surface. If we had chosen the unequal-bin setup for global simulations, we would be independently modeling processes only for the youngest age-classes and we would lose capacity to independently model processes at intermediate and older age-classes."

Specific comments/ Technical corrections: —— - Mixed usage of hyphens: grid cell, grid-cell, gridcell; age-widths, age widths; age class, age-class; land use change -> land-use change; land use transitions -> landuse transitions; land-use -> land use, . . .

> We changed 'grid-cell' to 'grid cell' throughout, except when it was used as a joint adjective.

We changed 'age-width' to 'age width' throughout, except when it was used as a joint adjective to describe the bins.

We changed 'land use change' to 'land-use change' throughout.

We changed 'land use transitions' to 'land-use transitions.

We verified that 'land use' was used appropriately to describe the use of the land.

**Reviewer_2_Technical_Comment_001**: l.13 "most global ecosystem models" – consider changing to "many" acknowledging the considerable list in l.55-57.

Edited accordingly.

**Reviewer_2_Technical_Comment_002**: l.15 Could you specify which gap exactly?

We changed the phrasing as below.

".. This paper aims to present the technical developments of a computationally-efficient approach for representing age-class dynamics within a global ecosystem model, the LPJ-wsl v2.0 Dynamic Global Vegetation Model, and to determine if explicit representation of demography influenced ecosystem stocks and fluxes at global scales or at the level of a grid-cell. .."

**Reviewer_2_Technical_Comment_003**: l.18 Could you maybe make this sentence a bit more precise? Could it also be fractions of an age-class which experience a stand-clearing disturbance? The simulated stand clearing disturbance is fire, and the prescribed ones are harvest and abandonment of agricultural area?

A disturbance can occur on a fraction of an age-class, yes.

We rephrased for clarity as below.

".. The modeled age-classes are initially created by simulated fire, and prescribed wood harvesting or abandonment of managed land, otherwise aging naturally until an additional disturbance is simulated or prescribed. .."

**Reviewer_2_Technical_Comment_004**: l.20 "that patterns of ecosystem function" -> simulated patterns? Patterns resulting in/from model simulations?

We added clarifying text throughout to specify whether a statement refers to simulated or observed data, as in ".. that **simulated** patterns of ecosystem function .."

**Reviewer_2_Technical_Comment_005**: l.24 land-use change

Edited accordingly.

**Reviewer_2_Technical_Comment_006**: l.25 "an additional" –in the sentence before, with regards to fire, only the difference between boreal and tropical lats is given, maybe you could give the absolute effect there, too?

> We removed the wording 'additional' and simply stated that "Between simulation years 1860 and 2016, land-use change and land management were responsible.."

**Reviewer_2_Technical_Comment_007**: l.25 "-21 years in temperate (23N-50N) and tropical latitudes" are these analysed together or independently but experience both the same decrease in age through landuse change and land management?

> Temperate and Tropical latitudes were analyzed separately, see results of the statistical model presented in Table 3. But yes, they experienced the same decrease in age over time as a result of land use change and land management

**Reviewer_2_Technical_Comment_008**: l.32-33 please clarify what kind of "Spatial datasets are provided for global ecosystem age" simulated ones? Do these stem from the 'fully-fledged' LPJ-wsl v2.0 simulations?

> We clarified as below. (bold for emphasis, here only)

> "..**Simulated** spatial datasets are provided for global ecosystem age..". Yes, these simulated datasets stem from the LPJ-wsl v2.0 age-class simulations, with simulated fire and prescribed land use change and wood harvest,

**Reviewer_2_Technical_Comment_009**: l.35 "40-Pg C" -> "40 Pg C"

> Edited accordingly.

**Reviewer_2_Technical_Comment_010**: l.35-36 A 40 Pg C increase over which time period?

> Over the full simulation period the live biomass carbon in the no_age simulation is greater by ~40 Pg C, as compared to the age-class simulation. In the age-class simulation, there soil carbon is greater by ~33 Pg C and litter carbon greater by ~7 Pg C, as compared to the no_age simulation.

> We revised as below. (bold for emphasis, here only).

> "..and a finding of a 40 Pg C increase **in biomass turnover when including age dynamics** at global scales.."

**Reviewer_2_Technical_Comment_011**: l.38 "upper limit" – what do you mean with upper limit? That the forest will not be younger? Please consider rephrasing/explaining. And couldn't the modelled disturbances (fire, harvest and land-use changes) also be too strong in some grid-cells

leading to forest which is too young (particularly due to the applied old forest first rule; see also general comments)?

We removed the term 'upper limit' throughout. We added clarifying text as below.

We edited the sentence in the abstract to the following, "The LPJ-wsl v2.0 age-module represents another step forward towards understanding the role of demography in global ecosystems."

We added the following text in the Discussion, "In some locations, it is possible that our wood harvest priority rules (harvest oldest age-class first) might lead to simulated stand ages that are younger than observed stand ages if other harvest rules were applied in practice, such as preferentially logging forests of intermediate age with a goal of preserving the oldest forests from harvest. We evaluated the age distribution by continent simulated by LPJ-wsl v2.0 to the Global Forest Age Database (GFAD v1.0, Poulter et al. 2018), which is derived from country-level inventory data (SM Figure 11). The comparison shows that the simulated ages are consistently older than the GFAD dataset."

Note that the GlobFIRM fire module definitively underestimates burned area, see data for fire-model intercomparisons from FireMIP (Hanston et al. 2020); we have updated the Hanston et al. 2017 reference to the 2020 paper throughout. The FireMIP results confirms similar findings about GlobFIRM.

**Reviewer_2_Technical_Comment_012**: l.41-45 this seems to rather be an enumeration than a sentence and pretty long, could it maybe be taken apart and rephrased?

We edited as suggested.

**Reviewer_2_Technical_Comment_013**: l.49-52 This sentence seems imprecise to me: From which of the publication exactly do the ~60% total sink stem from? Over which time period? What are the time periods for which Pan et al. 2011b and Pugh et al. 2019a report/estimate the specified sinks, respectively? Is this in combination with changes in environmental forcings?

60% is an approximation as to the role of regrowth in the global land carbon sink. It is correct to suggest that this is not a settle estimate. We edited the text as below to provide greater clarity.

"On global scales, forest age is a considerable factor in global carbon cycling and the total land carbon sink ($3.2 \pm 0.8$ Pg C yr$^{-1}$ for years 2008-2017; Le Quere et al. 2018). Regrowth following disturbance is comprises a large fraction of the land sink based on estimates of the global regrowth flux from country-level forest inventories (Pan et al. 2011a; tropical regrowth sink of $1.6 \pm 0.5$ Pg C yr$^{-1}$ from 1990 to 2007). A multi-model global regrowth analysis, for which LPJ-wsl v2.0 contributed, estimated a global regrowth sink of 0.3 to 1.1 Pg C yr$^{-1}$ due to demography alone over years 1981-2010 (Pugh et al. 2019a)."

**Reviewer_2_Technical_Comment_014**: l.50 Pan et al. 2011b not 2011a according to the references?

> Correct. The Pan et al. references have been reordered so that 2011a comes first in the text. The references have been updated throughout.

**Reviewer_2_Technical_Comment_015**: l.51 really 0.3 to 1.1 PgCyr-1?

> Yes.

**Reviewer_2_Technical_Comment_016**: l.51 When I understood it correctly than the findings in Pugh et al. 2019a are mainly build on exactly the model being described in this study? In this case I would find the line of argumentation circular, in-transparent and therefore somehow scientifically concerning.

> Yes, a version of the age-module was applied in Pugh et al. 2019a. This is now stated for transparency. The age-module was never fully described or presented elsewhere previously.

**Reviewer_2_Technical_Comment_017**: l.54 why is fire listed separately of "disturbances"?

> In the line referenced, we removed the term 'disturbances' and simply stated '..land use change and land management, and fire ..'

**Reviewer_2_Technical_Comment_018**: l.60 but have a look at e.g. Zaehle et al. (2006) or Bellassen et al. (2010)

> Following a similar comment from Reviewer #1, we rephrased as below.

> "... Following a call to the science community to improve demographic representation in models (Fisher et al. 2015), there is now a growing list of global models that are capable of simulating global ecosystem demographics (Gitz and Ciais 2003, *Model*: OSCAR; Shevliakova et al. 2009, *Model*: LM3V; Haverd et al. 2014, *Model*: CABLE-POP; Lindeskog et al. 2013, *Model*: LPJ-GUESS; Yue et al. 2018, *Model*: ORCHIDEE MICT; Nabel et al. 2019, *Model*: JSBACH4), although more models need the capability to represent landscape heterogeneity in forest structure and function. ..."

**Reviewer_2_Technical_Comment_019**: l.64-65: Unfortunately, I cannot find this order in Frolking et al. (2009). In section 3.1 in Frolking et al. (2009) globally disturbed fire area is largest (~3 × 106 km2 a−1) but only 1 × 105 km2 a−1 in forest – which is equal to that estimated for wind (~1 × 105 km2 a−1), while global estimates for wood harvest and shifting cultivation are larger–each ~1–2 × 105 km2 a−1 of forest area.

> We removed the sentence from the text. We were referring to general disturbances over all ecosystems. The reviewer is correct in the Frolking reference.

**Reviewer_2_Technical_Comment_018**: l.76-79: Please clarify: why forest management here – elsewhere land-use change and land management?

Edited text, changed 'forest management' to 'LUCLM'

**Reviewer_2_Technical_Comment_020**: l.81 Could you specify which gap exactly? Else maybe omit this phrase?

The sentence was rephrased as below.

"The overall aims of this study are to present new model developments that simulate the time-evolution of age-class distributions in a global ecosystem model and ..."

**Reviewer_2_Technical_Comment_021**: l.81-83: Note: Several of the studies listed in l.55-57 have demonstrated that a representation of demography influences ecosystem stocks and/or fluxes.

We changed the text to clarify as below. (bold for emphasis, here only).

"...to determine if explicit representation of demography **in this model** influenced ecosystem stocks and fluxes..."

**Reviewer_2_Technical_Comment_022**: l.85 is there any more recent reference than Sitch et al. 2003 (maybe Poulter et al, 2015; Zhang et al. 2017,2018)? Or maybe rephrase e.g. "a model building/based on the Lund..."?

Sitch et al. 2001 is the main reference for the LPJ model. Bondeau et al. (2007) provide technical details for additional advancements, namely the agriculture module, but we do not use the agriculture module in this paper. The development history is described in Section "2.1.1 LPJ History"

**Reviewer_2_Technical_Comment_023**: l.110 are?

Edited accordingly.

**Reviewer_2_Technical_Comment_024**: l.115 before and elsewhere in the text I understood that fire is also implemented as a stand replacing disturbance/ burned fraction moves to youngest age-class?

For clarity, we changed two mentions of 'stand replacement' or 'stand-clearing' in the text as below.

".. Although pest and pathogens, namely bark beetle infestations, affected a much larger area (up to 6% of total forested area in U.S.) than both logging and fire, their effects do not always cause  immediate tree mortality. .."

".. Not all trees are killed-off when a  disturbance occurs in LPJ. ..."

**Reviewer_2_Technical_Comment_025**: l.130 "unequalbin setup is applied to explore model dynamics at the level of a single grid-cell;" according to Table2 its not a single grid-cell but region, which is also suggested by e.g. Fig.4.

We edited the text as below.

".. The *10yr-equal*bin age setup is used for all simulations including the global simulation, whereas the *unequalbin* setup is used for regional and single grid cell simulations; simulation details in next section. ..."

**Reviewer_2_Technical_Comment_026**: l.127-131: I would appreciate a bit more information on and explanation of the choices that drove the separation in age-classes. Particularly, why is the cut off with 151years in the 10-yr equal bins and why is it with 101years in the unequal bins? Why is the age range of the pre-last class (code 11) in the 10-yr equal bin larger – making it an "unequal bin", too. Maybe also the motivation for the 2, 5 and 25 year ranges as well as the switches between these ranges could shortly be outlined? If this resulted e.g. from preliminary tests, the experiences of the authors could maybe be instructive to the readers.

The binning was chosen to align with U.S. forest inventory data and we wanted greater resolution in the age-classes between 1-100. The unequal-bin setup was primarily implemented to evaluate issues with the equal-bin setup. We did not explore other binning methods as we were satisfied that the equal-bin setup was sufficient. We added text to clarify why we use the 10-year equal bin setup for global simulations as below.

".. The use of equal or unequal age class setups is more than just for reporting purposes. Resources available to plants (space, light, soil water) differ between age-classes but not within age-classes, and we limit the model to represent a total of 12 ageclasses only. Also, there exists a greater range of forest ages at global scales and the equal age-class setup allows us to independently model resource dynamics for more of the terrestrial surface. If we had chosen the unequal-bin setup for global simulations, we would be independently modeling processes only for the youngest age-classes and we would lose capacity to independently model processes at intermediate and older age-classes. .."

**Reviewer_2_Technical_Comment_027**: l.146 . . . number "of" simulated . . .

Edited accordingly.

**Reviewer_2_Technical_Comment_028**: l.161 I would recommend to introduce a j on the w to indicate that the age-classes (can) have different widths EQ4 and l.173 personally I find f0,0 an unlucky choice and would prefer an extra term, such as fdis or the like EQ5 and l.179 why a capital F in F(t)_w,j-1, isn't this just one entry?

We changed f0,0 to $f_{new}$ clarity. Yes we reference a single element; we changed the text accordingly to $f_{w,j-1}^{(t)}$

We revised Equation 2 to show that the sum of fractional areas for all patches in a grid cell is defined by the sum of fractional areas for all age classes and age widths.

**Reviewer_2_Technical_Comment_029**: l.192-199: is this an enumeration? If so, could it maybe be separated with newlines? Else I would appreciate complete sentences.

Prior text in reference "Within-class Fractional Transitions: For every simulation year, the position of each element (fx) in the VTFT vector is incremented by the representative time of each element (x), which is simply 1. No changes occur to the state variables of the age-class during within-class transitions. Between-class Fractional Transitions: Upon incre195 menting the position of each element, if the value at (fw) is non-zero, then the corresponding fractional area fw, defined as the outgoing fraction, is used in an area-weighted average between the state variables of a1 fw and the next oldest age-class a2 F_total. Lastly, upon incrementing element position, if all elements < f1 ... fw > in the VTFT vector of the preceding age-class, in this example (a1), are zeros, then the age-class is simply deleted from computational memory."

Text above changed as below, with

"The following is a description for within-class and between-class transitions. *Within-class Fractional Transitions*: For every simulation year, the position of each element ($f_x$) in the VTFT vector is incremented by the representative time of each element (*x*), which is simply 1. No changes occur to the state variables of the age-class during within-class transitions. *Between-class Fractional Transitions*: Upon incrementing the position of each element in the VTFT vector, if the value at $f_w$ is non-zero then the corresponding fractional area ($f_w$), defined as the outgoing fraction, is used in an area-weighted average between the state variables of $a_1$ *fw* and the next oldest age-class $a_2$ *F_total*. Upon incrementing element position, if all elements in the VTFT vector of the preceding age-class are zeros then the age-class is simply deleted from computational memory."

**Reviewer_2_Technical_Comment_030**: l.202 to which age-widths does this refer to, those from the unequal setup or both setups? Is there a specific section of the manuscript where "it is demonstrated" or is this a more general statement as "in this study"?

The text was edited for clarity, as below.

"Two hypothetical scenarios are provided in Figure 2 that demonstrate age-class transitions using the VTFT procedure when there is a young age-class created, and when there are fractional age-class transitions between age-classes. With VTFT, any number of age-classes and age-widths can be modeled, but it is demonstrated in this study that ..."

**Reviewer_2_Technical_Comment_031**: l.206 and 220 "merged with a youngest" -> the? Or can there be several youngest?

'youngest' is use in the singular, I'm not sure there is a plural interpretation to the word. For plural, one might say the 'young' ageclasses.

**Reviewer_2_Technical_Comment_032**: l.213 I do not understand this, why can't the not burning fraction stay in the current age-class/patch and only the burned fraction move to the youngest age-class?

Only the fraction that burns gets moved to the youngest age-class. The fraction that does not burn stays in the current age-class. The text (L 213) refers to the PFT population that does not burn completely and kill-off all the trees. The simulated burned fraction may have surviving trees.

**Reviewer_2_Technical_Comment_033**: l.215 This is the first time primary and secondary forest are mentioned. Also, the term tile has only been mentioned one time before ("Age-classes are represented as subtiles within a grid-cell"). Maybe it would help to already introduce these aspects in 2.1.1?

Removed the sentences below.

We have replaced most instances of 'patch' and 'tile' with simply 'age-class' throughout.

**Reviewer_2_Technical_Comment_034**: l.217 Does managed land refers to crop/pasture here (i.e. not forest management)?

Yes. Text edited for clarity as below.

"Age-classes get created when managed land (i.e., crop/pasture) is abandoned..."

**Reviewer_2_Technical_Comment_035**: l.225-226 mix of singular and plural?

'give' changed to 'gives'.

**Reviewer_2_Technical_Comment_036**: l.224 I assume this is not a only "if the". Consider rephrasing such that it gets apparent that net zero land-use change is just one example?

Not sure I understand the Reviewer's comment, but we edited the text as below.

"In the LUCLM module, gross transitions between land uses are simulated (Pongratz et al. 2014, Stocker et al. 2014), such that if the fraction of abandoned land equals the fraction of land deforested in the same year (net zero land use change) then the fluxes from the gross transitions are tracked independently and give an overall more accurate

accounting (and higher magnitude) of emissions from LUC than if we only tracked net transitions. ..."

**Reviewer_2_Technical_Comment_037**: l.229, l.263, l.527, l.602 consider updating to Hurtt et al. 2020 l.228 lost "and"?

The citations were updated to Hurtt et al. 2020 as suggested.

**Reviewer_2_Technical_Comment_038**: l.229: I do not understand what you mean with modifications 1a (and 2a) seem not to be modified with respect to LUH2?

The text was updated for clarity, as below.

"... but with the following modifications so that the LUHv2 data can be used in LPJ-wsl v2.0: ..."

**Reviewer_2_Technical_Comment_039**: l.233: LUH2 offers a separation of harvest to mature and young forests. Consider shortly stating why this separation is not used in LPJ-wsl 2.0?

LUHv2 does not provide distinction of stand age at finer granularity other than 'young' and 'mature'.

**Reviewer_2_Technical_Comment_040**: l.233-237: But wouldn't e.g. shifting cultivation rather make use of younger forests?

It is possible, yes.

**Reviewer_2_Technical_Comment_041**: l.237-238: LUH2 offers both, harvested area and harvested biomass. Here it is stated: "until two conditions are met" and in the next sentence: "until a prescribed harvest mass or harvested area is met". This requires clarification when which of these criteria is applied.

(1b) refers to land-use change, and land-use change is prescribed by an areal fraction. We clarified as follows, "(1b) For simplicity, deforestation (i.e., land-use change) .."

(2b) refers to wood harvest, and wood harvest is prescribed by an areal fraction and the biomass harvested on that fraction. We clarified as follows, "(2b) wood harvest (i.e., biomass harvest) also occurs in the ranking of oldest to youngest age-class ..."

**Reviewer_2_Technical_Comment_042**: l.244-245: I wonder if this would really be the case, I would assume that the ranking from old to younger age-classes decouples deforestation and abandonment?

The text (L 244-245) refers to a computational issue involved when modeling gross transitions. In a single year, crop/pasture can be abandoned (converted to secondary

forest) and forest can be converted to crop/pasture. The order in which these processes are simulated will introduce a bias, or more aptly a model artefact.

**Reviewer_2_Technical_Comment_043**: l.240-. . . Is this new in LPJ-wsl 2.0 or is this as it has been done already before? Noticing Earles et al (2012)and McGuire et al. (2001) in 249/251 I wondered if the authors could also give the reference for the % ratios in l.240-247?

> This is new in LPJ-wsl v2.0. The reference for Earles et al. (2020) is with regard to the concept of delayed emissions. The 40:60 rations are from McGuire et al. (2001), as referenced.

**Reviewer_2_Technical_Comment_044**: l.249-251: could you clarify which numbers are from Earles?

> See above.

**Reviewer_2_Technical_Comment_045**: l.251-256: "product pool" is used twice here – with different meanings?

> We think they have the same meanings. In the literal sense, 'product pool' means 'pooling products'. Another way of interpreting it would be 'carbon stock in [wood] products', or 'storage container for carbon products'.

**Reviewer_2_Technical_Comment_046**: l.253 "dataset described further in Sect 2.3.3" – I cannot find such a description there?

> We removed the text " "

**Reviewer_2_Technical_Comment_047**: l.265 "managed lands" = agricultural managed lands (since forests can also be managed?)

> 'Managed land' has different meanings in reference to LUHv2. We specify in the following sentence in the text that "In LPJ-wsl v2.0, managed lands (i.e., crop/pasture) are treated as grasslands with no irrigation, no fire, and tree PFTs were not allowed to establish."

**Reviewer_2_Technical_Comment_048**: l.271 "accessed . . ." consider moving to references.

> We decided to kept as is.

**Reviewer_2_Technical_Comment_049**: l.272 "fuzzed" is this relevant for this study?

> Yes.

**Reviewer_2_Technical_Comment_050**: l.275 Refer to SM2 here.

Edited accordingly.

**Reviewer_2_Technical_Comment_051**: l.275 "model-observation comparisons" – isn't the model resolution anyway 0.5◦ in the compared simulations?

Yes. By aggregating the plot data to larger domains (USFS Divisions) we intended to reduce the potential influence of differing climate, soils, and location between simulated data and the observations. Such aggregation has been done before for similar reasons, see Purves et al. 2008 PNAS and its supplementary materials.

**Reviewer_2_Technical_Comment_052**: l.310 and regrowth?

We left as is.

**Reviewer_2_Technical_Comment_053**: l.318 info does not match Table 2.

Yes it does. Section 2.3.4 refers to the single-cell simulation, not the Regional simulation.

**Reviewer_2_Technical_Comment_054**: l.320 it is unclear to me which of the deforestation rules from Section 2.2.2. also applies for the Snoage_event simulation, could you please give a bit more detail?

We rephrased as below for clarity.

"Treatment of deforestation byproducts (i.e., carbon in dead wood left on-site) were the same in both simulations."

**Reviewer_2_Technical_Comment_055**: l.322 NBP so far not introduced (NPP and Rh only in the abstract).

We clarified the text as follows, "Net Biome Production (NBP, defined as NBP = NEP – LUC_flux)"

**Reviewer_2_Technical_Comment_056**: l.330 Table lists 4 objectives/questions.

We edited the text accordingly, as below.

"The third and fourth objectives used data from $S_{age}$ to determine where the effect of demography was greatest and to identify the relative influence of demography versus climate on simulated fluxes (NEP, NPP, and Rh)."

**Reviewer_2_Technical_Comment_057**: l.336 Maybe already add here for clarification that Sage = SFireLU.

Edited accordingly.

**Reviewer_2_Technical_Comment_058**: l.339 "all three simulations" presumably refers to SFire, SLU and SFireLU? But what about Snoage? What was the spin-up procedure for this simulation?

We edited the text to clarify as follows, "For all global simulations.."

**Reviewer_2_Technical_Comment_059**: l.339 does the first spin-up also has "land use values" or does it assumes only natural vegetation?

We clarified as follows, "a spinup simulation...and no land use or wood harvest..."

**Reviewer_2_Technical_Comment_060**: l.341 Could you please specify what you mean by 'natural conditions' – fire?

Some of the simulations had fire turned-off. We removed the text in reference ('natural conditions') and the sentence now reads as follows, "..spinup ensured that age distributions and state variables were in dynamic equilibrium (i.e., no trend)."

**Reviewer_2_Technical_Comment_061**: l.342 please clarify "land use values" does that mean managed agricultural land distribution? What about harvest?

We restated for clarity as below, "..to initialize land use fractions of crop/pasture to year 1860..". Wood harvest was not simulated during spinup procedures.

**Reviewer_2_Technical_Comment_062**: l.342 please clarify: was the second spin-up procedure subsequently or alternatively for different simulations? Do all four simulations start from the same values in 1860?

Only simulations that used land use had a second spinup procedure. We edited the text to clarify as follows, "For simulations with land use, a second 'land-use-spinup'.."

**Reviewer_2_Technical_Comment_063**: l.356-359 I found this sentence a bit difficult to read since the "By contrast, fire . . ." seems to refer to the "Trends in LULCM are . . . prescribed" – please clarify by e.g. rephrasing.

Yes, "By contrast, fire is a fully simulated process . . ." does refer to the "Trends in LULCM are . . . prescribed". We rephrased for clarity as below.

"Trends in age distributions due to LUCLM are not prescribed by inputs per se; instead, the age module is a necessary model structure that allows full realization of the effect of forcing data on age distributions. Trends in age distribution due to Fire, which is a simulated process as opposed to prescribed, result from climate and fuel load feedbacks on fire simulation."

**Reviewer_2_Technical_Comment_064:** EQ6 I wonder if the last factor should be written as a sum with age classes as index?

The equation was edited as below, with the 'age' subscript in the ageclass term removed.

$$flux_{i,yr} = B1_i \times \text{total\_precipitation}_{i,yr} + B2_i \times \text{mean\_temperature}_{i,yr} + B3_{i,ageclass} \times ageclass_{i,yr}$$

In any manner the last term in Eq #6 is correct, it is not a sum. In the equation, the indices refer to how the terms vary. In the case of the last term in the equation, $B3_{i,age} \times ageclass_{i,yr}$, the beta coefficient (B3) vary as a function of grid cell (i) and ageclass code (*ageclass*). The ageclass code can vary as a function of grid cell (i), year (yr).

**Reviewer_2_Technical_Comment_065**: l.393 "age-structure patterns" – maybe "patterns of tree density and height per age"?

Yes, that was the meaning. We edited the text accordingly.

**Reviewer_2_Technical_Comment_066**: l.397 what does stand refers to –patch?

Yes. 'patch' changed to 'age-class' for consistency.

**Reviewer_2_Technical_Comment_067**: l.404 I do not understand this part of the sentence: "data be taken on every species; although species-level data are available".

The paragraph has been edited for clarity of meaning. The updated text is below.

"FIA data had greater variability among age-classes, regardless of Division. FIA data are not aggregated by PFT, instead they are species-level data. Changes in species composition over time do occur and it can add to the observed variability among age-classes in tree density and tree height. LPJ-wsl v2.0 includes a limited set of PFTs, which most likely limits the model's capacity to represent similar levels of variation in tree density and tree height. It is beyond the scope of this study to disentangle these patterns further, but greater agreement between observed and simulated patterns of forest structure might be acheived by including additional plant functional types that are representative of tree species for a given Division."

**Reviewer_2_Technical_Comment_068**: l.417 These survivor trees make me think if a classification as "time since disturbance" would make more sense than a classification as age-classes?

We maintain that the 'age-class' terminology is the correct terminology to use. Stand age ('age-class') is typically used to explain consistent and predictable patterns of ecosystem function and forest structure with stand age, survivor trees notwithstanding.

In the simulations, survivor trees represent a very small fraction of the PFT population. In Figure 3, the survivor trees are evident in lowest age-class of the tree height plot for the unequal-bin simulation setup. The corresponding tree density in the lowest age-class is

very low. Survivor trees are likely also present in the equal-bin setup, but the patterns for tree density and tree height are not affected.

**Reviewer_2_Technical_Comment_069**: l.417 LPJ? LPJ-wsl 1.0? LPJ-wsl2.0? All of them?

For consistency, we changed all references to the current model to LPJ-wsl v2.0.

**Reviewer_2_Technical_Comment_070**: l.442 all "three" U.S. States.

No, not 'three' states, but *all* states. The text in reference (L 442) refers the consistent and predictable pattern of NEP as a function of age. We estimated the exponent using data pooled over all states, and separately using data from each state. The exponent value was consistent.

**Reviewer_2_Technical_Comment_071**: l.452 Figure 5?

Yes. The text was edited accordingly.

**Reviewer_2_Technical_Comment_072**: l.457 missing t in event.

We cannot find the typographic error in question.

**Reviewer_2_Technical_Comment_073**: l.457 LPJ? LPJ-wsl 1.0? LPJ-wsl2.0? All of them?

We changed all references to the current model to 'LPJ-wsl v2.0'.

**Reviewer_2_Technical_Comment_074**: l.484 "(?)"?

We edited the sentenced reference as follows, "The question still remains – should there be an expectation for greater differences in NEE?"

**Reviewer_2_Technical_Comment_075**: l.487 Snosge -> Snoage

Edited accordingly.

**Reviewer_2_Technical_Comment_076**: l.498 LPJ? LPJ-wsl 1.0? LPJ-wsl2.0? All of them?

We changed all references to the current model to 'LPJ-wsl v2.0'.

**Reviewer_2_Technical_Comment_077**: l.508-509 the 23 years are not directly evident from Table 3, nor is it the decrease in zonal ecosystem age, could you help your readers specifying which of the values in Table 3 show these? This also holds for the rest of the paragraph; maybe consider extending Table 3 or adding another table showing integrated values?

We rewrote the paragraph to clarify as below. Originally, we arrived at the '23 year' difference in ages by expanding the statistical model to estimate the ecosystem age in year 1 (simulation year 1860). As an example, the age in year 1 for Boreal latitudes is given by `(141.7-(0.0098*1)) = 141.6`, the age in year 1 for Temperate latitudes is given by `(118.5-(0.0525*1)) = 118.4`, and the difference is given by `141.6 - 118.4 = 23.2 [years]`. To arrive at the age estimated in the simulation year 2016, the year index is 157, so the age would be given by `(141.7-(0.0098*157)) = 140.1614`.

"Ecosystem age by zonal band was oldest at boreal latitudes, followed by temperature latitudes, and youngest in tropical latitudes, which was primarily the results of frequent fires in simulated grassland ecosystems. The primary driver of zonal age distributions was Fire (Figure 8). According to results from the statistical model (Table 3), the average age difference due to fire among zonal bands in 1860 was 23 years between Boreal (older) and Temperature (younger) latitudes, and it was 32 years between temperature (older) and tropical (younger) latitudes. The difference in ecosystem age among zonal bands increased to 60 years in simulation year 2016 between boreal and temperate latitudes, while the difference in ages between temperature and tropical latitudes remained similar (31 yr age difference). There was a statistically significant decrease in zonal ecosystem age over time due to fire (Table 3)"

**Reviewer_2_Technical_Comment_078**: l.519 grammar issue?

Not sure we understand the Reviewer's concern with the sentence in question.

**Reviewer_2_Technical_Comment_079**: l.529 also here sum over B3age?

No, there is no summation in the statistical model.

**Reviewer_2_Technical_Comment_080**: l.530 simulated NPP and Rh.

Yes, we added 'simulated' to clarify we mean the simulated NPP and Rh.

**Reviewer_2_Technical_Comment_081**: l.533 consider to delete "slightly"!

Edited accordingly. The difference is more than 'slight', thanks.

**Reviewer_2_Technical_Comment_082**: l.587 LPJ? LPJ-wsl 1.0? LPJ-wsl2.0? All of them?

We changed all references to the current model to 'LPJ-wsl v2.0'.

**Reviewer_2_Technical_Comment_083**: l.593: again I would recommend clarifying "upper limit" and again I am not sure if this is correct, due to the oldest age-classes first principle for harvesting and deforestation (l. 233-239).

Per previous response, see below.

We removed the term 'upper limit' throughout. We added clarifying text as below.

"In some locations, it is possible that our wood harvest priority rules (harvest oldest age-class first) might lead to simulated stand ages that are younger than observed stand ages if other harvest rules were applied in practice, such as preferentially logging forests of intermediate age with a goal of preserving the oldest forests from harvest. We evaluated the age distribution by continent simulated by LPJ-wsl v2.0 to the Global Forest Age Database (GFAD v1.0, Poulter et al. 2018), which is derived from country-level inventory data (SM Figure 11). The comparison shows that the simulated ages are consistently older than the GFAD dataset."

**Reviewer_2_Technical_Comment_084**: l.606 same here with the underestimation – given the oldest age-classes first principle I am not fully persuaded that it underestimation is granted.

Per previous response, see below.

We added text to clarify that we think our simulations represent the upper bound of age distributions, where 'bound' is meant to convey a range of values [lower, upper] of expectation. We added clarifying text as below.

"In some locations, it is possible that our wood harvest priority rules (harvest oldest age-class first) might lead to simulated stand ages that are younger than observed stand ages if other harvest rules were applied in practice, such as preferentially logging forests of intermediate age with a goal of preserving the oldest forests from harvest. We evaluated the age distribution by continent simulated by LPJ-wsl v2.0 to the Global Forest Age Database (GFAD v1.0, Poulter et al. 2018), which is derived from country-level inventory data; we have added this comparison as a figure to the Supplement. The comparison shows that the simulated ages are consistently older than the GFAD dataset."

The GlobFIRM fire module underestimates burned area, see data for fire-model intercomparisons from FireMIP (Hanston et al. 2020); we have updated the Hanston et al. 2017 reference to the 2020 paper throughout. The FireMIP results confirms similar findings about GlobFIRM.

**Reviewer_2_Technical_Comment_085**: l.622 But isn't this model dependent? Maybe consider rephrasing, e.g. "suggesting that uncertainty in carbon residence time could potentially be reduced" or the like

Agreed, it can be model dependent. We added the clarifying text '..could potentially be reduced..' as suggested.

**Reviewer_2_Technical_Comment_086**: l.624-627: I do not agree that this is "the current state of knowledge", nor that "existing models that estimate the global land-use flux... do not include age dynamics". For the former and the latter please e.g. refer to findings of Yue et al. 2018, in addition, for the latter, the authors might have a look into other studies conducted with some of the models listed around l.55-57.

We respectfully disagree. Yue et al. 2018 report on a single grid cell (0.5 degree) idealized simulation and it does not represent consensus on the state of knowledge. We refer to global emissions from gross land use change being greater than net land use change based on Arneth et al. 2017.

**Reviewer_2_Technical_Comment_087**: l.631: consider adapting the subsection header since this subsection seems to be more about precipitation than demographic effects?

We added some text to the section in reference so that the section title reflects the content of the paragraphs therein.

**Reviewer_2_Technical_Comment_088**: l.662: is this only the case if using the unequal age-class setup?

No, NEP is greatest in the youngest age-class, regardless of the simulation setup.

**Reviewer_2_Technical_Comment_089**: l.664: LPJ? LPJ-wsl 1.0? Both?

We changed all references to the current model to 'LPJ-wsl v2.0'.

**Reviewer_2_Technical_Comment_090**: l.671: I assume this is the case in several of the models listed around l.55-57.

We aren't sure, one way or another, if this applies to other models. We know this does apply to LPJ-wsl v2.0.

**Reviewer_2_Technical_Comment_091**: l.675: consider adding "on the same machine" (if this is correct).

Not correct. LPJ-wsl v2.0 can be run distributed, in parallel, on multiple compute 'machines'.

**Reviewer_2_Technical_Comment_092**: l.680: The first 2-3 sentences seem to be incomplete?

The section was re-written and begins with the statements as below.

"In order of priority for improvement of the age-module: 1) improve age-class growth rates to align with observations, 2) improve representation of disturbances, 3) improve representation of early- and late-successional plant species and add vertical structural complexity such as understory/overstory canopy. Below, we provide suggestions and examples from the literature as how these improvements might be accomplished."

**Reviewer_2_Technical_Comment_093**: l.700: To my understanding JSBACH4 does not represent much vertical heterogeneity. You might want to have a look into e.g. ORCHIDEE-

CAN (Naudts et al. 2015) or in individual based models (in addition to ED), e.g. LPJ-Guess (Bayer et al., 2017).

> The phrasing 'much' is relative. Relative to LPJ-wsl v2.0, which has a single layer canopy, JSBACH4 provides a good example for future developments.

**Reviewer_2_Technical_Comment_094**: Table 1: LPJ-wsl v2.0?!

> As above, we changed all references to the current model to 'LPJ-wsl v2.0'.

**Reviewer_2_Technical_Comment_095**: Table 2: * single-cell: included processes might not match the description in 2.3.4, l.318. * global: Initially I tried to associate each of the four questions with one of the simulations, due to the visual structuring of the rows of the table. Maybe merge cells and number questions?

> We choose to leave as is. The questions are meant to be interpreted as 'objectives and questions', as in general questions we wished to answer using the global simulation data. There are other questions we address in the text for each simulation. The column was not meant to be a full enumeration of all research questions and associated findings we address within the main text.

**Reviewer_2_Technical_Comment_096**: Figure 1: please explain what you mean with "land use" in this context

> In the context of LPJ-wsl v2.0, 'Land Use' refers to crop/pasture.

**Reviewer_2_Technical_Comment_097**: Figure 3: explain MI,MN & WI again.

> We clarified as follows, "..for U.S. States of MI, MN, MN.."

**Reviewer_2_Technical_Comment_098**: Figure 5: consider increasing visibility by changing the y-axis of the first panel (max of -5/-6 kgCm-2).

> We leave as is. We think the important content of the panel is sufficiently displayed while leaving room for the legend.

**Reviewer_2_Technical_Comment_099**: Figure 6: *consider adding simulation names (Sage and Snoage if I understood it correctly). * could you show the simulation starting from the spin-up, i.e. starting 1860? Is the difference between the simulations due to the spin-up or evolving in the course of the simulation?

> The simulations have the same prescribed drivers (inputs). The differences between simulations are observed after spinup, yes.

**Reviewer_2_Technical_Comment_100**: Figure 7: * consider using (a) and (b) instead of left and right. * "LPJ-wsl simulations" consider adding simulation name from Table 2.

We kept 'left' and 'right'.

**Reviewer_2_Technical_Comment_101**: Figure 9: Since SFireLU is more complete than SFire, consider using the solid line for this more complete set-up?

We left as is. We annotate each line in the plot with the associated simulation.

**Reviewer_2_Technical_Comment_102**: Figure 10: * consider using the same y-axis for better comparability (same SMFig1). *

We left as is.

**Reviewer_2_Technical_Comment_103**: l.988 model is can -> model can

Edited accordingly.

**Reviewer_2_Technical_Comment_104**: Figure 11: * l.993 "black is zero"? On the colour map it is yellow? * red (-0.3) and pink (0.7) are difficult to distinguish, maybe consider a change in the colour map.

Black is zero, black is not yellow, it is stated as such in the legend. We provide the actual datasets for individual inspection, so we decided to leave the color scheme as is.

**Reviewer_2_Technical_Comment_105**: Figure 12: consider labelling panels (a)-(d) instead of using top row, bottom row, top left and bottom left.

We prefer the top/bottom, left/right referencing. We don't reference every panel in the plot in the text, otherwise this would be a good suggestion.

---

## Author Response (AR3)

**REVIEWER REPORT #1**

**General Comment from Reviewer 1 and 2:** (Reviewer #1) Previous points that did not seem to be addressed, at least based on the response letter. (Reviewer #2) * I noticed that responses to Reviewer_1_Minor_Comment_027, 028 and 029, and Reviewer_2_General_Comment_006 and 007 are missing.

> An honest oversight. We provide responses to the previous reviewer comments as below.

> **Reviewer_1_Minor_Comment_027**: Fig. 2. In case B, shouldn't 0.25 be in the 2nd row of the 3rd column, with a zero at the 1st row? Also, can logging be applied to other age-classes or just the last one? If multiple classes can be disturbed, then it may be worth showing such example too (or replacing the single-patch disturbance with a multi-patch disturbance example).

>> In Fig. 2, Case B, the oldest age class (last column of matrix) only has 1 element (1 row). As such, the position of the element in the matrix does not change, so 0.25 does not increment to the next row, which is Null.

>> Logging can be applied to any age class. We specify in the text the there is a priority rule for logging/harvest. Simulated wood harvest tries to meet the prescribed demand (as harvested biomass according to LUHv2) by harvesting biomass from the oldest to the youngest age class. It is possible that all age classes are logged (harvested) if the demand exceeds biomass on all age classes. A sentence in the Section 2.2.2 sub-section LUCLM trys to clarify this, "(2b) wood harvest (i.e., biomass harvest) also occurs in the ranking of oldest to youngest age class until two conditions are met. Timber harvest occurs on each age class until a prescribed harvest mass or harvest area is met."

>> We do not entirely understand what the reviewer means by multi-patch disturbance? If the reviewer refers to situations where fire and wood harvest (for example) both happen in the same year among different age classes, then yes, this can happen. We thought this point was extremely relevant to add comments on and we added the text below during discussion of age class integration with fire and land use.

>> "The disturbance processes of simulated fire, land use change, and land management can occur on multiple age classes at a time. That is, these processes are related but independent. For instance, fire can occur independently on each age class, and each age class would have its own independent estimate of the probability of fire. Wood harvest occurs first on the oldest age class and progressively harvests younger age classes until two demands are met (harvest area and harvest biomass); described in detail in the relevant section below. Clearly, each process influences the other as logging or fire both remove biomass that could be potential fuel for a future fire or biomass for a future harvest. These

relationships are not evaluated here, but are noted for its potential importance. Below, we describe in detail the integration of the age-class module with those the two prominent forms of disturbance: fire, land use change, and land management."

**Reviewer_1_Minor_Comment_028:** Fig. 4. It would be interesting to compare these trajectories for the two age-class approaches (equal bins, unequal bins).

> We agreed and added an additional Panel to Figure 4 to show NPP and Rh vs Age (linear across time as opposed to age class codes) for both Equal-bin and Unequal-bin age class simulations. We thank the reviewer for the recommendation and realize that this version of the figures is more informative and presents the comparisons more succinctly.

**Reviewer_1_Minor_Comment_029**: Fig. 9. These results are a bit surprising given that boreal forests burn frequently. Could this be caused by the zonal averaging, which puts drylands and savannas together with low-disturbance forests in tropical and temperate zones (but not so much in the boreal zone)?

> I would not use the term 'frequent' for fires at boreal latitudes, but perhaps that's relative to a certain baseline? It is unclear what the reviewer is referring to as a surprise finding. If the reviewer is commenting on the small trend due to fire at boreal latitudes (Fig. 9), then I would not expect this to be too surprising. The fire season is short at high latitudes, relative to the tropics for instance. Most fire simulation models underestimate burned area at boreal latitudes, including GlobFIRM, but even still, the burned area fraction at tropical and temperate latitudes is greater than at boreal latitudes based on multiple datasets (*see* Figure 1 and Figure 5 *in* Hanson et al. 2020 https://doi.org/10.5194/gmd-13-3299-2020).

> In regards to zonal averaging of multiple biomes within zonal bands: Yes, good point. I would also expect that the simulated trend in zonal age due to fire and/or land use change is certainly driven by some biomes over others. Certainly, the fire return interval in tropical rainforests is less common than in tropical savannahs, which often have annual fires. The intent of the analysis, summarized in Figure 9, is to evaluate general time-varying zonal patterns of ecosystem age as a function of fire and land use, with emphasis here on simulated age distributions. The simulated statistical trends among distinct zonal bands are informative, even if the analysis lacks attribution to specific biomes. That said, this point by the reviewer deserves a follow up focal analysis on causative drivers of trends in age distributions in specific biomes. The comment is well noted.

**Reviewer_2_General_Comment_006**: It did not become clear to me what exactly is compared in 2.3.2 and 3.1: Are these simulation results from a global simulation? From which? Sage? But if from Sage, why are the FIA data with disturbance, stocking or longing excluded?

We compared the FIA data against idealized Regional simulations results, aggregated to the USFS Divisions. We realized that the sub-section headers probably added to the confusion here because the subsequent section describes the regional simulations. We combined the (former) sections 2.3.2 and 2.3.3 and renamed as, "2.3.2 Examining age dynamics: qualitative evaluation of regional simulations against U.S. Forest Inventory Analysis (FIA) data to assess simulated changes in stand structure and ecosystem function".

The regional simulations do not prescribe logging or simulate fire. Instead, we impose a 5% fractional disturbance, as described in the (new) section 2.3.2, sub-heading 'Regional Simulations'. We describe our reasoning for this idealized simulation as, "The intent of the setup was to ensure that each grid cell maintained fractional area in every age class for each year of the simulation and avoided situations in which age classes were only present in 'bad years', or when growing conditions were poor."

For clarity, we rephrased the text describing exclusion of some FIA plot data as follows, "We only included plots ... with no history of major disturbance, stocking, or logging (DSTRBCD=0, TRTCD1=0), *which could alter natural patterns of tree density versus age and confound the comparison to simulated data.*" (italics for emphasis here only).

**Reviewer_2_General_Comment_007**: Figure 3 and 4: I would appreciate to also have Figure 4 for the 10-year age-widths, since this is what is used in the global simulations. Also, could for ease of readability maybe all panels with unequal age-widths start with the youngest age-class? Furthermore, it might increase comparability when changing the x-axis to show linearly increasing years instead of the classes and then to place the boxes for the different age-classes at age-class mean ages. This would particularly underline the differences in the NEP dynamics among the different age-class setups. Even more so, if the two age-class setups would be integrated in one plot/panel for each of the depicted variables instead of having separate panels with differing x-axis.

We agreed overall and have updated Figures 3 and 4. We thank the reviewer for the recommendation and realize that this version of the figures is more informative and presents the comparisons more succinctly.

**Reviewer_1_Specific_Comment_002:** I understand that the authors did not use these options, but in this case either delete the text in line 101–102, or at least make it clear upfront that these will not be discussed.

We added text as below to clarify, italics for emphasis here only.

"The model known as LPJ-wsl v2.0 is based on LPJmL v3.0, but includes modifications to managed lands that now includes modeling gross land cover transitions, forest age

cohorts, and also a modification that include permafrost and wetland methane*; the permafrost and wetland modules were not used in this study."*

**Reviewer_1_Specific_Comment_006:** I think the authors' response provides some useful examples on how survivorship is handled in the model, and could be incorporated in the text.

We added the following text (italics for emphasis here only) into Section 2.2.2 when discussing survivor trees.

"It is possible to have so called 'survivor' trees on the youngest age class that then skews the age-height distribution of the age class. *The model does not assume any structure of survivor trees. Instead, survivor trees occur as a function of the underlying process. For example, if a fire occurs on a stand, but the fire does not burn all the PFTs, then there will be survivor PFTs on the stand. Both fire and wood harvest (below) are simulated based on fractional area, and it is the fractional area, specifically, that gets reset to a young age class.*"

**Reviewer_1_Specific_Comment_007:** Same, the authors could complement the manuscript with the text from their response, as it at least provides some additional possible explanations for the observed discrepancies.

We integrated our prior response into the text as below. Bold to emphasize changes to the text, here only.

"However, LPJ-wsl v2.0 is a big-leaf, single-canopy model **that include space-filling 'packing' constraints on stem density, based on allometric rules for size and height of PFTs**. Also the model does not represent multiple PFT cohorts in an age class, or more simply, it does not represent vertical heterogeneity **such as understory growth that would otherwise increase stem density**. As such, and under the current model architecture and associated assumptions, the **exact** cause of the mis-match is unclear."

**Reviewer_1_Specific_Comment_010:** I could have been clearer in my previous iteration. The opening sentence of section 3.2.2 (Line 479–480 of the current version) implies that a 30-year recovery window for NEP is unrealistic, but the authors did not provide any independent observation to support that a 5-year recovery window is more adequate. I suggest to drop the first sentence, or to provide some support to this claim.

In Section 3.2.2 we added text throughout to clarify when we refer to gridcell-level fluxes. Perhaps 'unrealistic' is not the term we should have used. We changed the context of the text in reference to refer to 'model artefact' because the simulated pattern of 30-years of NEP recovery is purely a consequence of model construct. What we mean by that is that a patch-based model is more like reality, where the full 'grid' of space is an explicit representation of unique patches of ecosystem. It could be representation like Reimann's sum, where each bar is a patch and it approximates a fraction of unique ecosystem states, itself an interesting way to conceptualize patch models. Whether or not

the recovery times themselves are accurate is less concerning at this point. The growth rates and recovery trajectories will have to be optimized, ideally, to observed patterns, but this is beyond the scope of this paper. We added the following text to the end of section 3.2.2 to clarify,

"The VTFT module also uses the mean-individual approximation but stand dynamics are always allowed to occur in natural progression and the relatively small age widths (10-years) ensure that stand age dynamics (NEP-age trajectories in Figures 3 and 4) most evident in the first 50 years are discretely modeled. To reiterate, we think that the simulated flux dynamics in the no-age simulation is a pure model artefact. What we mean by that is that a patch-based (age class) model is more like reality, where the full 'grid' of space is an explicit representation of unique patches of ecosystem. Whether or not the recovery times themselves are accurate (30-years vs 5-years) is less concerning at this point. The growth rates and recovery trajectories will likely have to be optimized, ideally, to observed patterns, but this is beyond the scope of this paper."

**Reviewer_1_Specific_Comment_012:** The authors could add a sentence or two in the manuscript to address the need to account for parameter and process uncertainties.

We added the following paragraph at the end of the Discussion Section "4.3 Forecasting Demographic Effects with a Simplified Statistical Model":

"A last note on emulators. Useful statistical emulators have fidelity to the underlying process model, but such emulators often cannot address uncertainty from parameter values that are often fixed in the underlying process or uncertainty in process representation. In an ideal world, the statistical parameters for climate sensitivity and stand age, for instance, would be constrained by uncertainty simulations that are themselves bounded to a realistic range of parameter values in the process model (Zaehle et al. 2005) and alternate representations of ecosystem processes (Forkel et al. 2016)."

**REVIEWER REPORT #2**

**L 22:** How many are "a few"? Please, be specific.

We replaced 'few' with 'two'.

**L 91:** Substitute "prior studies" for "previous studies" to avoid repetition.

We edited the text accordingly.

**L 92:** Move "are presented" to the end of the sentence for better understanding.

We edited the text accordingly.

**L 120:** Is mortality also formulated as dependent on tree size? This formulation will link to competition by space among individuals.

> Mortality is not dependent on size, per se, but there is a space-filling constraint that results in mortality via density reduction. We clarified as below.

> Mortality occurs as in the original version of LPJ, "...a result of light competition, low growth efficiency, a negative annual carbon balance, heat stress, or when PFT bioclimatic limits are exceeded for a period of time" (Sitch et al. 2003).

**L 134 and over the text:** Replace ageclass and ageclasses with age-class and age-classes or with age class and age classes if preferred.

> We changed all instances to "age class(es)", but kept "age-class(es)" when used as a adjective.

**L 346:** Correct "by products".

> We specifically mean 'byproducts' (also written as 'by-products') to refer to "secondary products made in the manufacture of something else."

**L 425:** Change pft for PFT.

> We edited the text accordingly.

**L 431:** Remove "it" from the sentence as: "Changes in species composition over time do occur and can add…"

> We edited the text accordingly.

**L 645-647:** This sentence is a bit difficult to read. Consider break it or rephrase.

> We agreed and rephrased the text in reference as below.

> " In some geographic locations, it is certainly possible that our wood harvest priority rules (defined by harvesting oldest age class first) might lead to simulated stand ages that are younger than observed stand ages if other harvest rules were applied in real life. For example, if there are a mandates to preserve old-growth forests, then logging might preferentially occur on young or mid-aged forests, leaving older age class forests unharvested."

**Table 2:** Great summary of the study!

> Thanks!

**Figure 3:** Please, check the y-axes for Stem Density and NEP. They do not match between unequal age widths and 10-yrs age widths. Labels from the y-axes of the right column plots could also be removed.

> For clarity, we redesigned Figure 3 to be a joint figure, which puts both types of age class simulations (equal-bin and unequal-bin age classes) on the same plot. We thank the reviewer for the recommendation and realize that this version of the figures is more informative and presents the comparisons more succinctly.

**Figure 10:** Plots have different dimensions. Please, expand the width of the right column plots and if possible, make the axes comparable. The legend is duplicated and could be placed at the bottom of the figure.

> We edited the figure accordingly and kept a single legend.